# A Benchmark on Directed Graph Representation Learning in Hardware Designs

## Abstract

To keep pace with the rapid advancements in design complexity within modern computing systems, directed graph representation learning (DGRL) has become crucial, particularly for encoding circuit netlists, computational graphs, and developing surrogate models for hardware performance prediction. However, DGRL remains relatively unexplored, especially in the hardware domain, mainly due to the lack of comprehensive and user-friendly benchmarks. This study presents a novel benchmark comprising five hardware design datasets and 13 prediction tasks spanning various levels of circuit abstraction. We evaluate 21 DGRL models, employing diverse graph neural networks and graph transformers (GTs) as backbones, enhanced by positional encodings (PEs) tailored for directed graphs. Our results highlight that bidirected (BI) message passing neural networks (MPNNs) and robust PEs significantly enhance model performance. Notably, the top-performing models include PE-enhanced GTs interleaved with BI-MPNN layers and BI-Graph Isomorphism Network, both surpassing baselines across the 13 tasks. Additionally, our investigation into out-of-distribution (OOD) performance emphasizes the urgent need to improve OOD generalization in DGRL models. This benchmark, implemented with a modular codebase, streamlines the evaluation of DGRL models for both hardware and ML practitioners.[1]

## 1 Introduction

Directed graphs, where edges encode directional information, are widely utilized as data models in various applications, including email communication Kossinets et al. (2008); Khrabrov & Cybenko (2010), financial transactions Gale & Kariv (2007); Chinazzi & Fagiolo (2015); Tiwari et al. (2021), and supply chains Surana et al. (2005); Kaur et al. (2006); Wagner & Neshat (2010). Notably, hardware designs can be represented as directed graphs, such as circuit netlists Hachtel & Somenzi (2005); Vladimirescu (1994), control and data flow graphs Cummins et al. (2021); Wu et al. (2022b); Bai et al. (2023); Ye et al. (2024), or computational graphs Zhang et al. (2021a); Phothilimthana et al. (2023), often exhibiting unique properties. These graph structures reflect restricted connection patterns among circuit components or program operation units, with directed edges encapsulating long-range directional and logical dependencies.

Recently, employing machine learning (ML) to assess the properties of hardware designs via their directed graph representations has attracted significant attention Wu & Xie (2022); Bai et al. (2023); Dong et al. (2023); Li et al. (2020c); Ma et al. (2019b); Bücher et al. (2022); He et al. (2021); Guo et al. (2022); Phothilimthana et al. (2023). Traditional simulation-based methods often require considerable time (hours or days) to achieve the desired accuracy in assessing design quality Zhao et al. (2017); Dai et al. (2018); Wu et al. (2021a; 2022b), substantially slowing down the hardware development cycle due to repeated optimization-evaluation iterations. In contrast, ML models can serve as faster and more cost-effective surrogates for simulators, offering a balanced alternative between simulation costs and prediction accuracy Mirhoseini et al. (2021); Al-Hyari et al. (2021); Wu et al. (2021a); Chen et al. (2018); Jia et al. (2020); Cakir & Malik (2018); Dudziak et al. (2020); Cao et al. (2022); Liu et al. (2021); Wang et al. (2020); Wu et al. (2023). Such an approach is promising to expedite hardware evaluation, especially given the rapid growth of design complexity in modern electronics and computing systems Society.

---

[1]Document (PDF version) and code for the toolbox are provided in supplementary materials.

Despite the promising use cases, developing ML models for reliable predictions on directed graphs, particularly within hardware design loops, is still in its early stages, largely due to the lack of comprehensive and user-friendly benchmarks. Existing studies in the ML community have primarily focused on undirected graphs, utilizing Graph Neural Networks (GNNs) Kipf & Welling (2016); Xu et al. (2019); Veličković et al. (2018) or Graph Transformers (GTs) Rampášek et al. (2022); Kreuzer et al. (2021); Ying et al. (2021); Min et al. (2022). Among the limited studies on directed graph representation learning (DGRL) Zhang et al. (2021b); Tong et al. (2020b;a); Geisler et al. (2023), most have only evaluated their models for node/link-level predictions on single graphs in domains such as web networks, or financial networks He et al. (2024). These domains exhibit very different connection patterns compared to those in hardware design. To the best of our knowledge, CODE2 in the Open Graph Benchmark (OGB) Hu et al. (2020) is the only commonly used benchmark that may share some similarities with hardware data. However, the graphs in CODE2 are intermediate representations (IRs) of Python programs, which may not fully reflect the properties of data in hardware design loops.

Numerous DGRL models for hardware design tasks have been developed by domain experts. While promising, hardware experts tend to incorporate domain-specific insights with off-the-shelf GNNs (e.g., developing hierarchical GNNs to mimic circuit modules Wu et al. (2022b); Dong et al. (2023) or encoding circuit fan-in and fan-out in node features Ren et al. (2020); Alrahis et al. (2021b); Vasudevan et al. (2021)), with limited common design principles investigated in model development. In contrast, state-of-the-art (SOTA) DGRL techniques proposed by the ML community lack thorough investigation in these tasks. These techniques potentially offer a more general and effective manner of capturing data patterns that might be overlooked by domain experts.

**Present Benchmark.** This work addresses the aforementioned gaps by establishing a new benchmark consisting of representative hardware design tasks and extensively evaluating various DGRL techniques for these tasks. On one hand, the evaluation results facilitate the identification of commonly useful principles for DGRL in hardware design. On the other hand, the ML community can leverage this benchmark to further advance DGRL techniques.

Specifically, our benchmark collects five hardware design datasets encompassing a total of 13 prediction tasks. The data spans different levels of circuit abstraction, with graph sizes reaching up to 400+ nodes per graph across 10k+ graphs for graph-level tasks, and up to 50k+ nodes per graph for node-level tasks (see Fig. 1 and Table. 1). We also evaluate 21 DGRL models based on 8 GNN/GT backbones, combined with different message passing directions and various enhancements using positional encodings (PEs) for directed graphs Geisler et al. (2023). PEs are vectorized representations of node positions in graphs and have been shown to improve the expressive power of GT/GNNs for undirected graphs Wang et al. (2022a); Huang et al. (2024); Lim et al. (2022); Rampášek et al. (2022). PEs for directed graphs are still under-explored Geisler et al. (2023), but we believe they could be beneficial for hardware design tasks that involve long-range and logical dependencies.

Our extensive evaluations provide significant insights into DGRL for hardware design tasks. Firstly, bidirected (BI) message passing neural networks (MPNNs) can substantially improve performance for both pure GNN encoders and GT encoders that incorporate MPNN layers, such as GPS Rampášek et al. (2022). Secondly, PEs, only when used stably Wang et al. (2022a); Huang et al. (2024), can broadly enhance the performance of both GTs and GNNs. This observation contrasts with findings from undirected graph studies, particularly in molecule property prediction tasks, where even unstable uses of PEs may improve model performance Dwivedi et al. (2023); Kreuzer et al. (2021); Lim et al. (2022); Rampášek et al. (2022). Thirdly, GTs with MPNN layers typically outperform pure GNNs on small graphs but encounter scalability issues when applied to larger graphs.

With these insights, we identify two top-performing models: GTs with BI-MPNN layers (effective for small graphs in the HLS and AMP datasets) and the BI-Graph Isomorphism Network (GIN) Xu et al. (2019), both enhanced by stable PEs. These models outperform all baselines originally designed by hardware experts for corresponding tasks, across all 13 tasks. Notably, this work is the first to consider GTs with BI-MPNN layers and using stable PEs in DGRL, so the above two models have novel architectures essentially derived from our benchmarking effort.

Furthermore, recognizing that hardware design often encounters out-of-distribution (OOD) data in production (e.g., from synthetic to real-world Wu et al. (2022b), before and after technology mapping Wu et al. (2023), inference on different RISC-V CPUs He et al. (2021)), for each dataset we

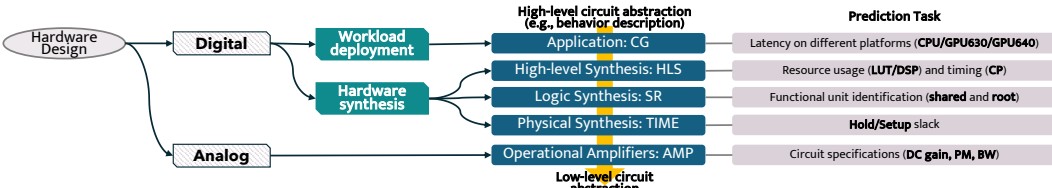

Figure 1: Coverage of Datasets/Tasks.

| | High-level Synthesis (HLS) Wu et al. (2022b) | Symbolic Reasoning (SR) Wu et al. (2023) | Pre-routing Timing Prediction (Time) Guo et al. (2022) | Computational Graph (CG) Zhang et al. (2021a) | Operational Amplifiers (AMP) Dong et al. (2023) |
|---|---|---|---|---|---|
| Type | digital | digital | digital | digital | analog |
| Level | graph | node | node | graph | graph |
| Target | regression | classification | regression | regression | regression |
| Task | LUT, DSP, CP | node shared by MAJ and XOR, root node of an adder | hold slack, setup slack | CPU/GPU630/GPU640 | gain, PM, BW |
| Evaluation Metric | mse, r2 | accuracy, f1 recall, precision | mse, r2 | rmse, acc5, acc10 | mse, rmse |
| In-Distribution | CDFG | 24-bit | graph structure | network structure | stage3 |
| Out-of-Distribution | DFG | 32, 36, 48- bit | graph structure | network structure | stage2 |
| # Training Graph | 16570 - 16570 | 1 - 1 | 7 - 7 | 5* - 10000 | 7223-7223 |
| #Train Nodes  average | 95 | 4440 | 29839 | 218 | 9 |
| #Train Nodes  max | 474 | 4440 | 58676 | 430 | 16 |
| # Train Edges  average | 123 | 10348 | 41268 | 240 | 15 |
| # Train Edges  max | 636 | 10348 | 83225 | 487 | 36 |

Table 1: Statistics of selected datasets. In row '# Training graph', we report '# Graph Structures - # Samples'. *: in CG, there are only five unique CNN designs, yet the structure of graphs within each design may vary slightly.

evaluate the methods data with distribution shift to simulate potential OOD challenges. We observe that while ML models perform reasonably well on tasks (8 of 13) with diverse graph structures in the training dataset, they generally suffer from OOD generalization issues on the remaining tasks. This finding highlights the urgent need for future research to focus on improving the OOD generalization capabilities of DGRL models.

Lastly, our benchmark is implemented with a modular and user-friendly codebase, allowing hardware practitioners to evaluate all 21 DGRL models for their tasks with data in a PyG-compatible format Fey & Lenssen (2019), and allowing ML researchers to advance DGRL methods using the collected hardware design tasks.

## 2 RELATED WORK

**Graph Representation Learning as Powerful Surrogate Models.** ML-based surrogate models have been widely adopted in scientific fields Olivier et al. (2021); Zuo et al. (2021) and recently extended in hardware design. Beyond traditional methods Biggs et al. (2022); Xiao et al. (2023a); Mack et al. (2023), graph-learning-based surrogate models for hardware design have already demonstrated effectiveness Wang et al. (2022b); Ustun et al. (2020); Wu et al. (2022b; 2021a); Bai et al. (2023); Ren et al. (2020); Zhang et al. (2019a); Li et al. (2020c); Ma et al. (2019b); Vasudevan et al. (2021); Bücher et al. (2022); Alrahis et al. (2021b); Lu et al. (2023); Qin et al. (2024); Sohrabizadeh et al. (2023; 2022)Xiao et al. (2023b) , several aspects warrant further investigation. First, existing studies often rely on task-specific heuristics to encode circuit structural information Ma et al. (2019b); Ren et al. (2020); Alrahis et al. (2021b); Bücher et al. (2022); Vasudevan et al. (2021); Mirhoseini et al. (2021), hindering the migration of model-design insights from one task to an even closely related task. Second, the majority of these studies conduct message passing of GNNs along edge directions, with few considering BI implementation He et al. (2021); Guo et al. (2022), and there is an absence of a comparative analysis of different DGRL approaches. Third, the designed models are often trained and tested within similar data distributions Alrahis et al. (2021b); Zhao & Shamsi (2022); He et al. (2021), lacking systematic OOD evaluation for new or more complicated designs. Hence, it is imperative to establish a comprehensive benchmark to compare different DGRL approaches for hardware design tasks.

**Methods for DGRL.** NN architectures for DGRL can be classified into three types: spatial GNNs, spectral GNNs, and transformers. Spatial GNNs use graph topology as inductive bias, some employ bidirected message passing for regular directed graphs Jaume et al. (2019); Wen et al. (2020); Kollias

et al. (2022); Rossi et al. (2024), others use asynchronous message passing exclusively designed for directed acyclic graphs (DAGs) Zhang et al. (2019b); Dong et al. (2022); Thost & Chen (2020). Spectral GNNs generalize the ideas of Fourier transform and corresponding spectral convolution from undirected to directed graphs Furutani et al. (2020); He et al. (2022); Fiorini et al. (2023); Zhang et al. (2021b); Singh et al. (2016); Ma et al. (2019a); Monti et al. (2018); Tong et al. (2020b); Koke & Cremers (2023); Transformers with attention mechanism reply on designing direction-aware PEs to capture directed graph topology. This benchmark is the first to consider combining transformers with MPNN layers for DGRL, extending the ideas in Rampášek et al. (2022). Regarding the choices of PEs, most studies are on undirected graphs Wang et al. (2022a); Huang et al. (2024); Lim et al. (2022); Dwivedi et al. (2022a). For directed graphs, the potential PEs are Laplacian eigenvectors of the undirected graphs by symmetrizing the original directed ones Dwivedi et al. (2023), singular vectors of adjacency matrices Hussain et al. (2022) and the eigenvectors of Magnetic Laplacians Shubin (1994); Fanuel et al. (2017; 2018); Geisler et al. (2023). No previous investigate benefit for DGRL from stably incorporating PE Wang et al. (2022a); Huang et al. (2024), and we are the first to consider stable PEs for DGRL.

**Existing Relevant Benchmarks.** Dwivedi et al. Dwivedi et al. (2022b) benchmark long-range reasoning of GNNs on undirected graphs; PyGSD He et al. (2024) benchmarks signed and directed graphs, while focusing on social or financial networks. We also compare all the methods for directed unsigned graphs in PyGSD and notice that the SOTA spectral method therein - MagNet Zhang et al. (2021b) still works well on node-level tasks on a single graph (SR), which shares some similar insights. The hardware community has released graph-structured datasets from various development stages to assist surrogate model development, including but not limited to NN workload performance Zhang et al. (2021a); Phothilimthana et al. (2023), CPU throughput Chen et al. (2019); Sỳkora et al. (2022); Mendis et al. (2019), resource and timing in HLS Wu et al. (2022b); Bai et al. (2023), design quality in logic synthesis Chowdhury et al. (2021), design rule checking in physical synthesis Guo et al. (2022); Chai et al. (2023); Xun et al. (2024); Chhabria et al. (2024), and hardware security Yu et al. (2021). In addition to datasets, ProGraML Cummins et al. (2021) introduces a graph-based representation of programs derived from compiler IRs (e.g., LLVM/XLA IRs) for program synthesis and compiler optimization. Very recently, Google launched TPUgraph for predicting the runtime of ML models based on their computational graphs on TPUs Phothilimthana et al. (2023). Our CG dataset includes computational graphs of ML models, specifically on edge devices.

## 3 DATASETS AND TASKS

This section introduces the five datasets with thirteen tasks used in this benchmark. The datasets cover both digital and analog designs, selected to represent different circuit abstraction levels. From a bottom-up perspective, we include operational amplifiers (AMP) at the device level, timing prediction (TIME) in physical synthesis, symbolic reasoning (SR) in logic synthesis, performance prediction in high-level synthesis (HLS), and workload mapping on hardware platforms (CG), as illustrated in Fig. 1. Table 1 displays the statistics of each dataset. Next, we briefly introduce the five datasets, with details provided in Appendix. D. Although these datasets are generated by existing studies, we offer modular pre-processing interfaces to make them compatible with PyTorch Geometric and user-friendly for integration with DGRL methods.

**High-Level Synthesis (HLS) Wu et al. (2022b):** Constructing C/C++ into graphs originates form Xiao et al. (2017). The HLS dataset collects IR graphs of C/C++ code after front-end compilation Alfred et al. (2007), and provides post-implementation performance metrics on FPGA devices as labels for each graph, which are obtained after hours of synthesis with Vitis vit and implementation with Vivado viv. The labels to predict include resource usage, (i.e., look-up table (LUT) and digital signal processor (DSP)), and the critical path timing (CP). See Appendix. D.1 for graph input details.

*Significance:* The HLS dataset is crucial for testing NNs' ability to accurately predict post-implementation metrics to accelerate design evaluation in the stage of HLS.

*OOD Evaluation:* For training and ID testing, we use control data flow graphs (CDFG) that integrate control conditions with data dependencies, derived from general C/C++ code; As to OOD cases, we use data flow graphs (DFG) derived from basic blocks, leading to distribution shifts.

**Symbolic Reasoning (SR) Wu et al. (2023):** The SR dataset collects bit-blasted Boolean networks (BNs) (unstructured gate-level netlists), with node labels annotating high-level abstractions on local graph structures, e.g., XOR functions, majority (MAJ) functions, and adders, generated by the logic

synthesis tool ABC Brayton & Mishchenko (2010). Each graph supports two tasks: root nodes of adders, and nodes shared by XOR and MAJ functions. See Appendix. D.2 for detailed input encoding and label explanation.

*Significance:* Reasoning high-level abstractions from BNs has wide applications in improving functional verification efficiency Ciesielski et al. (2019) and malicious logic identification Mahzoon et al. (2019). GNN surrogate models are anticipated to replace the conventional structural hashing and functional propagation Li et al. (2013); Subramanyan et al. (2013) and boost the scalability with significant speedup. For graph ML, due to significant variation in the size of gate-level netlists under different bit widths, SR is an ideal real-world application to evaluate whether GNN designs can maintain performance amidst the shifts in graph scale.

*OOD Evaluation:* We use a 24-bit graph (4440 nodes) for training, and 32, 36, 48-bit graphs (up to 18096 nodes) for ID testing, derived from carry-save-array multipliers before technology mapping. OOD testing data are multipliers after ASAP 7nm technology mapping Xu et al. (2017) with the same bits.

**Pre-routing Timing Prediction (TIME) Guo et al. (2022):** The TIME dataset collects real-world circuits with OpenROAD ope (b) on SkyWater 130nm technology sky. The goal is to predict slack values at timing endpoints for each circuit design by using pre-routing information. Two tasks are considered: hold slack and setup slack. Details are provided in Appendix. D.3.

*Significance:* In physical synthesis, timing-driven placement demands accurate timing information, which is only available after routing. Repetitive routing and static timing analysis provide accurate timing but are prohibitively expensive. ML models that precisely learn routing behaviors and timing computation flows are highly expected to improve the efficiency of placement and routing.

*OOD Evaluation:* We divide ID-OOD based on the difference in graph structures (e.g. 'blabla' and 'xtea' are different circuit designs, allocated into ID or OOD groups). See details in Appendix. D.3.1.

**Computational Graph (CG) Zhang et al. (2021a):** The CG dataset consists of computational graphs of convolutional neural networks (CNNs) with inference latency on edge devices (i.e., Cortex A76 CPU, Adreno 630 GPU, Adreno 640 GPU) as labels. The CNNs have different operator types or configurations, either manually designed or found by neural architecture search (NAS). Details are in Appendix. D.4.

*Significance:* Accurately measuring the inference latency of DNNs is essential for high-performance deployment on hardware platforms or efficient NAS Ren et al. (2021); Shi et al. (2022), which however is often costly. ML-based predictors offer the potential for design exploration and scaling up to large-scale hardware platforms.

*OOD Evaluation:* We split ID-OOD with different graph structures. (e.g. 'DenseNets' and 'ResNets' are CNNs with different structures, allocated into different groups). See Appendix. D.4.1 for details.

**Operational Amplifiers (AMP) Dong et al. (2023):** AMP dataset contains 10, 000 distinct 2- or 3-stage operational amplifiers (Op-Amps). Circuit specifications (i.e. DC gain, phase margin (PM), and bandwidth (BW)) as labels are extracted after simulation with Cadence Spectre spe. Details are in Appendix. D.5.

*Significance:* Analog circuit design is less automated and requires more manual effort compared to its digital counterpart. Mainstream approaches such as SPICE-based circuit synthesis and simulation Vladimirescu (1994), are computationally expensive and time-consuming. If ML algorithms can approximate the functional behavior and provide accurate estimates of circuit specifications, they may significantly reduce design time by minimizing reliance on circuit simulation Afacan et al. (2021).

*OOD Evaluation:* For training and ID testing, we use 3-stage Op-Amps, which have three single-stage Op-Amps in the main feed-forward path). For OOD evaluation, we use 2-stage Op-Amps.

**Extensions** Although the datasets cover different levels of circuit abstraction, there are additional tasks in hardware design worth exploration with DGRL surrogates, as reviewed in Section 2. Our modular benchmark framework allows for easy extension to accommodate new datasets.

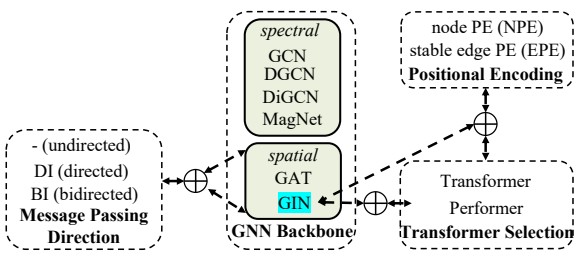

Figure 2: The benchmark considers 21 combinations ('+') of message passing direction, GNN backbone, transformer selection and PE incorporation, covers 10 existing SOTA methods from graph ML community and discovers 2 novel top-performing models (see Table. 2).

| Method | type | layer-wise complexity |
|---|---|---|
| GCN Kipf & Welling (2016) | spectral | $O(\|E\|)$ |
| MagNet Zhang et al. (2021b) | spectral | $O(\|E\|)$ |
| DGCN Tong et al. (2020b) | spectral | $O(\|E\|)$ |
| DiGCN Tong et al. (2020a) | spectral | $O(\|E\|)$ |
| GAT Veličković et al. (2018) | spatial | $O(\|E\|)$ |
| GIN(E) Xu et al. (2019) | spatial | $O(\|E\|)$ |
| EDGNN Jaume et al. (2019) | spatial | $O(\|E\|)$ |
| GPS-T Rampášek et al. (2022) | spatial+transformer | $O(\|V\|^2 + \|E\|)$ |
| GPS-P Choromanski et al. (2020) | spatial+transformer | $O(\|V\| + \|E\|)$ |
| TmD Geisler et al. (2023) | transformer | $O(\|V\|^2)$ |
| BI-GIN(E)+EPE(new) | spatial | $O(\|E\|)$ |
| BI-GPS-T+EPE(new) | spatial+transformer | $O(\|V\|^2 + \|E\|)$ |

Table 2: Existing methods and two top-performing methods highlighted at bottom.

## 4 BENCHMARK DESIGN

### 4.1 DESIGN SPACE FOR DIRECTED GRAPH REPRESENTATION LEARNING

In this section, we introduce the DGRL methods evaluated in this benchmark. Our evaluation focuses on four design modules involving GNN backbones, message passing directions, transformer selection, and PE incorporation, illustrated in Fig. 2. Different GNN backbones and transformer adoptions cover 10 methods in total with references in Tab. 2. We also consider their combinations with different message-passing directions and various ways to use PEs, which overall gives 21 DGRL methods.

For GNNs, we consider 4 spectral methods, namely GCN Kipf & Welling (2016), DGCN Tong et al. (2020b), DiGCN Tong et al. (2020a) and MagNet Zhang et al. (2021b), where the latter three are *SOTA spectral GNNs* specifically designed for DGRL He et al. (2024); For spatial GNNs, we take GIN Xu et al. (2019) and Graph Attention Network (GAT) Veličković et al. (2018), which are the most commonly used MPNN backbones for undirected graphs. We evaluate the combination of GCN, GIN and GAT with three different message-passing directions: a) 'undirected'(-) treats directed graphs as undirected, using the same NN parameters to perform message-passing along both forward and reverse edge directions; b) 'directed'(DI) only passes messages exclusively along the forward edge directions; c) 'bidirected'(BI) performs message passing in both forward and reverse directions with distinct parameters for either direction. The other GNNs (DGCN, DiGCN and MagNet) adopt spectral convolution that inherently considers edge directions. The combination of 'BI' with spatial GNN layers gives *the state-of-the-art spatial GNNs* for DGRL, i.e., EDGNN Jaume et al. (2019).

For GTs, we adopt the eigenvectors of the graph Magnetic Laplacian (MagLAP) matrix as the PEs of nodes Furutani et al. (2020); Shubin (1994), as they are directional-aware. The MagLap matrix $L_q$ is a complex Hermitian matrix with parameter $q \in [0, 1)$ named potential, which is treated as a hyper-parameter in our experiments. Note that when $q = 0$, MagLap degenerates to the symmetric Laplacian matrix $L_0$ as a special case. See Appendix B for a brief review of MagLap. The GT with the MagLap PEs attached to node features gives *the SOTA GT model* for DGRL, named TmD for brevity, proposed in Geisler et al. (2023). GPS Rampášek et al. (2022) is a GT model with MPNN layers Hamilton et al. (2017); Gilmer et al. (2017) interleaving with transformer layers Vaswani et al. (2017), originally proposed for undirected graphs. We extend GPS to directed graphs by using MagLap PEs for transformer layers and DI/BI message passing in its MPNN layers. Hence, GPS is also an extension of TmD by incorporating MPNN layers. As transformers may not scale well on large graphs, we evaluate vanilla transformer layers and their lower-rank approximation Performer Kreuzer et al. (2021) for efficient computation, named as GPS-T and GPS-P, respectively.

### 4.2 STABLE DIRECTION-AWARE POSITIONAL ENCODINGS

Recent studies on undirected graphs have demonstrated that models by naively attaching PEs to node features may suffer from an issue of instability because small changes in the graph structure may cause big changes in

$$\mathbf{NPE} = [\text{Re}\{\boldsymbol{V_q}\}, \text{Im}\{\boldsymbol{V_q}\}]$$

$$\mathbf{EPE} = \rho(\text{Re}\{\boldsymbol{V_q}\text{diag}(\kappa_1(\lambda))\boldsymbol{V_q}^\dagger\}, ..., \text{Re}\{\boldsymbol{V_q}\text{diag}(\kappa_m(\lambda))\boldsymbol{V_q}^\dagger\},$$
$$\text{Im}\{\boldsymbol{V_q}\text{diag}(\kappa_1(\lambda))\boldsymbol{V_q}^\dagger\}, ..., \text{Im}\{\boldsymbol{V_q}\text{diag}(\kappa_m(\lambda))\boldsymbol{V_q}^\dagger\})$$

Table 3: Different ways to handle PEs. NPE directly concatenates the eigenvectors to node features. EPE computes an edge-level PE in a permutation equivariant and stable manner.

PEs Wang et al. (2022a); Huang et al. (2024); Lim et al. (2022). We name this way of using PEs as node-PE (NPE). The instability provably leads to undesired OOD generalization Huang et al. (2024). We think this is also true for directed graphs and indeed observe the subpar model performance with NPE.

Therefore, besides NPE, we also consider a stable way of incorporating PEs for DGRL, namely 'edge PE' (EPE), inspired by Wang et al. (2022a); Huang et al. (2024). Specifically, we take the smallest $d$ eigenvalues $\lambda_q \in \mathbb{R}^d$ and their corresponding eigenvectors $\boldsymbol{V}_q \in \mathbb{C}^{|V| \times d}$ from $\boldsymbol{L}_q$. Then, permutation equivariant and smooth functions $\kappa : \mathbb{R}^d \to \mathbb{R}^d$ (we have $m$ many $\kappa$) are adopted to map $d$-dim eigenvalues to an embedding vector. It will then produce $m$ many $n$-by-$n$ matrices $\boldsymbol{V}_q \text{diag}(\kappa_1(\lambda))\boldsymbol{V}_q^{\dagger}, ..., \boldsymbol{V}_q \text{diag}(\kappa_m(\lambda))\boldsymbol{V}_q^{\dagger}$ by doing matrix multiplication between eigenvalue embeddings and eigenvectors. Lastly, these $n$-by-$n$ matrices are concatenated and passed to $\rho$, a permutation equivaraint and smooth function (e.g., standard GNNs). The output **EPE** is of dimension $n \times n \times d$, representing PE-refined features of edges (node pairs). In a high level, the key to achieve stability here is the permutation equivariance as well as smoothness of $\kappa$, which ensures smooth weights $\kappa(\lambda)$ of eigenvectors' inner product across different eigenvalues (see similar argument of Theorem 3.1 in Huang et al. (2024)). Finally, in GTs, **EPE**$_{u,v}$ is added to the attention weight between nodes $u$ and $v$ as a bias term at each attention layer.

We note that PEs can also be used in more than GTs, to improve the expressive power of GNNs Li et al. (2020b); Ying et al. (2021); Lim et al. (2022); Huang et al. (2024). We leverage this idea and enhance the GNN models for directed graphs with PEs. Specifically, for the GNNs NPE will use **NPE**$_v$ as extra node features of node $v$ while EPE will use **EPE**$_{u,v}$ as extra edge features of edge $uv$ if $uv$ is an edge.

The incorporation with EPE helps discover a novel GT model for directed graphs, i.e., GT with BI-MPNN layers enhanced by EPE, abbreviated as BI-GPS+EPE. We also make the first attempt to combine GNNs with PEs for directed graphs, which yields the model BI-GIN(E)+EPE.

### 4.3 HYER-PARAMETER SPACE AND TUNING

For each combination of DGRL method in this benchmark, we perform automatic hyper-parameter tuning with RAY Liaw et al. (2018) adopting Tree-structured Parzen Estimator (TPE) Watanabe (2023), a state-or-the-art bayesian optimization algorithm. The hyper-parameter space involves searching batch size, learning rate, number of backbone layers, dropout rate in MPNN and MLP layers, hidden dimension, and MLP layer configurations. The detailed hyper-parameter space of each model is shown in Appendix. E.2. We auto-tune the hyper-parameters with seed 123 with 100 trial budgets and select the configuration with the best validation performance. Then, the selected configuration is used for model training and testing ten times with seeds $0 - 9$ and the average is reported as the final performance.

## 5 MODULAR TOOLBOX

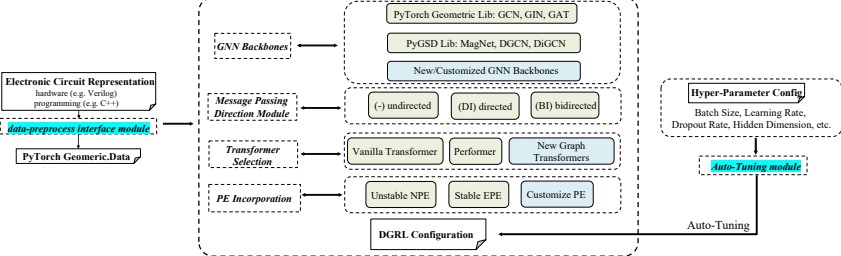

Figure 3: Illustration of the directed graph representation learning (DGRL) toolbox.

We develop a highly modular toolbox involving designing, auto hyper-parameter tuning, and evaluation for DGRL methods. The framework is shown in Fig. 3. The toolbox comes with the 21 DGRL methods, allowing practitioners to evaluate them on any new task with data compatible with PyTorch Geometric (PyG) Fey & Lenssen (2019). This may be used even beyond hardware design applications. Users can also customize new methods. Once the method is configured, auto hyper-parameter tuning can be performed using RAY Liaw et al. (2018). The toolbox also includes the above 5 datasets

| Distribution | | | | | | | | | | | | | | | In-Distribution (ID) / Out-of-Distribution (OOD) | | | | | | | | | | | | | |
|---|---|---|---|---|---|---|---|---|---|---|---|---|---|---|---|---|---|---|---|---|---|---|---|---|---|---|---|---|
| Dataset | | HLS | | | AMP | | | SR | | TIME | | CG | | | HLS | | | AMP | | | SR | | TIME | | CG | | | |
| Task | | DSP | LUT | CP | gain | PM | BW | share | root | hold | setup | CPU | GPU630 | GPU640 | DSP | LUT | CP | gain | PM | BW | share | root | hold | setup | CPU | GPU630 | GPU640 |
| **Spectral** | DGCN | 15.0 | 15.0 | 15.0 | 14.0 | 8.0 | 15.0 | 10.0 | 9.0 | 15.0 | 5.5 | 13.0 | 15.0 | 14.0 | 15.0 | 14.0 | 15.0 | 14.0 | 3.0 | 15.0 | 7.5 | 5.0 | 15.0 | 7.0 | 13.3 | 11.7 | 11.2 |
| | DiGCN | 12.0 | 14.0 | 13.0 | 12.0 | 9.0 | 14.0 | 8.5 | 7.8 | 13.5 | 15.0 | 14.0 | 14.0 | 15.0 | 12.5 | 15.0 | 14.0 | 9.0 | 4.0 | 14.0 | 9.0 | 5.0 | 13.5 | 14.0 | 13.2 | 13.2 | 13.3 |
| | MagNet | 7.0 | 7.0 | 10.5 | 8.0 | 11.0 | 8.0 | 1.8 | 2.0 | 11.0 | 11.5 | 1.3 | 1.3 | 4.7 | 7.0 | 7.0 | 10.5 | 3.0 | 12.0 | 8.0 | 3.5 | 8.8 | 9.0 | 7.0 | 4.2 | 8.2 | 7.3 |
| | GCN | 14.0 | 12.0 | 14.0 | 15.0 | 13.0 | 12.0 | 13.3 | 13.5 | 9.5 | 14.0 | 15.0 | 12.3 | 11.7 | 12.5 | 10.0 | 12.0 | 14.5 | 14.0 | 11.0 | 14.8 | 14.5 | 7.5 | 10.5 | 12.7 | 12.7 | 11.5 |
| | DI-GCN | 13.5 | 13.0 | 12.0 | 11.0 | 3.0 | 13.0 | 15.0 | 15.0 | 11.0 | 13.0 | 11.0 | 11.3 | 12.0 | 14.0 | 11.0 | 13.0 | 12.0 | 7.0 | 13.0 | 13.5 | 11.8 | 10.0 | 8.0 | 11.2 | 11.5 | 12.0 |
| | BI-GCN | 11.0 | 10.5 | 9.0 | 5.0 | 14.0 | 6.0 | 5.5 | 5.3 | 5.0 | 9.0 | 12.3 | 12.3 | 12.3 | 11.0 | 12.5 | 8.0 | 2.0 | 13.0 | 5.0 | 2.3 | 4.8 | 3.0 | 6.5 | 13.2 | 11.3 | 12.5 |
| **Spatial** | GIN | 6.0 | 5.5 | 8.0 | 7.0 | 6.0 | 10.0 | 10.0 | 11.0 | 1.0 | 3.0 | 5.0 | 3.3 | 8.3 | 6.0 | 3.5 | 5.0 | 8.0 | 10.0 | 7.0 | 9.0 | 7.3 | 3.0 | 8.5 | 5.2 | 4.2 | 4.8 |
| | DI-GIN | 2.5 | 4.0 | 6.5 | 9.0 | 10.0 | 7.0 | 6.5 | 4.8 | 3.0 | 2.0 | 5.7 | 8.0 | 3.3 | 2.0 | 2.5 | 7.0 | 10.0 | 5.0 | 12.0 | 6.3 | 9.0 | 6.5 | 7.0 | 3.5 | 5.7 | 4.2 |
| | BI-GIN | 1.0 | 1.0 | 5.0 | 3.0 | 4.0 | 3.0 | 2.8 | 4.8 | 2.0 | 1.0 | 2.3 | 4.7 | 1.0 | 4.5 | 1.0 | 3.0 | 4.0 | 9.0 | 3.0 | 1.5 | 2.0 | 1.0 | 7.5 | 2.3 | 4.5 | 4.0 |
| | GAT | 8.5 | 9.0 | 6.5 | 6.0 | 15.0 | 5.0 | 13.8 | 13.5 | 10.5 | 11.5 | 9.0 | 9.0 | 8.7 | 9.0 | 9.0 | 5.5 | 7.0 | 15.0 | 6.0 | 12.3 | 10.5 | 13.5 | 5.5 | 7.7 | 5.7 | 6.2 |
| | DI-GAT | 10.0 | 10.5 | 10.5 | 10.0 | 12.0 | 9.0 | 11.8 | 10.0 | 13.5 | 10.0 | 10.0 | 10.0 | 10.0 | 10.0 | 12.5 | 11.0 | 11.0 | 6.0 | 10.0 | 11.3 | 10.0 | 13.5 | 8.5 | 6.2 | 5.7 | 7.3 |
| | BI-GAT | 9.0 | 8.0 | 1.0 | 4.0 | 2.0 | 11.0 | 4.0 | 6.3 | 6.5 | 7.5 | 8.0 | 5.3 | 7.0 | 8.0 | 8.0 | 1.5 | 1.0 | 2.0 | 9.0 | 9.8 | 8.5 | 4.5 | 6.5 | 6.2 | 10.7 | 10.5 |
| **Transformer** | GPS-T | 4.0 | 3.0 | 2.5 | 13.0 | 7.0 | 2.0 | -- | -- | -- | -- | -- | -- | -- | 5.0 | 6.0 | 9.5 | 13.0 | 8.0 | 1.0 | -- | -- | -- | -- | -- | -- | -- |
| | DI-GPS-T | 5.0 | 5.5 | 4.0 | 1.0 | 5.0 | 1.0 | -- | -- | -- | -- | -- | -- | -- | 3.0 | 5.0 | 1.5 | 5.0 | 11.0 | 2.0 | -- | -- | -- | -- | -- | -- | -- |
| | BI-GPS-T | 3.0 | 2.0 | 2.0 | 1.0 | 1.0 | 4.0 | -- | -- | -- | -- | -- | -- | -- | 2.5 | 3.5 | 4.5 | 6.0 | 1.0 | 4.0 | -- | -- | -- | -- | -- | -- | -- |
| | GPS-P | -- | -- | -- | -- | -- | -- | 5.5 | 12.0 | 6.5 | 4.0 | 6.3 | 2.0 | 5.3 | -- | -- | -- | -- | -- | -- | 7.8 | 11.3 | 8.0 | 7.5 | 6.2 | 4.2 | 6.2 |
| | DI-GPS-P | -- | -- | -- | -- | -- | -- | 6.5 | 4.8 | 4.0 | 7.5 | 2.7 | 5.7 | 3.0 | -- | -- | -- | -- | -- | -- | 5.8 | 7.5 | 7.5 | 8.0 | 7.5 | 7.0 | 5.8 |
| | BI-GPS-P | -- | -- | -- | -- | -- | -- | 7.8 | 7.5 | 8.0 | 5.5 | 4.7 | 6.0 | 4.3 | -- | -- | -- | -- | -- | -- | 6.8 | 4.3 | 7.5 | 8.0 | 8.5 | 5.2 | 4.2 |

Table 4: Average ranking (↓) of methods across datasets/tasks/metrics on ID and OOD data.

with 13 tasks that can be used to develop new DGRL models. For details please refer to the official document for this toolbox.

# 6 EXPERIMENTS

In this section, we first evaluate DGRL methods combining different GNN backbones, message passing directions, transformer selection, and PE incorporation, across all 5 datasets and 13 tasks, using in-distribution (ID) and out-of-distribution (OOD) testing data.

## 6.1 MAIN RESULTS

The performances of the methods under all evaluation metrics for both in-distribution and out-of-distribution testing across all 13 tasks are reported from Table. 11 to Table. 33 in Appendix. H.1. We summarize the averaged ranking with respect to all evaluation metrics given a task in Table. 4. The details of ranking calculation is in Appendix. G.1. The results tell the following insights:

'Bidirected' (BI) message passing in the MPNN layers significantly boosts the models' performance on three GNN backbones (GCN, GIN, GAT) and one GT backbone (GPS-T): BI-GCN outperforms GCN on 10 out of 13 tasks in both ID and OOD evaluations. Similarly, in ID/OOD evaluations, BI-GIN outperforms GIN in 11/12 out of 13 tasks, BI-GAT outperforms GAT in 11/9 out of 13 tasks and BI-GPS-T outperforms GPS-T in 5/5 out of 6 tasks, respectively.

As to the models, on datasets with small graphs (HLS and AMP), BI-GPS-T consistently delivers excellent results, achieving top-3 performance in 5 out of 6 tasks on both ID and OOD testing data. BI-GIN also demonstrates competitive performance on these datasets. However, for datasets with larger graphs (SR, CG, and TIME), BI-GPS-T encounters a scalability issue. BI-GIN secures top-three performance in 6 out of 7 tasks in both ID and OOD testing data. For the 'shared' and 'root' tasks from the SR dataset and the 'CPU' and 'GPU630' tasks from the CG dataset, MagNet Zhang et al. (2021b) performs best in the ID setting. This is likely because training and testing are conducted on the same graph structures for these specific datasets, reducing the need for significant generalization across different graph structures. This scenario aligns well with the spectral filtering approach used by MagNet. These observations match findings from previous studies on directed networks Zhang et al. (2021b); He et al. (2024). However, MagNet's performance falters in OOD evaluations which ask for the ability to generalize across different graph structures. GPS-P, despite its capability to handle large graphs, delivers only mediocre performance overall. *In conclusion, BI-GPS is well-suited for small (around one hundred nodes) directed graphs. For larger graphs, BI-GIN is efficient and performs well. For tasks where the training and testing data share the same graph structures, one may also attempt to adopt MagNet.*

**Comparing PE-enhanced methods:** We further investigate the impact of different ways of using PEs. We combine NPE or EPE with the top-performing models from the previous section and evaluate BI-GIN+NPE, BI-GIN+EPE, and BI-GPS+EPE. Note that BI-GPS already utilizes NPE. We have chosen not to consider adding PE to MagNet because MagNet only accepts 1-dimensional edge

| Distribution | In-Distribution (ID) | | | | | | | | | | | | | Out-of-Distribution (OOD) | | | | | | | | | | | | |
|---|---|---|---|---|---|---|---|---|---|---|---|---|---|---|---|---|---|---|---|---|---|---|---|---|---|---|
| Dataset | HLS | | | AMP | | | SR | | TIME | | CG | | | HLS | | | AMP | | | SR | | TIME | | CG | | |
| Task | DSP | LUT | CP | gain | PM | BW | share | root | hold | setup | CPU | GPU630 | GPU640 | DSP | LUT | CP | gain | PM | BW | share | root | hold | setup | CPU | GPU630 | GPU640 |
| MagNet | 14.5 | 11.0 | 14.5 | 12.0 | 15.0 | 12.0 | 2.3 | 2.5 | 13.0 | 13.0 | 2.3 | 1.7 | 6.7 | 11.0 | 11.0 | 14.5 | 3.0 | 16.0 | 12.0 | 5.5 | 10.8 | 11.0 | 16.0 | 5.2 | 9.5 | 8.3 |
| BI-GIN(E) | 9.0 | 2.0 | 9.0 | 6.0 | 6.0 | 6.0 | 4.8 | 6.8 | 2.5 | 2.0 | 6.0 | 3.3 | 6.0 | 7.5 | 3.5 | 6.0 | 4.0 | 13.0 | 5.0 | 3.0 | 3.0 | 2.5 | 7.5 | 4.7 | 5.8 | 4.8 |
| BI-GIN(E)+NPE | 5.0 | 4.0 | 5.0 | 5.0 | 13.0 | 5.0 | 3.0 | 6.8 | 5.5 | 5.0 | 8.3 | 5.0 | 5.3 | 9.0 | 4.0 | 1.5 | 7.0 | 8.0 | 7.0 | 2.8 | 3.5 | 5.5 | 12.5 | 6.0 | 4.5 | 6.0 |
| BI-GIN(E)+EPE | 5.0 | 1.0 | 5.0 | 9.0 | 10.0 | 3.0 | 4.0 | 6.8 | 2.0 | 1.0 | 1.0 | 6.7 | 1.7 | 7.0 | 1.0 | 2.5 | 6.0 | 6.0 | 4.0 | 1.5 | 3.0 | 1.0 | 7.5 | 3.8 | 5.5 | 4.7 |
| BI-GPS-T (NPE) | 4.5 | 5.5 | 4.5 | 2.0 | 2.0 | 7.0 | -- | -- | -- | -- | -- | -- | -- | 4.0 | 7.0 | 8.0 | 9.0 | 2.0 | 6.5 | -- | -- | -- | -- | -- | -- | -- |
| BI-GPS-T+EPE | 2.5 | 3.0 | 2.0 | 1.0 | 1.0 | 4.0 | -- | -- | -- | -- | -- | -- | -- | 1.5 | 1.5 | 3.5 | 5.5 | 1.0 | 1.0 | -- | -- | -- | -- | -- | -- | -- |

Table 5: Comparison of competitive methods involving NPE and EPE. The ranking (↓) is based on all the 18 methods in Table 4 plus BI-GIN(E)+NPE, BI-GIN(E)+EPE and BI-GPS-T+EPE.

| dataset (baseline's name) | AMP Dong et al. (2023) (CKTGNN) | | | HLS Wu et al. (2022b) (Hierarchical GNN) | | | SR Wu et al. (2023) (GAMORA) | CG Zhang et al. (2021a) (nn-meter) | | | TIME Guo et al. (2022) (Timer-GNN) |
|---|---|---|---|---|---|---|---|---|---|---|---|
| task | gain | PM | BW | dsp | lut | cp | shared | cpu (average) | | | hold |
| metric | rmse↓ | rmse↓ | rmse↓ | mse↓ | mse↓ | mse↓ | accuracy↑ | rmse↓ | acc5↑ | acc10↑ | r2↑ |
| Baseline | 0.52 | 1.15 | 4.47 | 3.94 | 2.45 | 0.88 | 0.99 | 3.20 | 0.80 | 0.99 | 0.97 |
| BI-GINE+EPE | 0.51±0.07 | 1.14±0.00 | 4.20±0.13 | 2.13±0.08 | 1.73±0.10 | 0.61±0.02 | 0.99±0.00 | 2.79±0.14 | 0.86±0.02 | 0.99±0.01 | 0.99±0.00 |
| BI-GPS-T+EPE | 0.34±0.08 | 1.15±0.00 | 3.79±0.11 | 2.13±0.15 | 1.96±0.13 | 0.60±0.01 | -- | -- | -- | -- | -- |

Table 6: Comparison of BI-GIN+EPE and BI-GPS-T+EPE with baselines specific for each dataset.

weights, limiting its ability to leverage EPE. We provide a summary of the performance data from Table 34 to Table 43 in Appendix H.2 and report the average rankings of the methods for each task. All 18 methods in Table 4, along with the 3 new combinations, are included in the ranking. We detail the results of the most competitive methods in Table 5. For BI-GIN, EPE enhances its performance on 10 out of 13 tasks in the in-distribution (ID) testing data and 11 tasks in the out-of-distribution (OOD) testing data. Conversely, NPE only improves the performance of BI-GIN on 7 tasks in the ID testing and 4 tasks in the OOD testing and performs unstable for the rest tasks. Notably, EPE-enhanced BI-GIN surpasses MagNet on the CPU task in the CG dataset. For BI-GPS-T, EPE improves its performance on all 6 tasks in both ID and OOD testing, while NPE does not yield substantial improvements. This observation contrasts with previous work Rampášek et al. (2022) on undirected graphs for molecular property prediction. *In conclusion, we find that incorporating PEs in a stable way as EPE significantly boosts the performance of different models across the selected tasks and datasets.*

## 6.2 SUMMARY: THE RECIPE FOR DGRL

Through benchmarking various combinations within the design space, we have formulated a design recipe for DGRL methods tailored for encoding hardware data: *The use of 'bidirected' (BI) message passing and stable positional encodings (PE) can significantly enhance model performance. Therefore, we recommend BI-GPS-T+EPE for encoding small graphs and BI-GIN+EPE for large graphs.*

We further compare the two models' performance with the baseline methods proposed by hardware design practitioners specifically for the corresponding tasks in the original papers. Results are shown in Table. 6. The comparison focuses on ID evaluation as for most of the tasks, the original studies did not even report OOD evaluations. We follow the same data split as baseline methods for fair comparison (see the details in Appendix C). BI-GIN+EPE achieves results comparable to, or better than, the baseline methods. BI-GPS+EPE achieves even better performance than BI-GIN+EPE for small graphs. Note that the baseline methods for certain tasks may incorporate domain-specific expert knowledge and additional data processing. For example, CKTGNN Dong et al. (2023) for the AMP dataset modifies the graph structures into DAGs and employs an asynchronized message passing to mimic the signal flow in these amplifiers; 'timer-GNN' Guo et al. (2022) is tailored for the TIME dataset to mimic the transmission rules of clock signals and designs a non-linear delay model (NLDM) along with a novel module 'cell library'. Such domain knowledge may further enhance BI-GPS+EPE and BI-GIN+EPE for these specific tasks, which is left for future research.

**Discussion on OOD Evaluation:** Despite BI-GPS-T+EPE and BI-GIN+EPE outperforming other methods in OOD testing across all tasks, we cannot yet conclude that these methods are sufficiently effective for practical OOD usage. *In fact, making accurate predictions with OOD data in hardware design remains a significant challenge.* When the graph structures in training sets are sufficiently diverse, such as in datasets with a large number of small graphs (e.g., AMP, HLS) or those with abundant local structures (e.g., SR), BI-GIN+EPE and BI-GPS-T+EPE tend to maintain reasonably good performance on OOD data. However, OOD generalization becomes challenging when the

diversity of graph structures in the training set is limited. For instance, in the TIME dataset, which has a limited variety of graph structures for training and OOD testing data with entirely different graph structures, both BI-GIN+EPE and BI-GPS-T+EPE perform worse than timer-GNN Guo et al. (2022), which integrates the knowledge of the physical structure of circuits (as shown in Table 21). We identify ensuring OOD performance, especially when training sets lack sufficiently diversified graph structures, as a key direction for future DGRL research.

## 7 CONCLUSIONS AND LIMITATIONS

Through benchmarking 21 methods on in-distribution and out-of-distribution test sets across 13 tasks and 5 datasets within the hardware design loop, we find bidirected (BI) message passing neural networks can substantially improve the performance of both Graph Transformer (GT) encoders that incorporate MPNN layers and pure GNN encoders. Positional Encodings (PEs), particularly when used stably, can broadly enhance the performance of both GTs and GNNs. With these insights, we identify two top-performing models: BI-GPS-T+EPE and BI-GIN+EPE, both of which outperform the baseline models originally proposed for the corresponding tasks.

**Limitations**: Although the benchmark covers multiple stages in hardware design loop, there are other tasks Mendis et al. (2019); Sỳkora et al. (2022); Xun et al. (2024); Chai et al. (2023); Chen et al. (2019); Alrahis et al. (2021b); Zhang et al. (2019a) that could be included in this benchmark as DGRL tasks. Given technological advancements and the diversity of design tools, ensuring OOD performance remains an urgent open problem in hardware design. Future research may involve high-quality data collection Jain et al. (2020); Gupta et al. (2021); Wu et al. (2021b); Whang et al. (2023); Wu et al. (2020) or the development of OOD-aware DGRL methods Liu et al. (2023); Shi et al. (2024); Liu et al. (2024); Liu & Ding (2024).

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

## A  MORE RELATED WORK

In this section, we review the extensive previous studies that use ML-based surrogate models.

ML-based surrogate models have been widely used in hardware system design, such as predicting energy/power consumption, latency, throughput, or reliability on CPUs Lo et al. (2015); Zheng et al. (2016); Mishra et al. (2018); Mendis et al. (2019); Lin et al. (2019); Chen et al. (2019); Sỳkora et al. (2022), GPUs Jia et al. (2012); Baldini et al. (2014); O'Neal et al. (2017); Pattnaik et al. (2016); Chen et al. (2018); Li et al. (2020a), tensor processing units (TPUs) Kaufman et al. (2021); Phothilimthana et al. (2023), and data centers Murray et al. (2005); Mahdisoltani et al. (2017); Xu et al. (2018). Similar trends are observed in quickly estimating quality-of-results of circuit designs in EDA flows, spanning high-level synthesis (HLS) Wang et al. (2022b); Zhao et al. (2019); Makrani et al. (2019); Ustun et al. (2020); Lin et al. (2020); Wu et al. (2022b; 2021a); Bai et al. (2023); Ye et al. (2024), logic synthesis Yu et al. (2018); Yu & Zhou (2020); Wu et al. (2022a; 2023); Zhao & Shamsi (2022), physical synthesis Tabrizi et al. (2018); Liang et al. (2020); Chen et al. (2020); Mirhoseini et al. (2021); Guo et al. (2022); Esmaeilzadeh et al. (2023); Lu et al. (2024; 2020), analog circuit designs Wang et al. (2020); Zhang et al. (2019a); Ren et al. (2020); Shook et al. (2020); Li et al. (2020c); Dong et al. (2023), and design verification Ma et al. (2019b); Hughes et al. (2019); Shibu et al. (2021); Vasudevan et al. (2021); Wu et al. (2024). *As circuits can naturally be represented as directed graphs, the adoption of GNN-based surrogate models is increasingly prominent.* We discuss several examples for each of the aforementioned tasks as follows.

In CPU throughput estimation, Granite Sỳkora et al. (2022) adopts a GNN model to predict basic block throughput on CPUs. Basic blocks are represented as graphs to capture the semantic relationships between instructions and registers. A GNN model is then trained to learn expressive embeddings for each basic block, followed by a decoder network to predict the throughput.

In HLS, many studies leverage the IR graphs generated by HLS front-ends. Ustun et al. Ustun et al. (2020) employs GNNs to predict the mapping from arithmetic operations in IR graphs to different resources on FPGAs. GNN-DSE Sohrabizadeh et al. (2022) also apply graph neural network to learn resource consumption mappings on HLS codes. Sohrabizadeh et al. (2023) propose Harp, a hierarchical abstract of the HLS graph, Qin et al. (2024) apply pre-trained language models with GNNs to conduct multi-modality prediction on the mapping. IronMan Wu et al. (2021a) exploits GNNs to generate graph embeddings of IR graphs, which serve as state representations in its reinforcement learning (RL-)based search engine to find the Pareto curve between two types of computing resources on FPGAs. The same problem can also be solved by carefully designing a GNN surrogate model as a continuous relaxation of the actual cost model, allowing for a soft solution that can be decoded into the final discrete solution of resource assignments Wang et al. (2022b). In terms of HLS datasets, Wu et al. Wu et al. (2022b) develop an HLS dataset and benchmark GNNs for predicting resource usage and timing, however, they enhance accuracy with domain-specific information and do not explore message passing directions or the benefit from positional encoding. Bai et al. Bai et al. (2023) contribute a new HLS dataset and combine pre-trained language models Wang et al. (2021); Guo et al. (2020) and GNNs to predict the optimization effects of different directives.

In logic synthesis or logic design, LOSTIN Wu et al. (2022a) employs a GNN to encode circuit graphs and an LSTM to encode logic synthesis sequences, where the two embeddings are concatenated to predict logic delay and area. To identify functional units from gate-level netlists, different GNN models can be leveraged to classify sub-circuit functionality Alrahis et al. (2021a), predict the functionality of approximate circuits Bücher et al. (2022), analyze impacts of circuit rewriting on functional operator detection Zhao & Shamsi (2022), and predict boundaries of arithmetic blocks He et al. (2021). Gamora Wu et al. (2023) leverages the message-passing mechanism in GNN computation to imitate structural shape hashing and functional propagation in conventional symbolic reasoning, achieving up to six orders of magnitude speedup compared to the logic synthesis tool ABC in extracting adder trees from multipliers.

In physical synthesis, Mirhoseini et al. Mirhoseini et al. (2021) combine GCN with deep RL to place macros (i.e., memory cells), after which standard cells are placed by a force-directed method. The GCN model encodes the topological information of chip netlists to generate graph embeddings as the inputs to the RL agent, as well as to provide proxy rewards to guide the search process. Lu et al. Lu et al. (2020) apply GraphSAGE Hamilton et al. (2017) to circuit netlists to learn node representations

that capture logical affinity. These representations are grouped by a weighted K-means clustering to provide placement guidance, informing the placer about which cells should be placed nearby in actual physical layouts. Guo et al. Guo et al. (2022) develop a hierarchical GNN with BI message passing to estimate post-routing timing behaviors by using circuit placement results.

In hardware design verification, test point insertion is a common technique aimed at enhancing fault coverage, which modifies target hardware designs by inserting extra control points or observation points. Ma et al. Ma et al. (2019b) use GCNs to predict whether a node in hardware designs is easy or hard to observe, based on which new observation points are inserted. To improve branch coverage, Vasudevan et al. Vasudevan et al. (2021) exploit IPA-GNN Bieber et al. (2020) to predict the probability of current test parameters covering specific cover points by characterizing RTL semantics and computation flows; new tests targeting uncovered points are generated by maximizing the predicted probability with respect to test parameters through gradient-based search.

In analog circuit design, by using circuit schematics, CktGNN Dong et al. (2023) employs a nested GNN to predict analog circuit properties (i.e., gain, BW, PM) and reconstruct circuit topology. By using pre-layout information, ParaGraph Ren et al. (2020) builds a GNN model to predict layout-dependent parasitics and physical device parameters; GCN-RL circuit designer Wang et al. (2020) combines RL with GCNs for automatic transistor sizing. By using layout information, GNN surrogate models can predict the relative placement quality of different designs Li et al. (2020c), and other circuit properties, such as the electromagnetic properties of high-frequency circuits Zhang et al. (2019a).

## B A Brief Review of Magnetic Laplacian and Positional Encodings for Directed Graphs

Positional encodings (PE) for graphs are vectorized representations that can effectively describe the global position of nodes (absolute PE) or relative position of node pairs (relative PE). They provide crucial positional information and thus benefits many backbone models that is position-agnostic. For instance, on undirected graphs, PE can provably alleviate the limited expressive power of Message Passing Neural Networks Xu et al. (2019); Morris et al. (2019); Li et al. (2020b); Lim et al. (2022); PE are also widely adopted in many graph transformers to incorporate positional information and break the identicalness of nodes in attention mechanism Kreuzer et al. (2021); Ying et al. (2021); Rampášek et al. (2022); Chen et al. (2022). As a result, the design and use of PE become one of the most important factors in building powerful graph encoders.

Likely, one can expect that direction-aware PE are also crucial when it comes to directed graph encoders. "Direction-aware" implies that PE should be able to capture the directedness of graphs. A notable example is Magnetic Laplacian PE Geisler et al. (2023), which adopts the eigenvectors of Magnetic Laplacian as PE. Note that Magnetic Laplacian can encode the directedness via the sign of phase of $\exp\{\pm i2\pi q\}$. Besides, when $q = 0$, Magnetic Laplacian reduces to normal symmetric Laplacian. Thus, Magnetic Laplacian PE for directed graphs can be seen as a generalization of Laplacian PE for undirected graphs, and the latter is known to enjoy many nice spectral properties Chung (1997) and be capable to capture many undirected graph distances Kreuzer et al. (2021). Therefore, Magnetic Laplacian appears to be a strong candidate for designing direction-aware PE. The definition is as follows:

Magnetic Laplacian (MagLap) matrix is a Hermitian complex matrix defined by $L_q = I - D^{-1/2} A_q D^{-1/2}$, where $D$ is the diagonalized degree matrix counting both in-degree and out-degree, and $A_q$ refers to the complex matrix as follows:

$$[A_q]_{u,v} = \begin{cases} \exp\{i2\pi q\}, & \text{if } (u,v) \in \mathcal{E}, \\ \exp\{-i2\pi q\}, & \text{if } (v,u) \in \mathcal{E}, \\ 1, & \text{if } (u,v), (v,u) \in \mathcal{E}, \end{cases} \tag{1}$$

with a parameter $q \in [0, 1)$ called potential. Hermitian refers to the property that complex conjugate $L_q^\dagger$ equals to $L_q$. It is also worth noticing that when $q = 0$, MagLap $L_{q=0}$ degenerates to the standard symmetric Laplacian matrix $L = I - D^{-1/2}(A + A^\top)D^{-1/2}$ as a special case, where $A$ is the Adjacency matrix. See Furutani et al. (2020) for a comprehensive introduction to Magnetic Laplacian.

Note that it is worth mentioning that there are also other PE for directed graphs, such as SVD of Adjacency matrix Hussain et al. (2022) and directed random walk Geisler et al. (2023).

## C    DATA SPLIT WHEN COMPARING WITH BASELINES IN THE ORIGINAL PAPERS

When comparing with the baselines from original papers, for training and testing the proposed new methods 'BI-GINE+EPE' and 'BI-GPS+EPE', we follow the dataset split of the original paper for fair comparison.

In the AMP dataset, we follow Dong et al. (2023) to merge the graphs with 2-stage and 3-stage Op-Amps together into one dataset, we take the last 1000 graphs for test and the rest for training and validation. The performance of baseline method cktGNN and the proposed new methods 'BI-GINE+EPE' and 'BI-GPS+EPE' are reported on such data split; for the HLS dataset, both the baseline method and the proposed new methods are trained and tested on control data flow graphs (CDFG) only, following the same data split ratio that randomly divide the data into training, validation and testing as described in the original paper Wu et al. (2022b); in the SR dataset, both the baseline and the new methods are trained with 24-bit netlists and tested on 48-bit netlists, note that both the training and testing data are obtained before technology mapping Wu et al. (2023); for the CG dataset both the baselines and the proposed methods are tested to predict the runtime of neural networks on the Cortex A76 CPU platform Zhang et al. (2021a); for the TIME dataset, we follow the dataset split in the original paper Guo et al. (2022) and compare the results of the baseline method and the new methods on the ID designs.

## D    DATASET SELECTION DETAILS

**License for the datasets and codes.**

|  | code implementation | dataset license |
| --- | --- | --- |
| HLS Wu et al. (2022b) | MIT License | MIT License |
| AMP Dong et al. (2023) | MIT License | MIT License |
| SR Wu et al. (2023) | The MIT License | The MIT License |
| CG Zhang et al. (2021a) | MIT License | MIT License |
| our benchmark | CC BY-NC | - - |

Table 7: License of the datasets and the toolbox implementation of this benchmark.

For detailed information of the license of each origin dataset, please refer to their original paper/documents, the final interpretation regarding the five dataset's licensing information rests with the owner of the original paper. To the best of our knowledge, these hardware datasets contain no personally identifiable information or offensive content.

### D.1    HIGH-LEVEL SYNTHESIS (HLS) DATASET

After HLS front-end compilation, six node features are extracted, as summarized in Table 8. Each edge has two features, the edge type represented in integers, and a binary value indicating whether this edge is a back edge. Each graph is labeled based on its post-implementation performance metrics, which are synthesized by Vitis HLS vit and implemented by Vivado viv. Three metrics are used for regression: DSP, LUT, and CP. The first two are integer numbers indicating the number of resources used in the final implementation; the last one is CP timing in fractional number, determining the maximum working frequency of FPGA. The DFG and CDFG datasets consists of 19,120 and 18,570 C programs, respectively. Figure 4 shows an example C program from the CDFG dataset, with the corresponding control dataflow graph shown in Figure 5. More information can be found in the original paper Wu et al. (2022b).

| Feature | Description | Values |
|---------|-------------|--------|
| Node type | General node type | `operation nodes`, `blocks`, `ports`, `misc` |
| Bitwidth | Bitwidth of the node | $0 \sim 256$, `misc` |
| Opcode type | Opcode categories based on LLVM | `binary_unary`, `bitwise`, `memory`, etc. |
| Opcode | Opcode of the node | `load`, `add`, `xor`, `icmp`, etc. |
| Is start of path | Whether the node is the starting node of a path | $0, 1$, `misc` |
| Cluster group | Cluster number of the node | $-1 \sim 256$, `misc` |

Table 8: Node features and their example values.

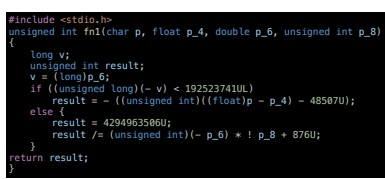

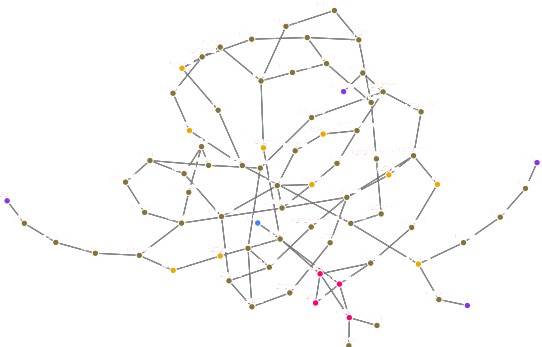

Figure 4: An example C program from the CDFG dataset.

Figure 5: Control dataflow graph of the example program in Figure 4.

## D.2 SYMBOLIC REASONING (SR) DATASET

In this dataset, all the circuit designs are represented as and-inverter graphs (AIGs), a concise and uniform representation of BNs consisting of inverters and two-input AND gates, which allows rewriting, simulation, technology mapping, placement, and verification to share the same data structure Mishchenko et al. (2006). In an AIG, each node has at most two incoming edges; a node without incoming edges is a primary input (PI); primary outputs (POs) are denoted by special output nodes; each internal node represents a two-input AND function. Based on De Morgan's laws, any combinational BN can be converted into an AIG Brayton & Mishchenko (2010) in a fast and scalable manner.

For each node, there are three node features represented in binary values denoting node types and Boolean functionality. The first node feature indicates whether this node is a PI/PO or intermediate node (i.e., AND gate). The second and the third node features indicate whether each input edge is inverted or not, such that AIGs can be represented as homogeneous graphs without additional edge features.

This dataset aims to leverage graph learning based approaches to accelerate the adder tree extraction in (integer) multiplier verification, which involves two reasoning steps Li et al. (2013); Subramanyan et al. (2013): (1) detecting XOR/MAJ functions to construct adders, and then (2) identifying their boundaries. Thus, there are two sets of node labels, i.e., two node-level classification tasks. One task provides labels specifying whether a node (i.e., a gate) in the AIG belongs to MAJ, XOR, or is shared by both MAJ and XOR. The other task provides labels specifying whether a node is the root node of an adder. These AIGs and ground truth labels are generated by the logic synthesis tool ABC Brayton & Mishchenko (2010). Figure 6 shows the AIG of an 8-bit multiplier: the blue and red nodes are the root nodes of XOR functions, with the red nodes directly connecting to the POs; the green nodes are the root nodes of MAJ functions. By pairing one XOR function with one MAJ function sharing the same set of inputs, we can extract the adder tree, which is shown in Figure 7. More information can be found in the original paper Wu et al. (2023).

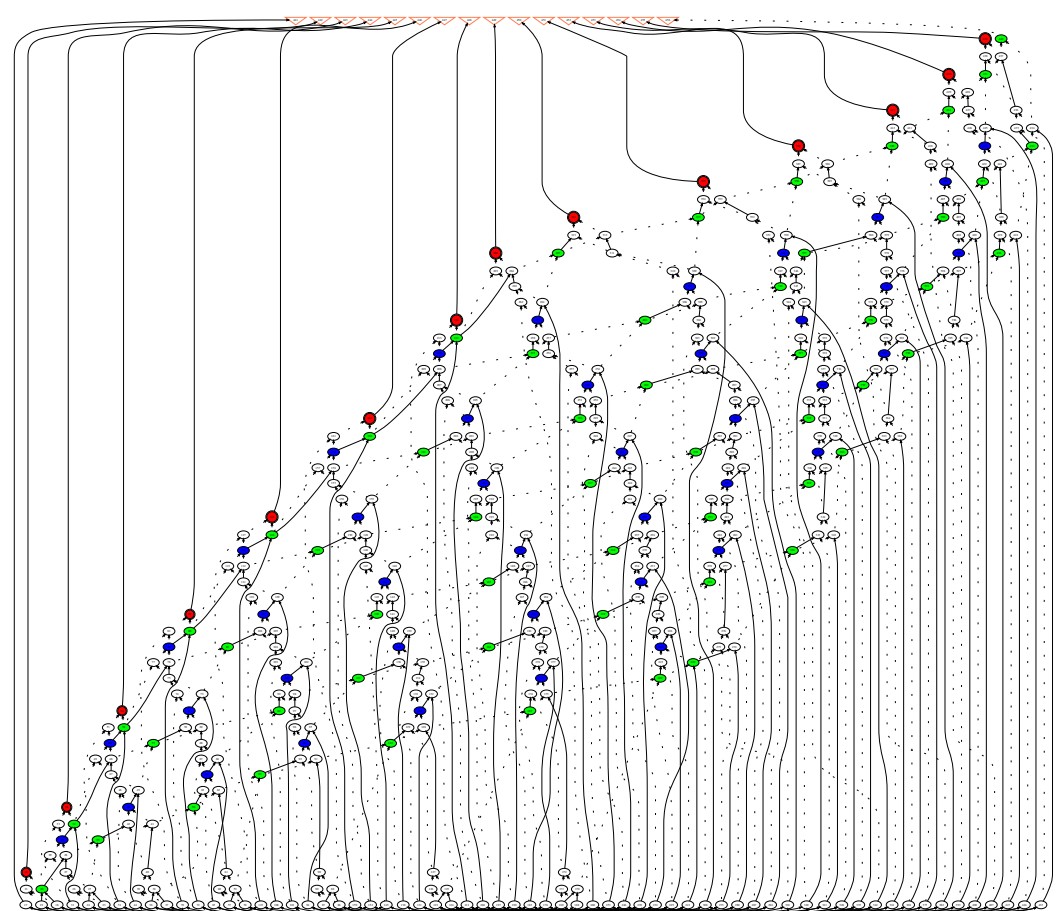

Figure 6: 8-bit multiplier in AIG.

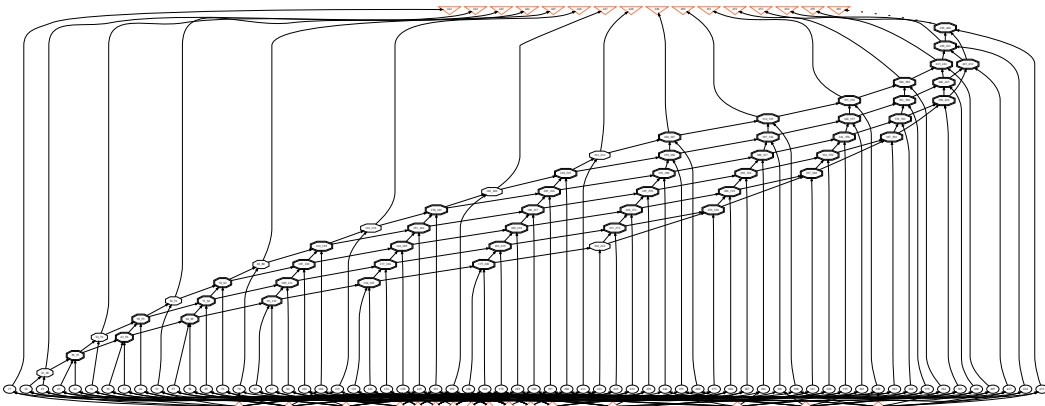

Figure 7: 8-bit multiplier with adders extracted.

### D.3 PRE-ROUTING TIMING PREDICTION (TIME) DATASET

Similar to timing analysis tools, circuits in this dataset are represented as heterogeneous graphs consisting of two types of edges: net edges and cell edges, with edge features shown in Table 9a. The nodes in graphs denote pins in circuits, with features summarized in Table 9b. The TIME dataset collects 21 real-world benchmark circuits from OpenCores ope (a) with OpenROAD ope (b) on SkyWater 130nm technology sky (i.e. blabla, usb_cdc_core, BM64, salsa20, aes128, aes192,

| Description | Size | Description | Size |
|---|---|---|---|
| (Net edge) Distances along x/y direction | 2 | Is primary I/O pin or not | 1 |
| (Cell edge) LUT is valid or no | 8 | Is fan-in or fan-out | 1 |
| (Cell) LUT indices | $8 \times (7 + 7)$ | Distance to the 4 die area boundaries | 4 |
| (Cell) LUT value matrices | $8 \times (7 \times 7)$ | Pin capacitance | 4 (EL/RF) |

(a) Edge features in the TIME dataset. For each cell edge, 8 LUTs are used to model cell delay and slew under four timing corner combinations (EL/RF).

(b) Pin (i.e., node) features in the TIME dataset. EL/RF stands for early/late and rise/fall, i.e., the four timing corner combinations in STA.

Table 9: Node and edge features for pre-routing timing prediction.

aes256, wbqspiflash, cic_decimator, des, aes_cipher, picorv32a, zipdiv, genericfir, usb, jpeg_encoder, usbf_device, xtea, spm, y_huff, and synth_ram). More information can be found in the original paper Guo et al. (2022).

We select the slack prediction task in this dataset, including setup slack and hold slack. Slack values are used by STA tools to identify paths that violate timing constraints, enabling further optimization of placement and routing. Setup/hold slack is defined as the difference between the required arrival time (based on setup or hold time) and the actual arrival time of data/signals at timing endpoints, making it a node-level regression task.

Figure 8 shows the most common timing path, register-to-register path. (1) For setup slack, the signal should arrive *earlier* than the required arrival time (i.e., clock period - setup time). Setup time $t_{\text{setup}}$ refers to the time before the clock edge that data must be stable. (2) For hold slack, the signal should arrive *later* than the required hold time to ensure no impact on signals for the current clock edge. Hold time $t_{\text{hold}}$ refers to the time after the clock edge that data must be stable.

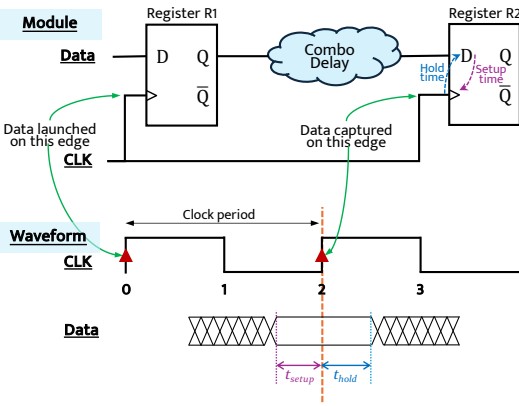

Figure 8: Register-to-register timing path.

### D.3.1 TIME DATASET DISTRIBUTION SHIFT DEFINITION

For training and ID testing, we take the designs 'blabla', 'usb_cdc_core', 'wbqspiflash', 'cic_decimator', 'picorv32a', 'zipdiv', 'usb'. For OOD testing, we use 'xtea', 'spm', 'y_huff', 'synth_ram'.

### D.4 COMPUTATIONAL GRAPH (CG) DATASET

This dataset includes (1) 12 state-of-the-art CNN models for the ImageNet2012 classification task (i.e., AlexNet, VGG, DenseNet, ResNet, SqueezeNet, GoogleNet, MobileNetv1, MobileNetv2, MobileNetv3, ShuffleNetv2, MnasNet, and ProxylessNas), each with 2,000 variants that differ in output channel number and kernel size per layer, and (2) 2,000 models from NASBench201 Dong & Yang (2019) with the highest test accuracy on CIFAR10, each featuring a unique set of edge connections. In total, this dataset contains 26,000 models with different operators and configurations. Figure 9 shows an example of the computational graph of a model in NASBench201.

Node features include input shape (5 dimensions), kernel/weight shape (padding to 4 dimensions), strides (2 dimensions), and output shape (5 dimensions). Each computational graph is labeled with the inference latency on three edge devices (i.e., Cortex A76 CPU, Adreno 630 GPU, Adreno 640 GPU). There is no edge feature in this dataset. More information can be found in the original paper Zhang et al. (2021a).

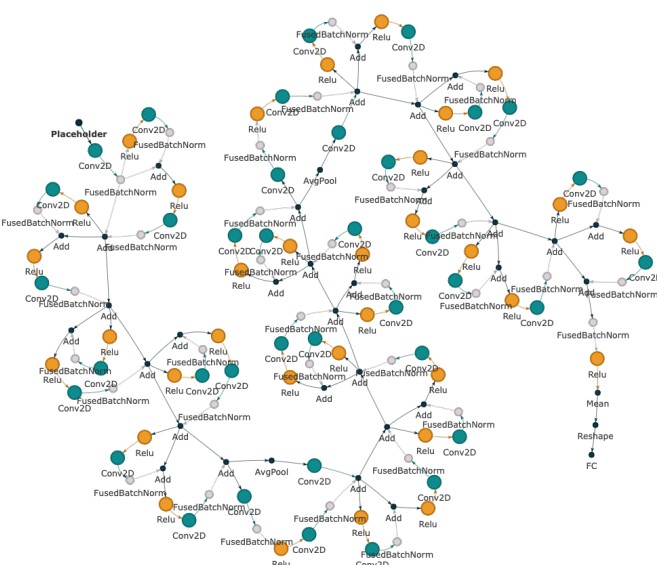

Figure 9: Computational graph of an example NN model from NASBench201 Dong & Yang (2019).

### D.4.1 CG Dataset Distribution Shift Definition

For training and ID testing, we take 'DenseNets', 'MnasNets', 'MobileNetv2s', 'MobileNetv3s', 'nasbench201s'. For OOD testing, we select 'Proxylessass', 'ResNets', and 'SqueezeNets'.

### D.5 Multi-Stage Amplifiers (AMP) Dataset

This dataset focuses on predicting circuit specifications (e.g., DC gain, bandwidth (BW), phase margin (PM)) of 2/3-stage operational amplifiers (Op-Amps), which are simulated by the circuit simulator Cadence Spectre spe. A 2/3-stage Op-Amp consists of (1) two/three single-stage Op-Amps on the main feedforwoard path and (2) several feedback paths, with one example shown in the right part of Figure 10. To make multi-stage Op-Amps more stable, feedforward and feedback paths are used to achieve different compensation schemes, each of which is implemented with a sub-circuit, e.g., single-stage Op-Amps, resistors, and capacitors. Due to the different topologies of single-stage Op-Amps and various compensation schemes, each sub-circuit is built as a subgraph. There are 24 potential sub-circuits in the considered 2/3-stage Op-Amps:

- Single R or C (① in Figure D.5, 2 types).
- R and C connected in parallel or serial (② in Figure D.5, 2 types).
- A single-stage Op-Amp ($g_m$) with different polarities (positive, $+g_m$, or negative, $-g_m$) and directions (feedforward or feedback) (③ in Figure D.5, 4 types).
- A single-stage Op-Amp ($g_m$) with R or C connected in parallel or serial (16 types). Note that we use the single-stage Op-Amp with feedforward direction and positive polarities as an example for ④ in Figure D.5.

Based on aforementioned formulation, node features include (1) subgraph type, (2) node type (e.g., R, C, $\pm g_m$ with feedforward/feedback, primary input/output), and (3) value of the component. There is no edge feature. More information can be found in the original paper Dong et al. (2023).

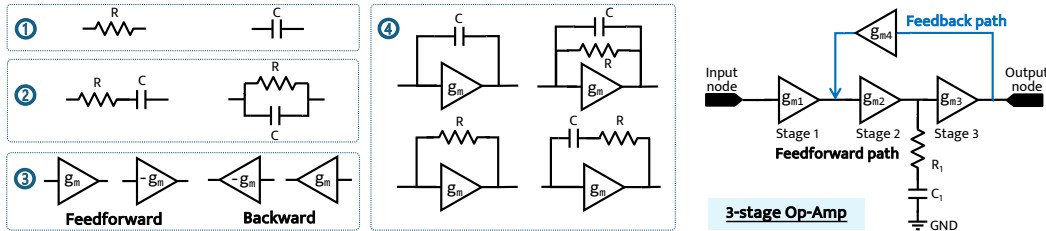

Figure 10: Subgraph basis for operational amplifiers and an example 3-stage Op-Amp.

# E BENCHMARK DESIGN DETAILS

## E.1 SELECTED BACKBONE FUNCTIONAL

Here we list the functions we implemented for the selected GNN backbone layers, note that here we show the forms of the backbone on undirected graphs, one may do slight modification by introducing $\omega(\cdot)$ on the neighbor message aggregation to consider message passing control for directed graphs.

$$\text{GIN}: \quad \mathbf{x}_i^{(k)} = \text{MLP}\left(\mathbf{x}_i^{(k-1)} + \sum_{j \in \mathcal{N}(i)} \mathbf{x}_j^{(k-1)}\right) \tag{2}$$

is the for graphs without edge features,

$$\text{GINE}: \quad \mathbf{x}_i^{(k)} = \text{MLP}\left(\mathbf{x}_i^{(k-1)} + \sum_{j \in \mathcal{N}(i)} \text{ReLU}(\mathbf{x}_j^{(k-1)} + \mathbf{e}_{j,i}^{(k-1)})\right) \tag{3}$$

is used for graphs with edge features.

$$\text{GCN}: \quad \mathbf{x}_i^{(k)} = \theta^\top \sum_{j \in \mathcal{N}(i) \cup \{i\}} \frac{\mathbf{e}_{j,i}^{(k-1)}}{\sqrt{\hat{d}_j \hat{d}_i}} \mathbf{x}_j^{(k-1)}, \tag{4}$$

where $\theta$ is the parameter to learn, for graphs with edge features $\mathbf{e}_{j,i}$ is the processed edge weight, for graphs without edge features $\mathbf{e}_{j,i}$ is set as 1.

$$\text{GAT}: \quad \mathbf{x}_i^{(k)} = \alpha_{i,i}^{(k-1)} \theta_s \mathbf{x}_i^{(k-1)} + \sum_{j \in \mathcal{N}(i)} \alpha_{i,j}^{(k-1)} \theta_t \mathbf{x}_j^{(k-1)}, \tag{5}$$

where $\theta_s, \theta_t$ are parameters to learn, for graphs without edge features,

$$\alpha_{i,j}^{(k-1)} = \frac{\exp\left(\text{LeakyReLU}\left(\mathbf{a}_s^\top \theta_s \mathbf{x}_i^{(k-1)} + \mathbf{a}_t^\top \theta_t \mathbf{x}_j^{(k-1)}\right)\right)}{\sum_{m \in \mathcal{N}(i) \cup \{i\}} \exp\left(\text{LeakyReLU}\left(\mathbf{a}_s^\top \theta_s \mathbf{x}_i^{(k-1)} + \mathbf{a}_t^\top \theta_t \mathbf{x}_m^{(k-1)}\right)\right)}, \tag{6}$$

and for graphs with edge features,

$$\alpha_{i,j}^{(k-1)} = \frac{\exp\left(\text{LeakyReLU}\left(\mathbf{a}_s^\top \theta_s \mathbf{x}_i^{(k-1)} + \mathbf{a}_t^\top \theta_t \mathbf{x}_j^{(k-1)} + \mathbf{a}_e^\top \theta_e \mathbf{e}_{i,j}^{(k-1)}\right)\right)}{\sum_{m \in \mathcal{N}(i) \cup \{i\}} \exp\left(\text{LeakyReLU}\left(\mathbf{a}_s^\top \theta_s \mathbf{x}_i^{(k-1)} + \mathbf{a}_t^\top \theta_t \mathbf{x}_m^{(k-1)} + \mathbf{a}_e^\top \theta_e \mathbf{e}_{i,m}^{(k-1)}\right)\right)}, \tag{7}$$

where $\mathbf{a}_s, \mathbf{a}_t$ are learnable parameterized attention parameters.

Each GPS backbone layer is implemented as follows:

$$\begin{aligned} \text{GPS}: \mathbf{X}_M^{(k)} &= \text{MPNN}^{(k-1)}(\mathbf{X}^{(k-1)}, \mathbf{E}^{(k-1)}) \\ \mathbf{X}_T^{(k)} &= \text{GlobalATTn}^{(k-1)}(\mathbf{X}^{(k-1)} \\ \mathbf{X}^{(k)} &= \text{MLP}(\mathbf{X}_M^{(k-1)} + \mathbf{X}_T^{(k-1)}), \end{aligned} \tag{8}$$

where $\mathbf{X}$, $\mathbf{E}$ denote node/edge features, we use GIN or GINE as the MPNN layer, and we use the transformer as the global attention reasoning layer.

For DGCN Tong et al. (2020b) and DiGCN Tong et al. (2020a), we follow the implementation in PyGSD He et al. (2024), please refer to `https://pytorch-geometric-signed-directed.readthedocs.io/en/latest/index.html` for backbone implementation details.

$$\text{MSGNN: } \mathbf{x}_i^{(k)} = \sigma \left( \sum_{i=1}^{F^{k-1}} \mathbf{Y}_{ij} \mathbf{x}_i^{(k-1)} + \mathbf{b}_j^{(k-1)} \right), \tag{9}$$

where $\sigma$ is a complex version of Rectified Linear Unit defined by:

$$\sigma(z) = \begin{cases} z & -\pi/2 \leq \arg(z) < \pi/2 \\ 0 & \text{otherwise,} \end{cases}$$

where $\arg(\cdot)$ is the complex argument of $z \in \mathbb{C}$, $F^{(k)}$ denotes the number of channels in the $k$-th layer, $\mathbf{b}$ is a bias vector with equal real and imaginary parts, $\mathbf{Y}$ denotes the convolution matrix defined in Equation.(4) and (5) in He et al. (2022).

### E.2 HYPER-PARAMETER SPACE

| | batch size | learning rate | dropout rate | hidden dimension* | # of GNN layers | # of MLP layers |
|---|---|---|---|---|---|---|
| DGCN | {64, 128, 256, 512, 1024} | [5e-4, 1e-2] | {0, 0.1, 0.2, 0.3} | [96, 336] | [3,8] | [2,5] |
| DiGCN | {64, 128, 256, 512, 1024} | [5e-4, 1e-2] | {0, 0.1, 0.2, 0.3} | [96, 336] | [3,8] | [2,5] |
| MagNet | {64, 128, 256, 512, 1024} | [5e-4, 1e-2] | {0, 0.1, 0.2, 0.3} | [96, 336] | [3,8] | [2,5] |
| GCN | {64, 128, 256, 512, 1024} | [1e-4, 1e-2] | {0, 0.1, 0.2, 0.3} | [96, 336] | [3,8] | [2,5] |
| GIN | {64, 128, 256, 512, 1024} | [1e-4, 1e-2] | {0, 0.1, 0.2, 0.3} | [96, 336] | [3,8] | [2,5] |
| GAT | {64, 128, 256, 512, 1024} | [1e-4, 1e-2] | {0, 0.1, 0.2, 0.3} | [96, 336] | [3,8] | [2,5] |
| GPS-T | {64, 128, 256} | [1e-4, 1e-2] | {0, 0.1, 0.2, 0.3} | [96, 288] | [3,6] | [2,5] |
| GPS-P | {32, 64, 128, 256, 512} | [1e-4, 1e-2] | {0, 0.1, 0.2, 0.3} | [96, 288] | [3,6] | [2,5] |

Table 10: Hyper-parameter space for each backbone. *:hidden dimension slightly vary in each task.

## F HARDWARE AND PLATFORM

All the experiments run on a server with an AMP EPYC 7763 64-Core Processor and 8 Nvidia RTX6000 GPU cards. The codes run on frameworks based on PyTorch Paszke et al. (2019), PyTorch Geometric Fey & Lenssen (2019), PyTorch Geometrc Signed and Directed He et al. (2024), RAY Liaw et al. (2018).

## G IMPLEMENTATION DETAILS OF EXPERIMENTS

### G.1 RANKING CALCULATION

In Table. 4 and Table. 3, we report the average ranking of different combination of methods w.r.t. per evaluation metrics for each task from each dataset. The calculation of the ranking can be expressed as:

$$\text{rank}_k^{t,D} = \frac{1}{M_D} \sum_{m=1}^{M_D} R_{t,m}^k, \tag{10}$$

where $R_{t,m}^k$ denotes the ranking of the DGRL method $k$ on task $t$ w.r.t. the $m$-th evaluation metric. $M_D$ denotes the number of tasks and metrics on dataset $D$.

For evaluation metric the larger the better, we adopt the ranking function from pandas pandas development team (2020) with parameter $ascending = Flase$ and $method = \text{`}max\text{'}$.

For evaluation metrics the smaller the better, we use $ascending = True$ and $method = \text{`}min\text{'}$.

# H DETAILED EXPERIMENT RESULTS

## H.1 MAIN RESULTS: IN-DISTRIBUTION AND OUT-OF-DISTRIBUTION PERFORMANCE

| Method | shared | | | | root | | | |
|---|---|---|---|---|---|---|---|---|
| | accuracy | precision | recall | f1 | accuracy | precision | recall | f1 |
| GCN | 0.879±0.013 | 0.669±0.141 | 0.653±0.125 | 0.620±0.119 | 0.882±0.005 | 0.860±0.045 | 0.811±0.131 | 0.773±0.091 |
| DI-GCN | 0.633±0.000 | 0.376±0.000 | 0.377±0.000 | 0.337±0.000 | 0.751±0.000 | 0.250±0.000 | 0.333±0.000 | 0.285±0.000 |
| BI-GCN | 0.992±0.005 | 0.816±0.123 | 0.820±0.122 | 0.818±0.123 | 0.999±0.001 | 0.999±0.000 | 0.998±0.003 | 0.999±0.001 |
| GIN | 0.882±0.046 | 0.787±0.132 | 0.704±0.191 | 0.683±0.179 | 0.909±0.005 | 0.900±0.055 | 0.877±0.101 | 0.850±0.050 |
| DI-GIN | 0.999±0.000 | 0.749±0.000 | 0.749±0.000 | 0.749±0.000 | 0.999±0.000 | 0.999±0.000 | 0.999±0.000 | 0.999±0.000 |
| BI-GIN | 0.999±0.000 | 0.937±0.088 | 0.974±0.079 | 0.949±0.080 | 0.999±0.000 | 0.999±0.000 | 0.999±0.000 | 0.999±0.000 |
| GAT | 0.881±0.003 | 0.626±0.074 | 0.571±0.083 | 0.539±0.073 | 0.877±0.007 | 0.821±0.074 | 0.825±0.126 | 0.786±0.090 |
| DI-GAT | 0.885±0.060 | 0.679±0.033 | 0.682±0.034 | 0.674±0.036 | 0.981±0.005 | 0.985±0.009 | 0.956±0.012 | 0.970±0.008 |
| BI-GAT | 0.984±0.034 | 0.941±0.108 | 0.939±0.108 | 0.940±0.108 | 0.998±0.002 | 0.996±0.006 | 0.998±0.003 | 0.997±0.004 |
| GPS-P | 0.895±0.002 | 0.899±0.031 | 0.845±0.066 | 0.829±0.051 | 0.893±0.002 | 0.888±0.047 | 0.837±0.121 | 0.806±0.071 |
| DI-GPS-P | 0.999±0.000 | 0.749±0.000 | 0.749±0.000 | 0.749±0.000 | 0.999±0.000 | 0.999±0.000 | 0.999±0.000 | 0.999±0.000 |
| BI-GPS-P | 0.997±0.002 | 0.747±0.003 | 0.748±0.001 | 0.748±0.002 | 0.994±0.003 | 0.997±0.001 | 0.986±0.008 | 0.991±0.005 |
| DGCN | 0.975±0.000 | 0.734±0.000 | 0.730±0.000 | 0.732±0.000 | 0.991±0.000 | 0.989±0.003 | 0.984±0.003 | 0.987±0.000 |
| DiGCN | 0.995±0.000 | 0.747±0.000 | 0.747±0.000 | 0.747±0.000 | 0.994±0.001 | 0.990±0.007 | 0.991±0.004 | 0.991±0.004 |
| MagNet | 0.999±0.000 | 1.000±0.000 | 1.000±0.000 | 1.000±0.000 | 0.999±0.000 | 1.000±0.000 | 1.000±0.000 | 1.000±0.000 |

Table 11: ID performance on the SR dataset.

| Method | gain | | PM | | BW | |
|---|---|---|---|---|---|---|
| | mse | rmse | mse | rmse | mse | rmse |
| GCN | 1.262±1.682 | 0.993±0.554 | 13.598±30.906 | 2.618±2.736 | 35.230±0.657 | 5.935±0.055 |
| DI-GCN | 0.337±0.002 | 0.580±0.002 | 1.243±0.014 | 1.115±0.006 | 36.302±0.245 | 6.025±0.020 |
| BI-GCN | 0.148±0.004 | 0.385±0.005 | 63.874±194.025 | 3.862±7.375 | 22.947±1.022 | 4.789±0.105 |
| GIN | 0.166±0.029 | 0.406±0.034 | 1.266±0.034 | 1.125±0.015 | 28.259±10.648 | 5.244±0.894 |
| DI-GIN | 0.200±0.136 | 0.433±0.117 | 1.300±0.024 | 1.140±0.010 | 24.091±1.270 | 4.906±0.130 |
| BI-GIN | 0.137±0.012 | 0.370±0.016 | 1.251±0.035 | 1.118±0.016 | 19.724±1.489 | 4.438±0.170 |
| GAT | 0.158±0.008 | 0.397±0.010 | 865.339±2684.901 | 11.663±28.466 | 22.770±1.045 | 4.770±0.111 |
| DI-GAT | 0.205±0.006 | 0.453±0.007 | 1.562±0.840 | 1.223±0.268 | 26.855±2.928 | 5.175±0.284 |
| BI-GAT | 0.138±0.007 | 0.372±0.010 | 1.213±0.055 | 1.101±0.024 | 30.333±13.386 | 5.409±1.088 |
| GPS-T | 0.405±0.022 | 0.636±0.017 | 1.277±0.072 | 1.129±0.031 | 16.758±0.754 | 4.092±0.093 |
| DI-GPS-T | 0.122±0.009 | 0.349±0.013 | 1.259±0.044 | 1.121±0.019 | 16.600±0.877 | 4.073±0.107 |
| BI-GPS-T | 0.122±0.007 | 0.349±0.010 | 1.212±0.058 | 1.100±0.026 | 20.475±8.853 | 4.456±0.825 |
| DGCN | 0.567±0.004 | 0.753±0.003 | 1.292±0.000 | 1.136±0.000 | 54.256±0.257 | 7.365±0.017 |
| DiGCN | 0.367±0.009 | 0.606±0.007 | 1.294±0.011 | 1.137±0.005 | 52.375±0.276 | 7.237±0.019 |
| MagNet | 0.185±0.008 | 0.431±0.009 | 1.315±0.082 | 1.146±0.035 | 24.800±2.834 | 4.972±0.283 |

Table 12: ID performance on the AMP dataset.

| Method | DSP | | LUT | | CP | |
|---|---|---|---|---|---|---|
| | mse | R2 | mse | R2 | mse | R2 |
| GCN | 12.700±0.324 | 0.877±0.004 | 4.909±0.123 | 0.647±0.021 | 0.713±0.037 | 0.829±0.012 |
| DI-GCN | 12.591±0.312 | 0.877±0.003 | 4.998±0.114 | 0.643±0.013 | 0.692±0.013 | 0.837±0.004 |
| BI-GCN | 10.285±0.336 | 0.902±0.004 | 4.311±0.149 | 0.732±0.010 | 0.665±0.025 | 0.847±0.007 |
| GINE | 2.707±0.133 | 0.975±0.001 | 2.172±0.108 | 0.861±0.008 | 0.653±0.014 | 0.849±0.003 |
| DI-GINE | 2.312±0.172 | 0.979±0.001 | 2.145±0.158 | 0.863±0.011 | 0.645±0.022 | 0.851±0.007 |
| BI-GINE | 2.137±0.076 | 0.981±0.000 | 1.759±0.087 | 0.892±0.005 | 0.629±0.020 | 0.855±0.005 |
| GAT | 4.680±0.264 | 0.957±0.002 | 3.267±0.142 | 0.778±0.011 | 0.643±0.012 | 0.850±0.004 |
| DI-GAT | 7.697±0.238 | 0.926±0.002 | 4.188±0.226 | 0.685±0.031 | 0.677±0.051 | 0.840±0.014 |
| BI-GAT | 4.718±0.532 | 0.957±0.004 | 3.028±0.143 | 0.801±0.016 | 0.590±0.011 | 0.863±0.006 |
| GPS | 2.444±0.207 | 0.978±0.002 | 2.114±0.153 | 0.872±0.011 | 0.621±0.028 | 0.858±0.010 |
| DI-GPS | 2.517±0.180 | 0.977±0.001 | 2.306±0.224 | 0.862±0.015 | 0.625±0.028 | 0.856±0.007 |
| BI-GPS | 2.442±0.303 | 0.979±0.002 | 2.112±0.216 | 0.873±0.014 | 0.621±0.018 | 0.859±0.009 |
| DGCN | 19.614±1.151 | 0.816±0.010 | 7.988±2.512 | 0.333±0.256 | 1.127±0.049 | 0.706±0.014 |
| DiGCN | 12.125±0.204 | 0.885±0.003 | 5.683±0.638 | 0.527±0.092 | 0.704±0.019 | 0.836±0.004 |
| MagNet | 4.375±0.452 | 0.961±0.003 | 2.381±0.175 | 0.848±0.015 | 0.684±0.045 | 0.843±0.014 |

Table 13: ID performance on the HLS dataset.

| Method | shared | | | | root | | | |
|---|---|---|---|---|---|---|---|---|
| | accuracy | precision | recall | f1 | accuracy | precision | recall | f1 |
| GCN | 0.510±0.021 | 0.283±0.014 | 0.328±0.018 | 0.292±0.016 | 0.569±0.056 | 0.364±0.069 | 0.317±0.038 | 0.321±0.038 |
| DI-GCN | 0.553±0.000 | 0.318±0.000 | 0.374±0.000 | 0.301±0.000 | 0.741±0.000 | 0.247±0.000 | 0.333±0.000 | 0.283±0.000 |
| BI-GCN | 0.671±0.045 | 0.489±0.035 | 0.563±0.045 | 0.513±0.037 | 0.651±0.054 | 0.512±0.057 | 0.514±0.064 | 0.505±0.062 |
| GIN | 0.677±0.039 | 0.414±0.034 | 0.395±0.064 | 0.371±0.067 | 0.743±0.008 | 0.646±0.084 | 0.378±0.027 | 0.367±0.045 |
| DI-GIN | 0.621±0.027 | 0.449±0.047 | 0.462±0.024 | 0.441±0.026 | 0.729±0.010 | 0.475±0.047 | 0.390±0.079 | 0.371±0.082 |
| BI-GIN | 0.712±0.024 | 0.514±0.040 | 0.561±0.071 | 0.500±0.049 | 0.773±0.018 | 0.592±0.070 | 0.502±0.051 | 0.515±0.054 |
| GAT | 0.583±0.118 | 0.341±0.062 | 0.383±0.071 | 0.343±0.079 | 0.633±0.088 | 0.456±0.065 | 0.435±0.043 | 0.408±0.055 |
| DI-GAT | 0.547±0.061 | 0.384±0.032 | 0.453±0.072 | 0.394±0.038 | 0.642±0.041 | 0.431±0.062 | 0.474±0.061 | 0.437±0.058 |
| BI-GAT | 0.554±0.078 | 0.416±0.069 | 0.450±0.080 | 0.397±0.072 | 0.632±0.046 | 0.483±0.066 | 0.477±0.065 | 0.461±0.061 |
| GPS-P | 0.670±0.059 | 0.407±0.055 | 0.454±0.056 | 0.404±0.047 | 0.659±0.040 | 0.395±0.056 | 0.410±0.045 | 0.385±0.037 |
| DI-GPS-P | 0.651±0.059 | 0.421±0.053 | 0.499±0.056 | 0.441±0.050 | 0.665±0.077 | 0.441±0.059 | 0.509±0.088 | 0.450±0.065 |
| BI-GPS-P | 0.651±0.057 | 0.417±0.040 | 0.482±0.040 | 0.435±0.045 | 0.671±0.023 | 0.491±0.020 | 0.578±0.049 | 0.507±0.029 |
| DGCN | 0.628±0.009 | 0.497±0.028 | 0.391±0.043 | 0.400±0.045 | 0.709±0.020 | 0.511±0.035 | 0.495±0.024 | 0.498±0.028 |
| DiGCN | 0.470±0.094 | 0.432±0.031 | 0.493±0.040 | 0.396±0.056 | 0.726±0.016 | 0.505±0.023 | 0.501±0.013 | 0.497±0.017 |
| MagNet | 0.703±0.040 | 0.445±0.047 | 0.499±0.052 | 0.463±0.046 | 0.683±0.032 | 0.457±0.057 | 0.413±0.045 | 0.413±0.043 |

Table 14: OOD performance on the SR dataset.

| Method | gain | | pm | | bw | |
|---|---|---|---|---|---|---|
| | mse | rmse | mse | rmse | mse | rmse |
| GCN | 0.877±1.063 | 0.856±0.399 | 101.444±308.123 | 4.823±9.320 | 42.921±0.993 | 6.551±0.075 |
| DI-GCN | 0.451±0.015 | 0.671±0.011 | 1.412±0.012 | 1.188±0.005 | 46.374±0.590 | 6.809±0.043 |
| BI-GCN | 0.270±0.036 | 0.519±0.033 | 21.490±58.748 | 2.861±3.844 | 30.004±2.192 | 5.474±0.201 |
| GIN | 0.337±0.041 | 0.580±0.034 | 1.436±0.053 | 1.198±0.022 | 34.398±11.114 | 5.806±0.852 |
| DI-GIN | 0.356±0.071 | 0.594±0.055 | 1.379±0.015 | 1.174±0.006 | 44.154±12.707 | 6.584±0.941 |
| BI-GIN | 0.293±0.026 | 0.541±0.024 | 1.419±0.046 | 1.191±0.019 | 25.822±1.977 | 5.078±0.190 |
| GAT | 0.330±0.032 | 0.574±0.027 | 567.911±1750.531 | 9.859±22.869 | 30.155±1.879 | 5.489±0.169 |
| DI-GAT | 0.412±0.035 | 0.641±0.027 | 1.406±0.078 | 1.185±0.032 | 41.750±10.476 | 6.421±0.753 |
| BI-GAT | 0.198±0.012 | 0.445±0.014 | 1.348±0.052 | 1.160±0.022 | 41.008±13.522 | 6.336±0.976 |
| GPS-T | 0.508±0.059 | 0.712±0.041 | 1.415±0.061 | 1.189±0.025 | 21.815±1.973 | 4.666±0.206 |
| DI-GPS-T | 0.301±0.012 | 0.549±0.011 | 1.447±0.037 | 1.202±0.015 | 22.161±1.355 | 4.705±0.144 |
| BI-GPS-T | 0.314±0.030 | 0.560±0.027 | 1.315±0.050 | 1.146±0.021 | 26.607±10.277 | 5.087±0.897 |
| DGCN | 0.772±0.046 | 0.878±0.025 | 1.364±0.000 | 1.168±0.000 | 69.019±1.345 | 8.307±0.081 |
| DiGCN | 0.345±0.014 | 0.587±0.012 | 1.377±0.021 | 1.173±0.009 | 59.337±0.659 | 7.703±0.042 |
| MagNet | 0.285±0.065 | 0.531±0.059 | 1.608±0.127 | 1.267±0.051 | 36.505±2.749 | 6.038±0.227 |

Table 15: OOD performance on the AMP dataset.

| Method | DSP | | LUT | | CP | |
|---|---|---|---|---|---|---|
| | mse | R2 | mse | R2 | mse | R2 |
| GCN | 10.817±0.284 | 0.878±0.006 | 0.440±0.084 | 0.879±0.022 | 0.559±0.033 | 0.784±0.014 |
| DI-GCN | 11.101±0.468 | 0.873±0.006 | 0.454±0.128 | 0.863±0.053 | 0.593±0.009 | 0.774±0.006 |
| BI-GCN | 9.996±0.319 | 0.884±0.007 | 0.627±0.158 | 0.857±0.026 | 0.514±0.019 | 0.803±0.013 |
| GINE | 3.720±0.154 | 0.961±0.001 | 0.129±0.042 | 0.966±0.009 | 0.491±0.017 | 0.809±0.008 |
| DI-GINE | 3.194±0.122 | 0.967±0.001 | 0.118±0.022 | 0.968±0.005 | 0.513±0.025 | 0.804±0.015 |
| BI-GINE | 3.244±0.102 | 0.966±0.001 | 0.110±0.029 | 0.971±0.007 | 0.476±0.027 | 0.815±0.011 |
| GAT | 5.440±0.271 | 0.941±0.003 | 0.368±0.054 | 0.900±0.015 | 0.496±0.019 | 0.809±0.007 |
| DI-GAT | 8.927±0.355 | 0.895±0.005 | 0.456±0.135 | 0.856±0.040 | 0.551±0.034 | 0.784±0.015 |
| BI-GAT | 5.418±0.305 | 0.942±0.003 | 0.218±0.054 | 0.938±0.023 | 0.466±0.026 | 0.821±0.013 |
| GPS | 3.343±0.147 | 0.966±0.001 | 0.145±0.026 | 0.962±0.007 | 0.543±0.124 | 0.793±0.024 |
| DI-GPS | 3.210±0.146 | 0.967±0.001 | 0.139±0.032 | 0.964±0.008 | 0.461±0.016 | 0.820±0.014 |
| BI-GPS | 3.209±0.263 | 0.967±0.001 | 0.133±0.027 | 0.968±0.006 | 0.496±0.017 | 0.812±0.016 |
| DGCN | 20.220±1.474 | 0.756±0.024 | 0.647±0.228 | 0.797±0.095 | 1.159±0.093 | 0.519±0.028 |
| DiGCN | 10.922±0.353 | 0.880±0.004 | 0.665±0.257 | 0.758±0.098 | 0.613±0.022 | 0.770±0.011 |
| MagNet | 5.048±0.499 | 0.947±0.004 | 0.168±0.049 | 0.955±0.012 | 0.557±0.053 | 0.793±0.024 |

Table 16: OOD performance on the HLS dataset.

| Method | blabla | usb_cdc_core | wbqspiflash | cic_decimator | picorv32a | zipdiv | usb | average |
|---|---|---|---|---|---|---|---|---|
| GCN | 9.263±1.994 | 2.888±0.576 | 0.382±0.266 | 0.684±0.151 | 8.796±1.773 | 8.796±1.773 | 0.501±0.076 | 3.278±0.318 |
| DI-GCN | 13.008±5.156 | 2.764±0.347 | 0.504±0.245 | 0.594±0.293 | 10.166±2.557 | 10.166±2.557 | 0.622±0.087 | 4.058±0.892 |
| BI-GCN | 0.731±0.143 | 1.308±0.559 | 0.091±0.044 | 0.069±0.027 | 1.447±0.198 | 1.447±0.198 | 0.093±0.058 | 0.575±0.108 |
| GINE | 0.183±0.023 | 0.073±0.014 | 0.027±0.005 | 0.009±0.003 | 0.549±0.033 | 0.549±0.033 | 0.024±0.005 | 0.125±0.008 |
| DI-GINE | 1.133±0.156 | 0.166±0.041 | 0.147±0.041 | 0.048±0.024 | 1.361±0.398 | 1.361±0.398 | 0.123±0.051 | 0.439±0.048 |
| BI-GINE | 0.142±0.014 | 0.111±0.083 | 0.023±0.006 | 0.010±0.005 | 0.574±0.041 | 0.574±0.041 | 0.032±0.013 | 0.132±0.012 |
| GAT | 17.032±16.149 | 3.128±0.521 | 0.477±0.603 | 0.393±0.219 | 6.910±3.546 | 6.910±3.546 | 0.448±0.296 | 4.204±3.054 |
| DI-GAT | 61.647±17.523 | 3.359±0.563 | 1.728±0.436 | 0.756±0.270 | 14.218±3.930 | 14.218±3.930 | 0.746±0.155 | 12.295±3.179 |
| BI-GAT | 1.311±0.435 | 1.291±0.466 | 0.213±0.104 | 0.221±0.059 | 2.065±0.507 | 2.065±0.507 | 0.258±0.078 | 0.787±0.112 |
| GPS-P | 0.415±0.091 | 1.846±0.170 | 0.204±0.055 | 0.140±0.064 | 2.308±0.084 | 2.308±0.084 | 0.078±0.017 | 0.772±0.036 |
| DI-GPS-P | 0.334±0.091 | 0.283±0.110 | 0.160±0.049 | 0.105±0.029 | 1.973±0.093 | 1.973±0.093 | 0.051±0.011 | 0.440±0.028 |
| BI-GPS-P | 3.469±0.531 | 1.786±0.606 | 0.389±0.251 | 0.283±0.412 | 3.908±0.344 | 3.908±0.344 | 0.172±0.169 | 1.544±0.294 |
| DGCN | 104.383±0.123 | 4.278±0.152 | 2.314±0.032 | 1.262±0.045 | 19.884±0.203 | 19.884±0.203 | 1.586±0.073 | 19.933±0.049 |
| DiGCN | 47.858±14.097 | 4.134±0.429 | 1.725±0.426 | 1.011±0.324 | 13.460±1.900 | 13.460±1.900 | 1.086±0.306 | 10.547±2.441 |
| MagNet | 12.326±5.494 | 3.738±0.707 | 0.576±0.159 | 0.569±0.302 | 5.715±1.130 | 5.715±1.130 | 0.539±0.239 | 3.459±0.782 |

Table 17: ID performance with 'mse' metric on the Time dataset to predict hold slack.

| Method | blabla | usb_cdc_core | wbqspiflash | cic_decimator | picorv32a | zipdiv | usb | average |
|---|---|---|---|---|---|---|---|---|
| GCN | 0.910±0.019 | -0.312±0.262 | 0.826±0.120 | 0.358±0.142 | 0.460±0.108 | 0.905±0.029 | 0.539±0.070 | 0.526±0.045 |
| DI-GCN | 0.873±0.050 | -0.255±0.157 | 0.771±0.110 | 0.442±0.275 | 0.376±0.156 | 0.835±0.070 | 0.428±0.079 | 0.496±0.056 |
| BI-GCN | 0.992±0.001 | 0.405±0.254 | 0.958±0.020 | 0.935±0.025 | 0.911±0.012 | 0.937±0.011 | 0.914±0.053 | 0.865±0.047 |
| GINE | 0.998±0.000 | 0.966±0.006 | 0.987±0.002 | 0.991±0.003 | 0.966±0.002 | 0.997±0.001 | 0.977±0.004 | 0.983±0.001 |
| DI-GINE | 0.988±0.001 | 0.924±0.018 | 0.933±0.018 | 0.954±0.022 | 0.916±0.024 | 0.978±0.002 | 0.886±0.047 | 0.940±0.007 |
| BI-GINE | 0.998±0.000 | 0.949±0.037 | 0.989±0.002 | 0.990±0.005 | 0.964±0.002 | 0.993±0.005 | 0.970±0.012 | 0.979±0.006 |
| GAT | 0.834±0.156 | -0.421±0.236 | 0.784±0.272 | 0.630±0.205 | 0.575±0.217 | 0.772±0.230 | 0.588±0.271 | 0.537±0.168 |
| DI-GAT | 0.401±0.170 | -0.526±0.255 | 0.218±0.197 | 0.290±0.253 | 0.127±0.241 | 0.209±0.165 | 0.314±0.142 | 0.147±0.143 |
| BI-GAT | 0.987±0.004 | 0.413±0.212 | 0.903±0.047 | 0.792±0.055 | 0.873±0.031 | 0.966±0.017 | 0.762±0.072 | 0.814±0.033 |
| GPS-P | 0.996±0.000 | 0.020±0.090 | 0.881±0.032 | 0.907±0.042 | 0.839±0.005 | 0.903±0.024 | 0.925±0.016 | 0.782±0.014 |
| DI-GPS-P | 0.996±0.000 | 0.856±0.055 | 0.941±0.018 | 0.881±0.033 | 0.857±0.006 | 0.951±0.028 | 0.927±0.016 | 0.916±0.017 |
| BI-GPS-P | 0.968±0.004 | 0.052±0.321 | 0.773±0.146 | 0.814±0.270 | 0.727±0.023 | 0.812±0.178 | 0.837±0.159 | 0.712±0.136 |
| DGCN | -0.013±0.001 | -0.943±0.069 | -0.046±0.014 | -0.183±0.042 | -0.220±0.012 | -0.276±0.026 | -0.455±0.067 | -0.305±0.029 |
| DiGCN | 0.535±0.136 | -0.877±0.195 | 0.219±0.192 | 0.050±0.304 | 0.173±0.116 | 0.002±0.239 | 0.003±0.280 | 0.015±0.169 |
| MagNet | 0.880±0.053 | -0.698±0.321 | 0.739±0.071 | 0.465±0.283 | 0.649±0.069 | 0.836±0.080 | 0.505±0.220 | 0.482±0.055 |
| Timer-inspired GNN | 0.9616 | 0.9751 | 0.9721 | 0.9840 | 0.9688 | 0.9753 | 0.9784 | 0.9736 |

Table 18: ID performance with 'R2' metric on the Time dataset to predict hold slack.

| Method | blabla | usb_cdc_core | wbqspiflash | cic_decimator | picorv32a | zipdiv | usb | average |
|---|---|---|---|---|---|---|---|---|
| GCN | 127.555±9.301 | 17.551±1.646 | 34.125±1.436 | 1.399±0.128 | 126.288±10.077 | 28.960±1.970 | 17.849±1.509 | 50.532±1.053 |
| DI-GCN | 99.699±13.849 | 18.302±5.510 | 30.164±2.874 | 2.984±4.527 | 135.526±16.281 | 26.859±5.355 | 15.223±4.432 | 46.965±2.204 |
| BI-GCN | 104.062±11.187 | 12.867±1.746 | 27.941±1.197 | 4.693±0.572 | 102.680±9.184 | 20.688±1.020 | 6.033±2.329 | 39.852±2.405 |
| GINE | 24.911±9.848 | 2.819±0.801 | 8.012±3.209 | 1.233±0.360 | 31.441±2.463 | 3.379±1.141 | 2.211±0.517 | 10.572±2.027 |
| DI-GINE | 25.642±6.938 | 2.207±0.586 | 7.105±2.835 | 1.823±0.632 | 21.690±1.080 | 4.088±1.363 | 1.785±0.315 | 9.191±1.564 |
| BI-GINE | 21.108±6.326 | 1.690±0.510 | 4.484±1.648 | 1.079±0.778 | 23.207±5.134 | 3.160±2.036 | 2.695±2.925 | 8.203±1.720 |
| GAT | 105.449±39.675 | 13.554±3.935 | 32.138±2.692 | 3.416±1.475 | 116.730±16.637 | 25.274±2.653 | 7.188±2.344 | 43.393±3.497 |
| DI-GAT | 94.262±14.843 | 14.390±2.879 | 44.365±4.951 | 2.265±1.418 | 94.712±6.261 | 25.569±3.563 | 7.875±3.932 | 40.491±1.637 |
| BI-GAT | 68.198±12.533 | 14.929±2.209 | 21.334±2.564 | 3.263±0.779 | 76.571±9.852 | 19.570±1.516 | 4.016±1.470 | 29.697±2.690 |
| GPS-P | 71.180±7.296 | 6.017±0.951 | 19.114±3.978 | 2.786±0.931 | 40.219±2.599 | 21.280±5.284 | 1.974±0.394 | 23.224±2.482 |
| DI-GPS-P | 76.600±6.188 | 6.559±2.542 | 22.725±1.503 | 15.529±9.577 | 39.556±2.732 | 23.995±3.420 | 4.789±1.457 | 27.107±0.881 |
| BI-GPS-P | 79.395±9.864 | 6.205±1.437 | 21.067±5.043 | 4.882±1.370 | 39.426±3.651 | 26.285±4.810 | 3.748±0.912 | 25.858±2.168 |
| DGCN | 84.413±11.949 | 5.364±1.023 | 21.575±3.178 | 4.393±2.014 | 40.810±3.904 | 27.617±4.680 | 3.125±1.675 | 26.757±1.564 |
| DiGCN | 1790.553±1159.661 | 68.846±51.069 | 86.761±47.732 | 33.962±17.288 | 546.052±196.931 | 73.563±40.920 | 52.554±32.869 | 378.899±320.555 |
| MagNet | 106.479±18.775 | 7.917±0.623 | 25.831±1.098 | 11.474±0.993 | 131.028±12.043 | 15.074±0.966 | 4.877±0.662 | 43.240±2.216 |

Table 19: ID performance with 'MSE' metric on the Time dataset to predict setup slack.

| Method | blabla | usb_cdc_core | wbqspiflash | cic_decimator | picorv32a | zipdiv | usb | average |
|---|---|---|---|---|---|---|---|---|
| GCN | -1.030±0.148 | -4.397±0.506 | -0.219±0.051 | 0.180±0.075 | -2.489±0.278 | -0.971±0.134 | -4.437±0.459 | -1.909±0.150 |
| DI-GCN | -0.586±0.220 | -4.628±1.694 | -0.077±0.102 | -0.747±2.650 | -2.744±0.449 | | -3.637±1.350 | -1.892±0.254 |
| BI-GCN | -0.656±0.178 | -2.956±0.537 | 0.001±0.042 | -1.747±0.335 | -1.837±0.253 | -0.408±0.069 | -0.837±0.709 | -1.206±0.144 |
| GINE | 0.603±0.156 | 0.132±0.246 | 0.713±0.108 | 0.278±0.211 | 0.131±0.068 | 0.769±0.077 | 0.326±0.157 | 0.422±0.078 |
| DI-GINE | 0.591±0.110 | 0.321±0.180 | 0.746±0.101 | -0.067±0.370 | 0.400±0.029 | 0.721±0.092 | 0.456±0.096 | 0.452±0.055 |
| BI-GINE | 0.664±0.100 | 0.480±0.157 | 0.839±0.058 | 0.367±0.455 | 0.358±0.141 | 0.784±0.138 | 0.178±0.891 | 0.524±0.183 |
| GAT | -0.678±0.631 | -3.168±1.210 | -0.148±0.096 | -1.000±0.863 | -2.225±0.459 | -0.720±0.180 | -1.189±0.714 | -1.304±0.175 |
| DI-GAT | -0.500±0.236 | -3.425±0.885 | -0.585±0.176 | -0.326±0.830 | -1.617±0.173 | -0.740±0.242 | -1.399±1.197 | -1.227±0.120 |
| BI-GAT | -0.085±0.199 | -3.591±0.679 | 0.237±0.091 | -0.910±0.456 | -1.115±0.272 | -0.332±0.103 | -0.223±0.447 | -0.860±0.174 |
| GPS-P | -0.091±0.111 | -0.121±0.177 | 0.010±0.205 | -0.313±0.439 | 0.034±0.062 | -0.056±0.262 | 0.207±0.158 | -0.047±0.151 |
| DI-GPS-P | -0.071±0.086 | -0.339±0.519 | -0.076±0.071 | -6.422±4.577 | -0.052±0.072 | -0.208±0.172 | -0.591±0.484 | -1.108±0.654 |
| BI-GPS-P | -0.217±0.151 | -0.156±0.267 | -0.090±0.261 | -1.302±0.645 | 0.053±0.087 | -0.304±0.238 | -0.504±0.366 | -0.360±0.168 |
| DGCN | -0.294±0.183 | 0.000±0.190 | -0.116±0.164 | -1.071±0.949 | 0.020±0.093 | -0.370±0.232 | -0.254±0.672 | -0.298±0.243 |
| DiiGCN | -27.500±18.458 | -20.171±15.704 | -2.100±1.705 | -18.882±10.121 | -14.088±33.349 | -4.007±2.785 | -15.008±10.012 | -14.537±8.681 |
| MagNet | -0.694±0.298 | -1.434±0.191 | 0.076±0.039 | -5.717±0.581 | -2.620±0.332 | -0.026±0.065 | -0.485±0.201 | -1.557±0.090 |

Table 20: ID performance with 'R2' metric on the Time dataset to predict setup slack.

| Method | hold | | | | setup | | | |
|---|---|---|---|---|---|---|---|---|
| | xtea | | synth_ram | | xtea | | synth_ram | |
| | mse | r2 | mse | r2 | mse | r2 | mse | r2 |
| GCN | 6.745±0.853 | -0.062±0.134 | 7.511±1.387 | -1.030±0.375 | 117.599±4.681 | -2.963±0.157 | 509.769±16.052 | -984.202±31.023 |
| DI-GCN | 7.010±0.592 | -0.104±0.093 | 9.764±2.355 | -1.640±0.636 | 116.332±16.048 | -2.920±0.540 | 418.819±24.262 | -808.427±46.890 |
| BI-GCN | 5.675±0.784 | 0.105±0.123 | 2.446±0.255 | 0.338±0.069 | 100.915±4.158 | -2.401±0.140 | 371.644±10.433 | -717.255±20.164 |
| GINE | 4.446±1.623 | 0.299±0.255 | 2.482±1.650 | 0.328±0.446 | 62.927±28.904 | -1.120±0.974 | 1341.500±430.261 | -2591.639±831.541 |
| DI-GINE | 3.075±0.887 | 0.515±0.139 | 9.545±4.222 | -1.581±1.141 | 61.694±6.981 | -1.079±0.235 | 612.807±101.175 | -1183.337±195.535 |
| BI-GINE | 2.116±0.486 | 0.666±0.076 | 1.314±1.197 | 0.644±0.323 | 83.775±17.953 | -1.823±0.605 | 498.357±239.343 | -962.146±462.565 |
| GAT | 7.446±1.012 | -0.173±0.159 | 7.741±1.932 | -1.093±0.522 | 96.486±7.970 | -2.251±0.268 | 347.589±30.441 | -670.765±58.833 |
| DI-GAT | 10.106±0.961 | -0.592±0.151 | 11.260±1.237 | -2.044±0.334 | 122.242±11.986 | -3.120±0.403 | 375.266±18.144 | -724.255±35.066 |
| BI-GAT | 5.699±1.155 | 0.101±0.181 | 2.683±0.927 | 0.274±0.250 | 78.499±6.022 | -1.645±0.202 | 463.886±29.786 | -895.527±57.565 |
| GPS-P | 7.241±0.520 | -0.140±0.082 | 5.487±1.351 | -0.483±0.365 | 124.375±114.416 | -3.191±3.856 | 82.559±46.321 | -158.557±89.522 |
| DI-GPS-P | 6.847±0.879 | -0.078±0.138 | 6.704±5.834 | -0.812±1.577 | 412.567±168.322 | -12.905±5.673 | 72.483±66.727 | -139.085±128.960 |
| BI-GPS-P | 7.284±0.676 | -0.147±0.106 | 4.192±1.584 | -0.133±0.428 | 69.680±21.461 | -1.348±0.723 | 572.724±92.117 | -1105.870±178.030 |
| DGCN | 13.362±0.281 | -1.105±0.044 | 13.303±0.332 | -2.597±0.089 | 66.196±22.392 | -1.231±0.754 | 539.409±140.777 | -1041.484±272.073 |
| DiGCN | 11.603±1.166 | -0.828±0.183 | 10.983±1.244 | -1.969±0.336 | 250.774±175.035 | -7.452±5.899 | 889.065±290.158 | -1717.245±560.771 |
| MagNet | 8.757±1.044 | -0.379±0.164 | 5.321±1.637 | -0.439±0.442 | 93.389±2.734 | -2.147±0.092 | 425.460±18.451 | -821.263±35.659 |
| Timer-inspired GNN | – | 0.9135 | – | 0.8656 | – | – | – | – |

Table 21: OOD performance on the TIME dataset.

| Method | densenets | mnasnets | mobilenetv2s | mobilenetv3s | nasbench201s | average |
|---|---|---|---|---|---|---|
| GCN | 43.042±1.663 | 7.764±0.775 | 30.222±0.817 | 5.981±1.454 | 2.134±0.772 | 17.828±0.802 |
| DI-GCN | 41.241±0.336 | 6.704±0.183 | 29.457±0.389 | 5.275±0.262 | 1.129±0.426 | 16.761±0.151 |
| BI-GCN | 41.271±1.729 | 8.054±0.399 | 29.473±0.399 | 5.365±0.159 | 2.114±0.731 | 17.256±0.238 |
| GIN | 7.694±0.442 | 1.833±0.187 | 3.535±0.343 | 1.491±0.343 | 0.617±0.216 | 3.034±0.172 |
| DI-GIN | 9.894±0.450 | 1.943±0.116 | 3.395±0.293 | 1.157±0.118 | 0.748±0.118 | 3.427±0.129 |
| BI-GIN | 7.615±0.247 | 1.901±0.151 | 3.339±0.172 | 1.177±0.160 | 0.493±0.069 | 2.905±0.098 |
| GAT | 14.272±4.836 | 4.160±0.460 | 8.733±0.936 | 3.227±0.456 | 0.577±0.071 | 6.194±1.242 |
| DI-GAT | 21.741±2.897 | 3.850±0.418 | 11.801±1.780 | 3.640±0.582 | 1.820±0.159 | 8.571±0.900 |
| BI-GAT | 9.322±0.535 | 2.371±0.469 | 4.957±0.497 | 1.723±0.481 | 0.498±0.086 | 3.774±0.262 |
| GPS-P | 7.080±0.302 | 2.125±0.294 | 3.951±0.289 | 1.692±0.395 | 0.723±0.197 | 3.114±0.132 |
| DI-GPS-P | 6.821±0.223 | 2.016±0.226 | 3.265±0.494 | 1.337±0.618 | 0.620±0.372 | 2.812±0.257 |
| BI-GPS-P | 6.863±0.265 | 1.971±0.222 | 3.794±0.311 | 1.399±0.361 | 0.618±0.260 | 2.929±0.149 |
| DGCN | 41.386±0.878 | 6.651±0.253 | 30.312±0.307 | 4.591±0.310 | 2.101±0.211 | 17.008±0.152 |
| DiGCN | 42.506±0.745 | 7.203±0.137 | 30.941±0.426 | 5.263±0.468 | 1.879±0.558 | 17.558±0.169 |
| MagNet | 7.368±0.189 | 1.910±0.152 | 3.015±0.280 | 1.290±0.315 | 0.514±0.166 | 2.819±0.177 |
| nn-meter | 7.1 | 3.19 | 3.25 | 2.03 | 0.44 | 3.20 |

Table 22: ID performance on the CG dataset on device 'Cortex A76 CPU' with 'rmse' metric.

| Method | densenets | mnasnets | mobilenetv3s | mobilenetv4s | nasbench202s | average |
|---|---|---|---|---|---|---|
| GCN | 0.217±0.034 | 0.267±0.033 | 0.118±0.026 | 0.239±0.090 | 0.290±0.183 | 0.226±0.037 |
| DI-GCN | 0.222±0.010 | 0.289±0.022 | 0.101±0.014 | 0.264±0.023 | 0.530±0.184 | 0.281±0.037 |
| BI-GCN | 0.245±0.038 | 0.269±0.012 | 0.111±0.022 | 0.287±0.010 | 0.272±0.152 | 0.236±0.031 |
| GIN | 0.876±0.028 | 0.863±0.054 | 0.680±0.050 | 0.723±0.078 | 0.716±0.190 | 0.771±0.052 |
| DI-GIN | 0.764±0.025 | 0.835±0.040 | 0.693±0.050 | 0.805±0.060 | 0.626±0.114 | 0.744±0.021 |
| BI-GIN | 0.892±0.020 | 0.842±0.047 | 0.681±0.033 | 0.861±0.053 | 0.858±0.059 | 0.826±0.024 |
| GAT | 0.662±0.107 | 0.456±0.067 | 0.359±0.032 | 0.476±0.060 | 0.786±0.056 | 0.547±0.050 |
| DI-GAT | 0.466±0.053 | 0.540±0.062 | 0.273±0.046 | 0.375±0.046 | 0.343±0.077 | 0.399±0.029 |
| BI-GAT | 0.808±0.034 | 0.746±0.077 | 0.511±0.065 | 0.689±0.116 | 0.844±0.091 | 0.719±0.051 |
| GPS-P | 0.904±0.019 | 0.809±0.051 | 0.650±0.041 | 0.666±0.126 | 0.697±0.173 | 0.745±0.057 |
| DI-GPS-P | 0.933±0.018 | 0.831±0.071 | 0.735±0.038 | 0.796±0.234 | 0.750±0.271 | 0.808±0.104 |
| BI-GPS-P | 0.922±0.020 | 0.836±0.048 | 0.666±0.051 | 0.738±0.141 | 0.750±0.215 | 0.782±0.072 |
| DGCN | 0.188±0.017 | 0.320±0.019 | 0.105±0.022 | 0.294±0.042 | 0.269±0.042 | 0.235±0.009 |
| DiGCN | 0.199±0.025 | 0.289±0.035 | 0.085±0.014 | 0.288±0.042 | 0.309±0.111 | 0.233±0.017 |
| MagNet | 0.896±0.012 | 0.835±0.060 | 0.730±0.048 | 0.830±0.113 | 0.851±0.108 | 0.828±0.062 |
| nn-meter | 0.931 | 0.824 | 0.676 | 0.738 | 0.824 | 0.798 |

Table 23: ID performance on the CG dataset on device 'Cortex A76 CPU' with 'acc5' metric.

| Method | densenets | mnasnets | mobilenetv4s | mobilenetv5s | nasbench203s | average |
|---|---|---|---|---|---|---|
| GCN | 0.434±0.041 | 0.511±0.042 | 0.222±0.031 | 0.453±0.122 | 0.515±0.233 | 0.426±0.053 |
| DI-GCN | 0.446±0.023 | 0.571±0.023 | 0.193±0.021 | 0.476±0.032 | 0.818±0.158 | 0.501±0.031 |
| BI-GCN | 0.481±0.025 | 0.483±0.026 | 0.243±0.018 | 0.536±0.029 | 0.517±0.199 | 0.451±0.037 |
| GIN | 0.998±0.004 | 0.997±0.004 | 0.936±0.024 | 0.941±0.044 | 0.972±0.035 | 0.968±0.016 |
| DI-GIN | 0.984±0.013 | 0.999±0.003 | 0.965±0.019 | 0.982±0.018 | 0.956±0.025 | 0.977±0.008 |
| BI-GIN | 1.000±0.000 | 0.996±0.005 | 0.952±0.018 | 0.991±0.017 | 0.994±0.012 | 0.986±0.007 |
| GAT | 0.924±0.091 | 0.808±0.039 | 0.645±0.063 | 0.730±0.060 | 0.973±0.017 | 0.816±0.036 |
| DI-GAT | 0.757±0.055 | 0.828±0.054 | 0.532±0.050 | 0.650±0.075 | 0.625±0.071 | 0.678±0.036 |
| BI-GAT | 0.993±0.009 | 0.977±0.034 | 0.821±0.057 | 0.937±0.081 | 0.994±0.007 | 0.944±0.026 |
| GPS-P | 1.000±0.000 | 0.996±0.006 | 0.918±0.033 | 0.901±0.080 | 0.928±0.085 | 0.949±0.034 |
| DI-GPS-P | 0.999±0.003 | 0.997±0.004 | 0.974±0.018 | 0.949±0.117 | 0.940±0.154 | 0.971±0.055 |
| BI-GPS-P | 0.999±0.003 | 0.998±0.004 | 0.938±0.021 | 0.922±0.085 | 0.953±0.078 | 0.962±0.030 |
| DGCN | 0.409±0.015 | 0.568±0.021 | 0.204±0.020 | 0.538±0.027 | 0.504±0.076 | 0.444±0.010 |
| DiGCN | 0.390±0.017 | 0.543±0.017 | 0.186±0.022 | 0.522±0.044 | 0.582±0.156 | 0.444±0.029 |
| MagNet | 1.000±0.000 | 0.996±0.005 | 0.973±0.028 | 0.983±0.031 | 0.988±0.019 | 0.988±0.015 |
| nn-meter | 0.999 | 0.992 | 0.977 | 0.990 | 0.999 | 0.991 |

Table 24: ID performance on the CG dataset on device 'Cortex A76 CPU' with 'acc10' metric.

| Method | proxylessnass | | | resnets | | |
|---|---|---|---|---|---|---|
| | rmse | acc5 | acc10 | rmse | acc5 | acc10 |
| GCN | 38.016±9.318 | 0.130±0.018 | 0.237±0.040 | 529.239±19.212 | 0.012±0.007 | 0.017±0.009 |
| DI-GCN | 33.581±1.016 | 0.143±0.019 | 0.264±0.016 | 517.553±4.419 | 0.010±0.002 | 0.020±0.003 |
| BI-GCN | 38.612±2.911 | 0.112±0.010 | 0.224±0.007 | 524.188±11.204 | 0.008±0.009 | 0.016±0.011 |
| GIN | 18.768±4.710 | 0.156±0.069 | 0.333±0.123 | 326.443±37.550 | 0.059±0.024 | 0.112±0.038 |
| DI-GIN | 10.742±0.876 | 0.306±0.021 | 0.566±0.051 | 455.235±16.044 | 0.032±0.010 | 0.073±0.012 |
| BI-GIN | 11.097±2.002 | 0.329±0.061 | 0.589±0.096 | 362.930±29.676 | 0.046±0.015 | 0.087±0.013 |
| GAT | 20.289±4.934 | 0.208±0.059 | 0.396±0.090 | 467.151±29.254 | 0.012±0.017 | 0.025±0.026 |
| DI-GAT | 19.306±3.641 | 0.192±0.043 | 0.386±0.063 | 386.181±44.349 | 0.033±0.023 | 0.069±0.030 |
| BI-GAT | 14.833±4.066 | 0.237±0.070 | 0.471±0.132 | 472.310±24.953 | 0.019±0.011 | 0.036±0.014 |
| GPS-P | 11.952±2.043 | 0.275±0.042 | 0.527±0.104 | 473.207±15.942 | 0.013±0.007 | 0.023±0.011 |
| DI-GPS-P | 10.122±0.911 | 0.293±0.038 | 0.588±0.075 | 490.252±11.336 | 0.005±0.005 | 0.012±0.009 |
| BI-GPS-P | 12.188±1.565 | 0.249±0.054 | 0.506±0.074 | 475.745±14.259 | 0.005±0.009 | 0.015±0.012 |
| DGCN | 28.038±1.707 | 0.123±0.016 | 0.274±0.015 | 535.961±5.274 | 0.003±0.005 | 0.007±0.007 |
| DiGCN | 26.308±1.739 | 0.138±0.017 | 0.287±0.022 | 542.416±5.718 | 0.001±0.002 | 0.005±0.004 |
| MagNet | 9.282±1.321 | 0.433±0.069 | 0.725±0.055 | 483.296±9.749 | 0.015±0.010 | 0.024±0.017 |
| nn-meter | 3.18 | 0.846 | 1.00 | 7.19 | 0.845 | 0.999 |

Table 25: OOD performance on the CG dataset on device 'Cortex A76 CPU'.

| Method | densenets | mnasnets | mobilenetv3s | mobilenetv4s | nasbench202s | average |
|---|---|---|---|---|---|---|
| GCN | 6.330±0.099 | 1.192±0.035 | 5.075±0.054 | 0.811±0.034 | 0.110±0.029 | 2.704±0.016 |
| DI-GCN | 6.226±0.061 | 1.165±0.029 | 5.088±0.035 | 0.795±0.032 | 0.128±0.036 | 2.681±0.013 |
| BI-GCN | 6.233±0.140 | 1.173±0.094 | 4.091±0.158 | 0.867±0.098 | 0.411±0.118 | 2.555±0.052 |
| GIN | 0.514±0.056 | 0.133±0.021 | 0.322±0.064 | 0.110±0.017 | 0.079±0.020 | 0.231±0.025 |
| DI-GIN | 0.909±0.064 | 0.176±0.017 | 0.385±0.036 | 0.156±0.029 | 0.250±0.022 | 0.375±0.019 |
| BI-GIN | 0.554±0.038 | 0.158±0.016 | 0.357±0.037 | 0.142±0.021 | 0.116±0.033 | 0.265±0.013 |
| GAT | 1.742±0.302 | 0.446±0.079 | 1.329±0.245 | 0.349±0.081 | 0.132±0.050 | 0.800±0.102 |
| DI-GAT | 1.904±0.173 | 0.531±0.087 | 1.810±0.414 | 0.481±0.068 | 0.497±0.022 | 1.044±0.133 |
| BI-GAT | 0.854±0.057 | 0.192±0.048 | 0.510±0.094 | 0.169±0.035 | 0.080±0.020 | 0.361±0.039 |
| GPS-P | 0.313±0.021 | 0.131±0.015 | 0.286±0.035 | 0.104±0.013 | 0.069±0.025 | 0.181±0.009 |
| DI-GPS-P | 0.320±0.020 | 0.149±0.019 | 0.321±0.081 | 0.147±0.029 | 0.186±0.134 | 0.225±0.032 |
| BI-GPS-P | 0.486±0.062 | 0.197±0.039 | 0.508±0.117 | 0.174±0.047 | 0.124±0.072 | 0.298±0.054 |
| DGCN | 6.695±0.428 | 1.267±0.101 | 5.396±0.050 | 0.817±0.200 | 0.614±0.079 | 2.958±0.074 |
| DiiGCN | 6.298±0.165 | 1.344±0.106 | 5.508±0.185 | 0.919±0.107 | 0.549±0.174 | 2.924±0.083 |
| MagNet | 0.478±0.032 | 0.120±0.015 | 0.230±0.017 | 0.106±0.008 | 0.060±0.008 | 0.199±0.008 |

Table 26: ID performance on the CG dataset on device 'Adreno 630 GPU' with 'rmse' metric.

| Method | densenets | mnasnets | mobilenetv3s | mobilenetv4s | nasbench202s | average |
|---|---|---|---|---|---|---|
| GCN | 0.325±0.032 | 0.383±0.019 | 0.119±0.015 | 0.410±0.035 | 0.943±0.052 | 0.435±0.013 |
| DI-GCN | 0.351±0.014 | 0.404±0.023 | 0.127±0.031 | 0.442±0.047 | 0.910±0.076 | 0.447±0.020 |
| BI-GCN | 0.354±0.030 | 0.410±0.040 | 0.164±0.031 | 0.424±0.038 | 0.485±0.100 | 0.367±0.028 |
| GIN | 1.000±0.000 | 1.000±0.000 | 0.956±0.032 | 1.000±0.000 | 0.994±0.009 | 0.990±0.007 |
| DI-GIN | 0.986±0.014 | 1.000±0.000 | 0.916±0.023 | 0.997±0.006 | 0.703±0.062 | 0.920±0.009 |
| BI-GIN | 1.000±0.000 | 1.000±0.000 | 0.917±0.016 | 0.991±0.012 | 0.945±0.057 | 0.970±0.012 |
| GAT | 0.872±0.054 | 0.864±0.068 | 0.515±0.070 | 0.862±0.065 | 0.907±0.131 | 0.804±0.050 |
| DI-GAT | 0.839±0.049 | 0.790±0.073 | 0.344±0.085 | 0.768±0.047 | 0.466±0.052 | 0.641±0.046 |
| BI-GAT | 0.989±0.005 | 0.992±0.013 | 0.829±0.049 | 0.993±0.012 | 0.994±0.009 | 0.959±0.011 |
| GPS-P | 1.000±0.000 | 1.000±0.000 | 0.966±0.021 | 1.000±0.000 | 0.994±0.015 | 0.992±0.005 |
| DI-GPS-P | 1.000±0.000 | 1.000±0.000 | 0.927±0.064 | 0.976±0.049 | 0.749±0.351 | 0.930±0.068 |
| BI-GPS-P | 1.000±0.000 | 0.995±0.005 | 0.861±0.043 | 0.972±0.043 | 0.892±0.156 | 0.944±0.045 |
| DGCN | 0.290±0.019 | 0.403±0.037 | 0.114±0.020 | 0.471±0.113 | 0.302±0.074 | 0.316±0.038 |
| DiiGCN | 0.302±0.028 | 0.432±0.042 | 0.118±0.022 | 0.441±0.054 | 0.363±0.146 | 0.331±0.031 |
| MagNet | 1.000±0.000 | 1.000±0.000 | 0.978±0.011 | 1.000±0.000 | 0.999±0.003 | 0.995±0.002 |

Table 27: ID performance on the CG dataset on device 'Adreno 630 GPU' with 'acc5' metric.

| Method | densenets | mnasnets | mobilenetv3s | mobilenetv4s | nasbench202s | average |
|---|---|---|---|---|---|---|
| GCN | 0.608±0.017 | 0.716±0.022 | 0.227±0.018 | 0.780±0.048 | 0.998±0.004 | 0.665±0.009 |
| DI-GCN | 0.614±0.014 | 0.741±0.021 | 0.225±0.015 | 0.792±0.048 | 0.992±0.013 | 0.672±0.013 |
| BI-GCN | 0.629±0.028 | 0.772±0.048 | 0.312±0.029 | 0.747±0.086 | 0.789±0.088 | 0.650±0.037 |
| GIN | 1.000±0.000 | 1.000±0.000 | 0.994±0.007 | 1.000±0.000 | 1.000±0.000 | 0.999±0.001 |
| DI-GIN | 1.000±0.000 | 1.000±0.000 | 0.994±0.005 | 1.000±0.000 | 0.904±0.010 | 0.979±0.002 |
| BI-GIN | 1.000±0.000 | 1.000±0.000 | 0.993±0.006 | 1.000±0.000 | 0.980±0.024 | 0.994±0.004 |
| GAT | 0.989±0.017 | 0.992±0.013 | 0.802±0.079 | 0.989±0.018 | 0.993±0.018 | 0.953±0.024 |
| DI-GAT | 0.991±0.005 | 0.980±0.020 | 0.630±0.106 | 0.962±0.014 | 0.715±0.033 | 0.855±0.029 |
| BI-GAT | 1.000±0.000 | 1.000±0.000 | 0.985±0.012 | 1.000±0.000 | 1.000±0.000 | 0.997±0.002 |
| GPS-P | 1.000±0.000 | 1.000±0.000 | 0.998±0.006 | 1.000±0.000 | 1.000±0.000 | 0.999±0.001 |
| DI-GPS-P | 1.000±0.000 | 1.000±0.000 | 0.988±0.021 | 1.000±0.000 | 0.947±0.157 | 0.986±0.030 |
| BI-GPS-P | 1.000±0.000 | 1.000±0.000 | 0.990±0.010 | 0.998±0.003 | 0.961±0.059 | 0.990±0.012 |
| DGCN | 0.564±0.022 | 0.708±0.031 | 0.200±0.016 | 0.781±0.103 | 0.611±0.099 | 0.573±0.041 |
| DiiGCN | 0.597±0.017 | 0.721±0.031 | 0.220±0.033 | 0.724±0.088 | 0.613±0.197 | 0.575±0.054 |
| MagNet | 1.000±0.000 | 1.000±0.000 | 1.000±0.000 | 1.000±0.000 | 1.000±0.000 | 1.000±0.000 |

Table 28: ID performance on the CG dataset on device 'Adreno 630 GPU' with 'acc10' metric.

| Method | proxylessnass | | | resnets | | |
|---|---|---|---|---|---|---|
| | rmse | acc5 | acc10 | rmse | acc5 | acc10 |
| GCN | 6.517±0.386 | 0.150±0.019 | 0.275±0.017 | 111.019±1.367 | 0.006±0.005 | 0.015±0.007 |
| DI-GCN | 6.135±0.149 | 0.154±0.021 | 0.305±0.014 | 111.631±1.095 | 0.007±0.005 | 0.012±0.006 |
| BI-GCN | 5.314±0.297 | 0.150±0.013 | 0.294±0.023 | 98.142±2.642 | 0.002±0.004 | 0.006±0.003 |
| GIN | 3.252±0.272 | 0.279±0.046 | 0.512±0.024 | 90.353±5.675 | 0.030±0.012 | 0.066±0.017 |
| DI-GIN | 3.121±0.549 | 0.271±0.076 | 0.514±0.079 | 109.263±2.076 | 0.028±0.013 | 0.057±0.011 |
| BI-GIN | 3.307±0.184 | 0.306±0.044 | 0.538±0.020 | 99.362±2.257 | 0.030±0.017 | 0.053±0.020 |
| GAT | 3.082±0.419 | 0.252±0.065 | 0.479±0.084 | 95.114±5.394 | 0.011±0.014 | 0.025±0.030 |
| DI-GAT | 3.982±0.341 | 0.172±0.019 | 0.355±0.027 | 92.212±9.020 | 0.035±0.019 | 0.072±0.029 |
| BI-GAT | 3.461±0.376 | 0.265±0.038 | 0.476±0.059 | 109.787±1.405 | 0.003±0.005 | 0.005±0.009 |
| GPS-P | 2.554±0.146 | 0.304±0.068 | 0.639±0.022 | 104.410±1.863 | 0.010±0.005 | 0.019±0.006 |
| DI-GPS-P | 2.515±0.148 | 0.270±0.070 | 0.614±0.061 | 105.275±2.238 | 0.006±0.010 | 0.009±0.013 |
| BI-GPS-P | 2.793±0.261 | 0.281±0.064 | 0.542±0.091 | 104.383±3.367 | 0.009±0.007 | 0.018±0.011 |
| DGCN | 5.794±0.249 | 0.160±0.012 | 0.293±0.026 | 116.229±1.179 | 0.006±0.006 | 0.017±0.010 |
| DiGCN | 6.042±0.906 | 0.140±0.023 | 0.271±0.043 | 117.250±1.149 | 0.006±0.005 | 0.013±0.008 |
| MagNet | 3.120±0.409 | 0.434±0.077 | 0.599±0.070 | 109.707±2.288 | 0.001±0.002 | 0.003±0.003 |

Table 29: OOD performance on the CG dataset on device 'Adreno 630 GPU'.

| Method | densenets | mnasnets | mobilenetv3s | mobilenetv4s | nasbench202s | average |
|---|---|---|---|---|---|---|
| GCN | 4.754±0.087 | 1.141±0.032 | 4.717±0.039 | 0.820±0.042 | 0.184±0.075 | 2.323±0.026 |
| DI-GCN | 4.899±0.188 | 1.159±0.064 | 4.869±0.179 | 0.834±0.070 | 0.162±0.102 | 2.385±0.084 |
| BI-GCN | 4.712±0.095 | 1.150±0.061 | 3.834±0.104 | 0.887±0.142 | 0.364±0.116 | 2.190±0.042 |
| GIN | 0.847±0.089 | 0.249±0.176 | 0.518±0.161 | 0.216±0.076 | 0.539±0.511 | 0.474±0.103 |
| DI-GIN | 0.639±0.034 | 0.119±0.013 | 0.303±0.019 | 0.102±0.023 | 0.050±0.010 | 0.243±0.011 |
| BI-GIN | 0.310±0.022 | 0.100±0.011 | 0.231±0.015 | 0.099±0.028 | 0.039±0.013 | 0.156±0.011 |
| GAT | 1.630±0.299 | 0.441±0.088 | 1.585±0.311 | 0.388±0.052 | 0.131±0.081 | 0.835±0.127 |
| DI-GAT | 1.763±0.174 | 0.576±0.097 | 1.589±0.162 | 0.513±0.087 | 0.429±0.049 | 0.974±0.077 |
| BI-GAT | 0.830±0.082 | 0.269±0.026 | 0.834±0.114 | 0.264±0.030 | 0.062±0.020 | 0.452±0.032 |
| GPS-P | 0.303±0.035 | 0.132±0.020 | 0.335±0.043 | 0.112±0.026 | 0.106±0.062 | 0.197±0.015 |
| DI-GPS-P | 0.316±0.030 | 0.145±0.020 | 0.332±0.042 | 0.110±0.009 | 0.060±0.032 | 0.193±0.018 |
| BI-GPS-P | 0.296±0.012 | 0.118±0.016 | 0.303±0.049 | 0.083±0.011 | 0.108±0.071 | 0.182±0.026 |
| DGCN | 4.903±0.105 | 1.162±0.047 | 5.048±0.030 | 0.756±0.060 | 0.409±0.045 | 2.456±0.032 |
| DiiGCN | 4.807±0.174 | 1.274±0.118 | 5.139±0.174 | 0.834±0.091 | 0.470±0.075 | 2.505±0.094 |
| MagNet | 0.583±0.051 | 0.155±0.019 | 0.337±0.042 | 0.150±0.029 | 0.074±0.031 | 0.260±0.019 |

Table 30: ID performance on the CG dataset on device 'Adreno 640 GPU' with 'rmse' metric.

| Method | densenets | mnasnets | mobilenetv3s | mobilenetv4s | nasbench202s | average |
|---|---|---|---|---|---|---|
| GCN | 0.367±0.014 | 0.381±0.024 | 0.133±0.020 | 0.422±0.034 | 0.678±0.320 | 0.396±0.066 |
| DI-GCN | 0.357±0.016 | 0.416±0.035 | 0.139±0.029 | 0.450±0.047 | 0.782±0.272 | 0.429±0.059 |
| BI-GCN | 0.412±0.040 | 0.447±0.038 | 0.178±0.024 | 0.422±0.084 | 0.383±0.180 | 0.368±0.050 |
| GIN | 0.986±0.009 | 0.951±0.124 | 0.806±0.103 | 0.943±0.081 | 0.515±0.409 | 0.840±0.078 |
| DI-GIN | 0.999±0.003 | 1.000±0.000 | 0.962±0.023 | 1.000±0.000 | 0.997±0.006 | 0.991±0.004 |
| BI-GIN | 1.000±0.000 | 1.000±0.000 | 0.983±0.006 | 0.999±0.003 | 1.000±0.000 | 0.996±0.001 |
| GAT | 0.846±0.057 | 0.861±0.075 | 0.427±0.064 | 0.821±0.064 | 0.830±0.287 | 0.757±0.078 |
| DI-GAT | 0.836±0.042 | 0.744±0.080 | 0.373±0.053 | 0.674±0.101 | 0.424±0.058 | 0.610±0.047 |
| BI-GAT | 0.990±0.006 | 0.981±0.015 | 0.671±0.047 | 0.926±0.038 | 0.989±0.009 | 0.911±0.014 |
| GPS-P | 1.000±0.000 | 1.000±0.000 | 0.938±0.038 | 1.000±0.000 | 0.890±0.234 | 0.965±0.049 |
| DI-GPS-P | 1.000±0.000 | 1.000±0.000 | 0.950±0.032 | 0.998±0.006 | 0.994±0.018 | 0.988±0.009 |
| BI-GPS-P | 1.000±0.000 | 1.000±0.000 | 0.960±0.043 | 1.000±0.000 | 0.876±0.196 | 0.967±0.047 |
| DGCN | 0.343±0.020 | 0.409±0.028 | 0.122±0.022 | 0.510±0.077 | 0.441±0.088 | 0.365±0.017 |
| DiiGCN | 0.343±0.031 | 0.415±0.036 | 0.116±0.017 | 0.456±0.080 | 0.383±0.092 | 0.342±0.028 |
| MagNet | 0.999±0.003 | 0.999±0.003 | 0.922±0.028 | 0.990±0.014 | 0.966±0.054 | 0.975±0.016 |

Table 31: ID performance on the CG dataset on device 'Adreno 640 GPU' with 'acc5' metric.

| Method | densenets | mnasnets | mobilenetv3s | mobilenetv4s | nasbench202s | average |
|---|---|---|---|---|---|---|
| GCN | 0.694±0.011 | 0.747±0.018 | 0.249±0.013 | 0.768±0.061 | 0.957±0.090 | 0.683±0.024 |
| DI-GCN | 0.684±0.027 | 0.717±0.023 | 0.257±0.037 | 0.728±0.068 | 0.941±0.128 | 0.665±0.032 |
| BI-GCN | 0.704±0.022 | 0.757±0.047 | 0.346±0.031 | 0.715±0.107 | 0.712±0.178 | 0.646±0.054 |
| GIN | 1.000±0.000 | 1.000±0.000 | 0.959±0.048 | 0.997±0.006 | 0.635±0.396 | 0.918±0.076 |
| DI-GIN | 1.000±0.000 | 1.000±0.000 | 0.999±0.003 | 1.000±0.000 | 1.000±0.000 | 0.999±0.000 |
| BI-GIN | 1.000±0.000 | 1.000±0.000 | 1.000±0.000 | 1.000±0.000 | 1.000±0.000 | 1.000±0.000 |
| GAT | 0.992±0.010 | 0.991±0.014 | 0.701±0.104 | 0.985±0.009 | 0.985±0.038 | 0.930±0.026 |
| DI-GAT | 0.984±0.015 | 0.956±0.032 | 0.669±0.051 | 0.949±0.043 | 0.675±0.055 | 0.846±0.028 |
| BI-GAT | 1.000±0.000 | 1.000±0.000 | 0.926±0.020 | 0.999±0.003 | 1.000±0.000 | 0.985±0.004 |
| GPS-P | 1.000±0.000 | 1.000±0.000 | 0.998±0.004 | 1.000±0.000 | 0.990±0.022 | 0.997±0.005 |
| DI-GPS-P | 1.000±0.000 | 1.000±0.000 | 0.999±0.003 | 1.000±0.000 | 1.000±0.000 | 0.999±0.000 |
| BI-GPS-P | 1.000±0.000 | 1.000±0.000 | 1.000±0.000 | 1.000±0.000 | 0.986±0.023 | 0.997±0.004 |
| DGCN | 0.664±0.024 | 0.722±0.024 | 0.231±0.022 | 0.800±0.061 | 0.699±0.082 | 0.623±0.026 |
| DiiGCN | 0.662±0.031 | 0.703±0.027 | 0.218±0.030 | 0.745±0.067 | 0.617±0.114 | 0.589±0.031 |
| MagNet | 1.000±0.000 | 1.000±0.000 | 0.994±0.007 | 1.000±0.000 | 0.999±0.003 | 0.998±0.001 |

Table 32: ID performance on the CG dataset on device 'Adreno 640 GPU' with 'acc10' metric.

| Method | proxylessnass | | | resnets | | |
|---|---|---|---|---|---|---|
| | rmse | acc5 | acc10 | rmse | acc5 | acc10 |
| GCN | 5.600±0.162 | 0.169±0.018 | 0.303±0.018 | 82.509±1.149 | 0.006±0.004 | 0.011±0.004 |
| DI-GCN | 6.163±0.463 | 0.145±0.013 | 0.283±0.022 | 82.785±1.409 | 0.007±0.004 | 0.015±0.004 |
| BI-GCN | 5.039±0.203 | 0.147±0.012 | 0.279±0.021 | 70.904±1.024 | 0.001±0.002 | 0.003±0.004 |
| GIN | 3.704±0.383 | 0.267±0.057 | 0.488±0.069 | 65.961±7.773 | 0.029±0.014 | 0.067±0.027 |
| DI-GIN | 3.672±0.328 | 0.249±0.028 | 0.461±0.044 | 62.820±4.603 | 0.038±0.008 | 0.078±0.017 |
| BI-GIN | 3.326±0.218 | 0.285±0.031 | 0.535±0.042 | 69.651±5.469 | 0.037±0.010 | 0.067±0.016 |
| GAT | 3.646±0.751 | 0.212±0.058 | 0.408±0.089 | 66.092±7.723 | 0.018±0.012 | 0.039±0.029 |
| DI-GAT | 4.893±0.580 | 0.155±0.031 | 0.312±0.048 | 69.358±4.897 | 0.022±0.012 | 0.047±0.022 |
| BI-GAT | 3.666±0.544 | 0.246±0.057 | 0.453±0.080 | 76.832±4.857 | 0.000±0.001 | 0.001±0.003 |
| GPS-P | 2.789±0.093 | 0.293±0.086 | 0.627±0.021 | 74.002±1.594 | 0.006±0.005 | 0.011±0.006 |
| DI-GPS-P | 2.883±0.134 | 0.303±0.053 | 0.605±0.022 | 76.163±2.222 | 0.006±0.004 | 0.016±0.011 |
| BI-GPS-P | 2.747±0.101 | 0.283±0.063 | 0.653±0.020 | 72.210±0.404 | 0.011±0.002 | 0.020±0.005 |
| DGCN | 4.985±0.157 | 0.160±0.016 | 0.300±0.018 | 85.713±1.394 | 0.007±0.007 | 0.011±0.009 |
| DiGCN | 5.362±0.645 | 0.148±0.012 | 0.299±0.020 | 86.605±0.664 | 0.003±0.003 | 0.004±0.005 |
| MagNet | 3.309±0.187 | 0.431±0.030 | 0.594±0.043 | 77.379±1.987 | 0.004±0.006 | 0.005±0.008 |

Table 33: OOD performance on the CG dataset on device 'Adreno 640 GPU'.

## H.2 COMPARISON BETWEEN NPE AND EPE

| Method | gain | | PM | | BW | |
|---|---|---|---|---|---|---|
| | mse | rmse | mse | rmse | mse | rmse |
| BI-GIN+NPE | **0.135±0.009** | **0.367±0.012** | 1.296±0.024 | 1.138±0.010 | 19.215±1.044 | 4.382±0.117 |
| BI-GINE+EPE | 0.149±0.009 | 0.386±0.012 | **1.283±0.033** | **1.132±0.014** | **17.399±0.644** | **4.170±0.076** |
| BI-GPS+NPE | 0.122±0.007 | 0.349±0.010 | 1.212±0.058 | 1.100±0.026 | 20.475±8.853 | 4.456±0.825 |
| BI-GPS+EPE | **0.115±0.008** | **0.339±0.011** | **1.206±0.090** | **1.097±0.040** | **18.153±2.235** | **4.253±0.256** |

Table 34: NPE v.s. EPE on ID data on the AMP dataset.

| method | dsp | | lut | | cp | |
|---|---|---|---|---|---|---|
| | mse | r2 | mse | r2 | mse | r2 |
| BI-GINE+NPE | 2.508±0.183 | 0.977±0.001 | 1.983±0.078 | 0.879±0.006 | 0.617±0.026 | 0.858±0.004 |
| BI-GINE+EPE | **2.127±0.085** | **0.981±0.000** | **1.729±0.096** | **0.895±0.007** | **0.607±0.022** | 0.857±0.007 |
| BI-GPS+NPE | 2.442±0.303 | 0.979±0.002 | 2.112±0.216 | 0.873±0.014 | 0.621±0.018 | 0.859±0.009 |
| BI-GPS+EPE | **2.133±0.148** | **0.981±0.001** | **1.957±0.125** | **0.883±0.011** | **0.602±0.017** | **0.861±0.007** |

Table 35: NPE v.s. EPE on ID data on the HLS dataset.

| | shared | | | | root | | | |
|---|---|---|---|---|---|---|---|---|
| | accuracy | precision | recall | f1 | accuracy | precision | recall | f1 |
| BI-GIN+NPE | **0.999±0.000** | **0.987±0.039** | **0.999±0.000** | **0.991±0.026** | **0.999±0.000** | **0.999±0.000** | **0.999±0.000** | **0.999±0.000** |
| BI-GIN+EPE | **0.999±0.001** | 0.974±0.079 | 0.974±0.078 | 0.974±0.079 | **0.999±0.000** | **0.999±0.000** | **0.999±0.000** | **0.999±0.000** |

Table 36: NPE v.s. EPE on ID data on the SR dataset.

| | method | blabla | usb_cdc_core | wbqspiflash | cic_decimator | picorv32a | zipdiv | usb | average |
|---|---|---|---|---|---|---|---|---|---|
| hold rmse | BI-GINE+NPE | 0.449±0.106 | 0.327±0.161 | 0.162±0.051 | 0.063±0.025 | 2.006±0.115 | 0.107±0.080 | 0.049±0.018 | 0.452±0.040 |
| | BI-GINE+EPE | **0.181±0.059** | **0.021±0.008** | **0.031±0.006** | **0.010±0.003** | **0.652±0.070** | **0.026±0.011** | **0.014±0.002** | **0.134±0.012** |
| hold R2 | BI-GINE+NPE | 0.995±0.001 | 0.834±0.081 | 0.940±0.018 | 0.928±0.028 | 0.855±0.008 | 0.970±0.022 | 0.928±0.026 | 0.921±0.018 |
| | BI-GINE+EPE | **0.998±0.000** | **0.988±0.004** | **0.981±0.003** | **0.993±0.002** | **0.954±0.004** | **0.993±0.002** | **0.986±0.002** | **0.985±0.001** |
| setup rmse | BI-GINE+NPE | 55.889±2.928 | 5.481±1.416 | 13.006±2.716 | 2.438±0.606 | 34.429±2.865 | 12.805±4.583 | 1.728±0.331 | 17.968±1.206 |
| | BI-GINE+EPE | **15.134±2.195** | **1.327±0.747** | **2.718±1.297** | **0.966±0.877** | **17.996±3.227** | **6.436±5.416** | **0.885±0.317** | **6.494±1.449** |
| setup R2 | BI-GINE+NPE | 0.142±0.044 | -0.021±0.264 | 0.326±0.140 | -0.149±0.285 | 0.173±0.068 | 0.364±0.227 | 0.306±0.132 | 0.163±0.093 |
| | BI-GINE+EPE | **0.767±0.033** | **0.752±0.139** | **0.859±0.067** | **0.544±0.413** | **0.567±0.077** | **0.680±0.268** | **0.644±0.127** | **0.688±0.091** |

Table 37: NPE v.s. EPE on ID data on the TIME dataset.

| | Method | densenets | mnasnets | mobilenetv2s | mobilenetv3s | nasbench201s | average |
|---|---|---|---|---|---|---|---|
| cpu rmse | BI-GIN+NPE | **7.734±0.602** | 2.053±0.235 | 3.788±0.424 | 1.590±0.371 | 0.725±0.173 | 3.178±0.168 |
| | BIGINE+EPE | 7.550±0.291 | **1.728±0.224** | **3.064±0.326** | **1.176±0.155** | **0.419±0.033** | **2.788±0.147** |
| cpu acc5 | BI-GIN+NPE | 0.893±0.014 | 0.821±0.056 | 0.653±0.052 | 0.705±0.153 | 0.545±0.351 | 0.723±0.104 |
| | BIGINE+EPE | **0.907±0.017** | **0.873±0.042** | **0.766±0.041** | **0.861±0.036** | **0.901±0.029** | **0.861±0.019** |
| cpu acc10 | BI-GIN+NPE | 0.999±0.003 | 0.994±0.006 | 0.940±0.025 | 0.907±0.066 | 0.940±0.060 | 0.956±0.028 |
| | BIGINE+EPE | **1.000±0.000** | **0.999±0.003** | **0.969±0.023** | **0.994±0.009** | **0.999±0.003** | **0.992±0.006** |
| gpu630 rmse | BI-GIN+NPE | 0.440±0.020 | 0.112±0.009 | **0.254±0.019** | 0.099±0.014 | **0.056±0.006** | 0.192±0.006 |
| | BIGINE+EPE | **0.293±0.024** | **0.109±0.020** | 0.272±0.035 | **0.092±0.021** | 0.073±0.027 | **0.168±0.009** |
| gpu630 acc5 | BI-GIN+NPE | **1.000±0.000** | **1.000±0.000** | **0.974±0.015** | **1.000±0.000** | **1.000±0.000** | **0.994±0.003** |
| | BIGINE+EPE | **1.000±0.000** | **1.000±0.000** | 0.968±0.026 | **1.000±0.000** | 0.978±0.052 | 0.989±0.010 |
| gpu630 acc10 | BI-GIN+NPE | **1.000±0.000** | **1.000±0.000** | **1.000±0.000** | **1.000±0.000** | **1.000±0.000** | **1.000±0.000** |
| | BIGINE+EPE | **1.000±0.000** | **1.000±0.000** | 0.999±0.003 | **1.000±0.000** | **1.000±0.000** | 0.999±0.000 |
| gpu640 rmse | BI-GIN+NPE | 0.355±0.031 | 0.146±0.017 | 0.289±0.024 | 0.157±0.031 | 0.071±0.018 | 0.204±0.009 |
| | BIGINE+EPE | **0.343±0.028** | **0.089±0.011** | **0.245±0.027** | **0.093±0.016** | **0.046±0.018** | **0.163±0.010** |
| gpu640 acc5 | BI-GIN+NPE | **1.000±0.000** | **1.000±0.000** | 0.971±0.015 | 0.949±0.081 | 0.986±0.012 | 0.981±0.016 |
| | BIGINE+EPE | **1.000±0.000** | **1.000±0.000** | **0.987±0.013** | **1.000±0.000** | **1.000±0.000** | **0.997±0.002** |
| gpu640 acc10 | BI-GIN+NPE | **1.000±0.000** | **1.000±0.000** | 0.996±0.005 | **1.000±0.000** | 0.999±0.003 | 0.999±0.001 |
| | BIGINE+EPE | **1.000±0.000** | **1.000±0.000** | **1.000±0.000** | **1.000±0.000** | **1.000±0.000** | **1.000±0.000** |

Table 38: NPE v.s. EPE on ID data on the CG dataset.

| Method | gain | | PM | | BW | |
|---|---|---|---|---|---|---|
| | mse | rmse | mse | rmse | mse | rmse |
| BI-GIN+NPE | 0.303±0.046 | 0.549±0.042 | 1.379±0.027 | 1.174±0.011 | 25.967±1.646 | 5.093±0.161 |
| BI-GINE+EPE | **0.302±0.037** | **0.549±0.033** | **1.373±0.005** | **1.171±0.002** | **22.339±1.413** | **4.724±0.150** |
| BI-GPS+NPE | 0.314±0.030 | 0.560±0.027 | 1.315±0.050 | 1.146±0.021 | 26.607±10.277 | 5.087±0.897 |
| BI-GPS+EPE | **0.302±0.071** | **0.546±0.060** | **1.314±0.143** | **1.126±0.060** | **21.815±1.973** | **4.666±0.206** |

Table 39: NPE v.s. EPE on OOD data on the AMP dataset.

| | dsp | | lut | | cp | |
|---|---|---|---|---|---|---|
| method | mse | r2 | mse | r2 | mse | r2 |
| BI-GINE+NPE | 3.434±0.238 | 0.964±0.002 | 0.113±0.019 | 0.971±0.004 | **0.450±0.013** | **0.830±0.008** |
| BI-GINE+EPE | **3.243±0.098** | **0.966±0.000** | **0.102±0.019** | **0.973±0.005** | 0.452±0.022 | 0.823±0.011 |
| BI-GPS+NPE | 3.209±0.263 | 0.967±0.001 | 0.133±0.027 | 0.968±0.006 | 0.496±0.017 | 0.812±0.016 |
| BI-GPS+EPE | **3.205±0.026** | **0.968±0.001** | **0.102±0.017** | **0.972±0.003** | **0.474±0.017** | **0.830±0.006** |

Table 40: NPE v.s. EPE on OOD data on the HLS dataset.

| | shared | | | | root | | | |
|---|---|---|---|---|---|---|---|---|
| | accuracy | precision | recall | f1 | accuracy | precision | recall | f1 |
| BI-GIN+NPE | 0.712±0.027 | **0.510±0.103** | 0.591±0.021 | 0.502±0.032 | 0.696±0.057 | 0.556±0.066 | **0.616±0.072** | **0.567±0.068** |
| BI-GINE+EPE | **0.725±0.037** | **0.510±0.059** | **0.604±0.030** | **0.530±0.038** | **0.747±0.035** | **0.569±0.113** | 0.508±0.080 | 0.520±0.090 |

Table 41: NPE v.s. EPE on OOD data on the SR dataset.

| | Method | xtea | | synth_ram | |
|---|---|---|---|---|---|
| | | mse | r2 | mse | r2 |
| hold | BI-GINE+NPE | 6.936±0.914 | -0.092±0.144 | 0.743±0.313 | 0.798±0.084 |
| | BI-GINE+-EPE | **2.074±0.474** | **0.673±0.074** | **0.617±0.906** | **0.832±0.245** |
| setup | BI-GINE+NPE | 98.690±55.964 | -2.326±1.886 | 629.630±107.437 | -1215.849±207.638 |
| | BI-GINE+-EPE | **59.401±13.573** | **-1.002±0.457** | **619.030±136.176** | **-1195.363±263.179** |

Table 42: NPE v.s. EPE on OOD data on the TIME dataset.

| | Method | proxylessnass | | | resnets | | |
|---|---|---|---|---|---|---|---|
| | | rmse | acc5 | acc10 | rmse | acc5 | acc10 |
| cpu | BI-GIN+NPE | 15.045±5.642 | 0.264±0.138 | 0.487±0.199 | 388.079±36.250 | **0.041±0.014** | **0.073±0.024** |
| | BI-GINE+EPE | **11.049±0.909** | **0.310±0.024** | **0.588±0.049** | **381.432±28.102** | 0.036±0.009 | 0.072±0.016 |
| gpu630 | BI-GIN+NPE | **2.843±0.289** | 0.277±0.031 | 0.538±0.055 | **97.056±7.138** | **0.034±0.012** | **0.065±0.020** |
| | BI-GINE+EPE | 2.947±0.239 | **0.370±0.082** | **0.595±0.051** | 108.959±2.594 | 0.020±0.011 | 0.033±0.018 |
| gpu640 | BI-GIN+NPE | 3.591±0.286 | **0.303±0.062** | **0.549±0.050** | 76.590±4.263 | 0.024±0.009 | 0.048±0.011 |
| | BI-GINE+EPE | **3.419±0.266** | 0.295±0.034 | 0.525±0.050 | **68.530±3.603** | **0.037±0.020** | **0.065±0.023** |

Table 43: NPE v.s. EPE on OOD data on the CG dataset.

