# OpenReview forum: "A Benchmark on Directed Graph Representation Learning in Hardware Designs"
_ICLR.cc/2025/Conference — Submitted to ICLR 2025_

### Official Review · Reviewer_kctB · 2024-10-28

**Soundness:** 2
**Presentation:** 3
**Contribution:** 1
**Rating:** 3
**Confidence:** 5

**Summary:**

This paper proposes a benchmark for learning representations of directed graphs in the hardware design field. It comprises 5 datasets and covers 13 different tasks, from node-level to graph-level. The authors also conduct experiments to evaluate the critical components in the design of graph neural networks.

**Strengths:**

1. OOD testing is proposed for each dataset to evaluate the generalization ability of the models.
2. Through various and rich experiments, bidirected MPNN and NPE+EPE are highlighted to enhance the model performance and stability.

**Weaknesses:**

1. The logical relation between different datasets is relatively weak. For instance, although all these datasets belong to the 'hardware design' field, what is the relation between the graph of CNN and the circuit netlist? It could be better to organize corresponding datasets more logically, instead of just gathering them together. I suggest classifying these circuits into FPGA, ASIC and computational graph (CNN architecture), and with each category, the datasets could be better to cover more/complete design stages.
2. Another widely used benchmark, CircuitNet 2.0 [1], covers the full chip design flow, including synthesis, floorplan, powerplan, placement, CTS and routing. While in this paper, only some separate stages are involved. It could be better to cover more/complete stages for each circuit in the benchmark. After covering the full chip design flow, this benchmark could be better to evaluate the performance of newly designed GNN across various tasks. For example, AIG --(logic synthesis)--> synthesized netlist --(placement)--> netlist with gate coordinate/position. These intermediate results focus on different graph properties, and the benchmark with these datasets could be utilized to test the performance of GNN specialized for these (continuous) stages.
3. All datasets in this benchmark are created by previous works, undermining the contribution of the work. I think it is not sufficient to support to release a new comprehensive benchmark.
4. The stability of the "stable positional encoding" is questionable, as it struggles to handle larger graphs with more than 100,000 nodes, a scale which is common in hardware design.

**References**

[1] Jiang, Xun, et al. "CircuitNet 2.0: An Advanced Dataset for Promoting Machine Learning Innovations in Realistic Chip Design Environment." ICLR. 2024.

**Questions:**

1. In Table 1, the OOD experiments for benchmark symbolic reasoning are 32, 36, and 48-bit. However, in lines 223 to 225, the training/ID test uses multipliers before technology mapping, and the OOD testing data uses multipliers after technology mapping. This is inconsistent with the table.
2. In the TIME dataset, the authors only conduct experiments on small circuits with a maximum number of nodes 58,676 and a maximum number of edges 83,225. In the original paper, i.e. Guo et al. (2022), there exists larger circuits whose number of nodes and edges exceeds 100,000 even 200,000. Could the authors provide results on these larger circuits or explain why they only conduct experiments on small circuits?
3. In the OOD testing of the TIME dataset, what is the principle for selecting the circuits to test the OOD performance? Could the authors provide more details about "the difference in graph structures" mentioned in line 235?
4. Could the authors provide more details about the "permutation equivariant functions $\kappa$ and $\rho$"? How do they contribute to the stability of positional encoding?

---

> ### Author Response · Authors · 2024-11-23
> **Response to Reviewer kctB (1/4)**
>
> We sincerely thank the reviewer for their valuable time and insightful suggestions.
>
> We would like to highlight the importance of bridging hardware applications with the machine learning community to foster greater attention and collaboration. On one hand, our work offers valuable insights for hardware experts, enabling them to better select backbone models and integrate domain-specific knowledge for their tasks. On the other hand, by consolidating accessible hardware datasets, we aim to provide ML researchers with a convenient resource for testing their methods, ultimately supporting retrospective advancements in the hardware domain.
>
> Below we try our best to address the reviewers’ questions:
>
> -------------------------
>
> **[weakness 1.]** The logical relation between different datasets is relatively weak. For instance, although all these datasets belong to the 'hardware design' field, what is the relation between the graph of CNN and the circuit netlist? It could be better to organize corresponding datasets more logically, instead of just gathering them together. I suggest classifying these circuits into FPGA, ASIC and computational graph (CNN architecture), and with each category, the datasets could be better to cover more/complete design stages.
>
>
>
> **Response:** We agree with reviewer kct8’s argument that having a benchmark to cover a wide range of hardware applications is essential. However, we think determining how to categorize the relevant datasets **is a subjective manner**.
>
> We selected the datasets to represent hardware design tasks across various levels of abstraction from a bottom-up perspective (from the lowest device level, physical synthesis level to higher levels like mapping onto general-purpose hardware platforms) (See Section.3 for detail). The selected ones are representative of this content.
>
> We actually have considered and also agreed with the reviewer’s categorization of the datasets: ASIC, FPGA, and computational graphs. We ultimately did not go for this approach because: FPGA, ASIC, and computational graphs are not entirely orthogonal, though there are significant differences in the development flow (e.g., HLS, logic synthesis, and physical synthesis) between ASIC and FPGA, their problem formulations could share commonalities: e.g. (1) HLS tools leverage IR graphs for scheduling and binding regardless of ASIC or FPGA, indicating that if one uses GNNs to approximate the synthesis, the problem formulation of performance prediction could be the same for either ASIC or FPGA; (2) Gate-level netlists can be represented as AIGs, regardless of ASIC/FPGA-based technology mapping, providing a consistent structure as inputs to GNNs; (3) timing problems is to identify whether a signal will arrive within the required time, and this formulation remains consistent regardless of whether the implementation is based on ASIC or FPGA.

---

> > ### Comment · Reviewer_kctB · 2024-11-23
> > **Response to part 1**
> >
> > - Thank you for your detailed response.
> > - For the authors' perspective of organizing these datasets, which is "from the device level, physical syntheses, and finally to higher levels", the authors should focus on more to construct a complete design flow or design chain, to cover the proposed abstraction. Collecting previously created datasets are not enough to support the claim of "a comprehensive benchmark suite for the evaluation of GNNs on hardware designs".
> > - Another weakness of the paper is that, **all datasets are originally created by previous works**, which is also recognized by reviewer `deWG`. As a result, the contribution of this work is undermined.
> > - Besides, from the perspective of technique, the paper is not innovative enough. The method of using bidirected MPNN and NPN+EPE are not originally proposed by the authors. And the results of using these techniques are not impressive.

---

> ### Author Response · Authors · 2024-11-23
> **Response to Reviewer kctB (2/4)**
>
> ----------------
> **[weakness 2.]** Another widely used benchmark, CircuitNet 2.0 [1], covers the full chip design flow, including synthesis, floorplan, powerplan, placement, CTS and routing. While in this paper, only some separate stages are involved. It could be better to cover more/complete stages for each circuit in the benchmark. After covering the full chip design flow, this benchmark could be better to evaluate the performance of newly designed GNN across various tasks. For example, AIG --(logic synthesis)--> synthesized netlist --(placement)--> netlist with gate coordinate/position. These intermediate results focus on different graph properties, and the benchmark with these datasets could be utilized to test the performance of GNN specialized for these (continuous) stages.
>
> **Response:** As discussed in the related works, we acknowledge that CircuitNet1.0 and 2.0 are very good datasets which focus on providing data in the ASIC hardware design loop for domain experts. In most of the hardware design stages in CircuitNets (including floor plan, power plan, placement, and routing), we think CNNs are potentially the most suitable model architecture to solve the tasks and indeed CNNs were those used in those benchmarks. They may not be the best benchmark for DGRL of our interest.
>
> The two major reasons are: (1) spatial information and the relative locations between cells play a critical role in these stages, (2) many of these tasks rely on grid/mesh-based representations; e.g., power planning utilizes power grids/meshes to distribute power to macros, standard cells, and other cells; routing employs routing grids, with most available routers being grid-based and typically adhering to preferred horizontal or vertical routing directions.
>
> Drawing from the CircuitNet research line, they opted to develop a multi-modal model that combines GNNs with CNNs, rather than relying solely on a single GNN to address these tasks [1]. This suggests that although they may be used as datasets for DGRL, DGRL alone cannot achieve ideal performance on these datasets. .
>
> Therefore, we consider other hardware applications in this benchmark that better align with our goal.
>
> *[1] Zhao, Yuxiang, Zhuomin Chai, Xun Jiang, Yibo Lin, Runsheng Wang, and Ru Huang. "PDNNet: PDN-Aware GNN-CNN Heterogeneous Network for Dynamic IR Drop Prediction." arXiv preprint arXiv:2403.18569 (2024).*
>
> ----------------------------
> **[Question 1. ]** In Table 1, the OOD experiments for benchmark symbolic reasoning are 32, 36, and 48-bit. However, in lines 223 to 225, the training/ID test uses multipliers before technology mapping, and the OOD testing data uses multipliers after technology mapping. This is inconsistent with the table.
>
> **Response:** Sorry for the confusion. For the OOD evaluation of SR dataset, we use 24 bit pre-mapping data for training, 32, 36, 48 bit pre-mapping for in-distribution testing, and 32, 36, 48 bit post 7nm tech mapping for out of distribution testing.

---

> ### Author Response · Authors · 2024-11-23
> **Response to Reviewer kctB (3/4)**
>
> -----------------
> **[Question 2. and 3. ]** Q2. In the TIME dataset, the authors only conduct experiments on small circuits with a maximum number of nodes 58,676 and a maximum number of edges 83,225. In the original paper, i.e. Guo et al. (2022), there exists larger circuits whose number of nodes and edges exceeds 100,000 even 200,000. Could the authors provide results on these larger circuits or explain why they only conduct experiments on small circuits?
>
> Q3. In the OOD testing of the TIME dataset, what is the principle for selecting the circuits to test the OOD performance? Could the authors provide more details about "the difference in graph structures" mentioned in line 235?
>
>
> **Response:** At the time of submission, performing Laplacian decomposition for larger graphs faced scalability issues, and therefore we dropped the large graphs. We have now resolved this scalability issue with the help of sparse.linalg package from scipy. Within the limited time available during rebuttal, we re-auto-tuned GINE, BIGINE, BIGINE w/ NPE and BIGINE w/ EPE while maintaining the exact same data split as that in TIME[1]. The results below reports the MSE metric, which align with our conclusions, demonstrating that bidirected message passing effectively improves performance, and stable PE continues to further enhance the model’s performance. We will include additional methods, task setups, and other evaluation metrics in the final version.
>
> In-distribution results:
>
> |        method        |      blabla     |   usb_cdc_core  |   wbqspiflash   |  cic_decimator  |    picorv32a    |      zipdiv     |   ~~aes_cipher~~      |      ~~aes128~~     |      ~~aes256~~    |       BM64      |       des       |    genericfir   |     salsa20     |
> |:--------------------:|:---------------:|:---------------:|:---------------:|:---------------:|:---------------:|:---------------:|:---------------:|:---------------:|:---------------:|:---------------:|:---------------:|:---------------:|:---------------:|
> |         GINE         |   0.546±0.298   |   0.102±0.080   |   0.145±0.113   |   0.151±0.081   |   1.116±0.381   |   0.468±0.177   |   ~~3.066±0.405~~   |  ~~7.745±0.548~~   |   ~~8.014±0.535~~   | **0.308±0.231** |   1.998±0.249   |   0.130±0.137   |   0.327±0.084   |
> |        BIGINE        |   1.207±0.800   |   0.203±0.060   | **0.109±0.062** |   0.099±0.080   |   1.641±0.233   |   0.352±0.170   |   ~~3.301±0.510~~   |   ~~6.702±0.731~~   |   ~~7.253±0.734~~   |   0.542±0.275   |   2.153±0.360   | **0.110±0.037** |   0.359±0.150   |
> | BIGINE w/ NPE|   1.867±1.241   |   1.814±0.225   |   0.432±0.146   |   0.158±0.061   |   2.738±0.279   |   0.498±0.176   |   ~~3.250±0.661~~   |**~~4.767±0.461~~** |   ~~7.412±0.551~~   |   0.556±0.285   |   3.342±0.202   |   0.183±0.192   |   2.886±1.256   |
> |  BIGINE w/ EPE | **0.495±0.365** | **0.101±0.087** |   0.122±0.103   | **0.073±0.055** | **1.006±0.339** | **0.169±0.122** | ~~**2.580±0.529**~~ |   ~~5.761±0.370~~   | ~~**6.347±0.489**~~ |   0.329±0.320   | **1.794±0.310** |   0.111±0.111   | **0.242±0.093** |
>
>
> Out-of-distribution results:
>
>
> |     method    |   jpeg_encoder  |   usbf_device   |      ~~aes192~~      |       xtea      |       spm       |      y_huff     |    synth_ram    |
> |:-------------:|:---------------:|:---------------:|:----------------:|:---------------:|:---------------:|:---------------:|:---------------:|
> |      GINE     |   11.756±2.943  |   10.818±5.437  |   ~~45.438±5.938~~   |   2.869±0.751   |   0.587±0.496   |   1.139±2.518   |   2.454±0.530   |
> |     BIGINE    |   8.688±1.646   |   6.814±2.422   |   ~~24.406±6.208~~   |   5.454±1.675   |   8.373±6.953   | **0.396±0.456** |   1.878±0.590   |
> | BIGINE w/ NPE |   21.648±2.770  |   5.612±1.840   | ~~**10.521±2.041**~~ |   21.880±3.865  |   1.209±0.856   |   1.995±1.019   | **0.668±0.354** |
> | BIGINE w/ EPE | **6.486±1.308** | **2.894±1.742** |   ~~30.048±8.819~~   | **2.657±0.779** | **0.401±0.222** |   0.492±0.682   |   1.840±0.834   |
>
> ~~number~~: Withdrawn due to numerical issues.
>
> *[1] A Timing Engine Inspired Graph Neural Network Model for Pre-Routing Slack Prediction. Zizheng Guo1,2, Mingjie Liu3, Jiaqi Gu, Shuhan Zhang, David Z. Pan, Yibo Lin. DAC 22.*

---

> > ### Comment · Reviewer_kctB · 2024-11-23
> > **Response to part 3**
> >
> > - Thank you for providing additional experiments on larger graph of TIME dataset.
> > - The performance on large graphs in dataset TIME is not good as expected. For the given MSE metric, MSE on larger graphs (aes_cipher, aes128, aes256, des) is much higher than that on smaller graphs. The stability of these models on large graphs is not guaranteed. And the MSE on circuit aes192 in OOD testing is very high. I think these GNN backbone failed to learn the features of these large graphs.
> > - I am still confused about dataset split on dataset TIME. In section D.3.1 in the originally submitted paper, the authors mentioned that "For training and ID testing, we take the designs 'blabla', 'usb_cdc_core', 'wbqspiflash', 'cic_decimator', 'picorv32a', 'zipdiv', 'usb'." My question is, what is the training set and testing set? And the training set in the rebuttal response is also unclear.
> > - In the original paper, the authors selected R2 coefficient to evaluate the performance. However, in rebuttal phase, only MSE is reported, while MSE was not evaluated in the original paper. Could the authors provide the explanation for this inconsistency?
> > - The previous question 3, "In the OOD testing of the TIME dataset, what is the principle for selecting the circuits to test the OOD performance? Could the authors provide more details about "the difference in graph structures" mentioned in line 235?" Could the authors provide more details about this question?

---

> ### Author Response · Authors · 2024-11-23
> **Response to Reviewer kctB (4/4)**
>
> -----------------
> **[Question 4. ]** Could the authors provide more details about the "permutation equivariant functions  and "? How do they contribute to the stability of positional encoding?
>
> **Response:** We apologize for the lack of clarity in this section.
>
> For the permutation equivariant function  $\kappa$, we implement element-wise MLP layers. As to $\rho$, we adopt GNN layers.
> Detailed discussion of stability of PE could be found in [1]. In the revised manuscript, we have added a description in Section 4.2 and below Table 3 regarding more details or how EPE improves stability. Here, we provide a high-level explanation of the reason:
>
> Stable means ``Small perturbations to the input Laplacian should only induce a limited change of final positional encodings.’’ The key of EdgePE to achieve stability is the permutation equivariance as well as smoothness of $\kappa$ (  element-sise MLP layers), which ensures smooth weights $\kappa(\lambda)$ of eigenvectors' inner product across different eigenvalues.
>
> *[1] On the Stability of Expressive Positional Encodings for Graphs. Yinan Huang, William Lu, Joshua Robinson, Yu Yang, Muhan Zhang, Stefanie Jegelka, Pan Li. ICLR 2024.*

---

> > ### Comment · Reviewer_kctB · 2024-11-23
> > **Response to part 4**
> >
> > - Thank you for the detailed explanation. I have no more questions about the "permutation equivariant functions".

---

> ### Comment · Reviewer_kctB · 2024-11-23
> **Response to part 2**
>
> - Thank you for the detailed explanation.
> - I am afraid the authors misunderstand my point. I do not suggest the authors to evaluate a single GNN model on tasks proposed by CircuitNet series. Instead, I want to highlight the proposed benchmark should **cover the whole the design flow, or the perspective mentioned by the authors**, "from the device level, physical syntheses, and finally to higher levels". Current datasets in the benchmark came from different circuits/domains/sources, which are not relevant to each other.
> - Besides, the authors state that "CNNs are potentially the most suitable model architecture to solve the tasks (floorplan, powerplan, placement and routing)", which is too absolute. Many previous works focusing on floorplan [1,2,3], placement [4,5,6] and routing [7] also utilize GNN architectures. Some of these works combine both CNN and GNN, and others only incorporate GNN. This is not the reason why DGRL is not suitable for these tasks.
> - And for the PPA prediction tasks, such as congestion prediction [8], wirelength prediction [9] and timing prediction [10], GNN are powerful backbone to solve these tasks. A common feature of these tasks is that spatial information is encoded in the graph structure, and GNN have been proved to be successful practice. I think it is too absolute to say that CNNs are more suitable than GNNs for these tasks.
>
>
> **References**
>
> [1] Xu, Qi, et al. "GoodFloorplan: Graph convolutional network and reinforcement learning-based floorplanning." TCAD. 2021.
>
> [2] Amini, Mohammad, et al. "Generalizable floorplanner through corner block list representation and hypergraph embedding." SIGKDD. 2022.
>
> [3] Guan, Wenbo, et al. "Thermal-Aware Fixed-Outline 3-D IC Floorplanning: An End-to-End Learning-Based Approach." IEEE Transactions on Very Large Scale Integration (VLSI) Systems. 2023.
>
> [4] Mirhoseini, Azalia, et al. "A graph placement methodology for fast chip design." Nature. 2021.
>
> [5] Cheng, Ruoyu, and Junchi Yan. "On joint learning for solving placement and routing in chip design." NeurIPS. 2021.
>
> [6] Lai, Yao, et al. "Chipformer: Transferable chip placement via offline decision transformer." ICML. 2023.
>
> [7] Liu, Siting, et al. "Concurrent Sign-off Timing Optimization via Deep Steiner Points Refinement." DAC. 2023.
>
> [8] Yang, Shuwen, et al. "Versatile multi-stage graph neural network for circuit representation." NeurIPS. 2022.
>
> [9] Xie, Zhiyao, et al. "Net2: A graph attention network method customized for pre-placement net length estimation." ASP-DAC. 2021.
>
> [10] Guo, Zizheng, et al. "A timing engine inspired graph neural network model for pre-routing slack prediction." DAC. 2022.

---

> ### Author Response · Authors · 2024-11-24
> **Further Response to Reviewer kctB**
>
> Thanks for the timely response from reviewer kct8. Below we make further response to their comments.
>
> **1. Dataset Coverage and New data:**
>
> We respectfully disagree with Reviewer kct8's point. First, we emphasize that defining "coverage" is inherently subjective. While we agree that covering an entire design loop is a valid approach for dataset selection, we believe that selecting representative datasets from different levels of abstraction is also a legitimate and effective strategy.
>
> Regarding the lack of newly created datasets, we contend that benchmarking does not necessarily require the creation of new datasets per the conference policy on "call for paper". Instead, our contribution as a benchmark lies in evaluating state-of-the-art DGRL techniques on well-established datasets to ensure a fair comparison of their effectiveness. Our goal is to provide insights and practical guidance for future DGRL designs in real-world applications rather than focusing on data collection.
>
> Furthermore, we have open-sourced a modular toolbox to facilitate the customization of new hardware datasets, making it easier for future work to collect high-quality data and incorporate a broader range of tasks. This ensures that our contributions remain extensible and beneficial for the community.
>
> Interestingly, we note that in the reviewer’s initial feedback, they highlighted *“The authors collect datasets from previous papers to construct a comprehensive benchmark for graph representation learning in hardware design”* as a **strength** of our work. We wonder the reason they now critique this as a weakness, suggesting the need to create new datasets.
>
>
> **2. Method Novelty:**
>
> We want to first argue that our benchmark does include methodological innovation. We are the first to introduce the Positional Encoding (PE) method into hardware domain. We are also the first to further discuss how to effectively leverage PE on hardware data. Moreover,  to the best of our knowledge, no one has even tried to apply stable positional encoding to directed graphs prior to this work.
>
> In addition, we would like to point out that benchmark papers are not required to introduce methodological innovations. Instead, their purpose is to evaluate existing methods across datasets to provide valuable insights. For example, the CircuitNet series[1][2] did not propose any new methods but still made significant contributions.
>
> *[1] CircuitNet: A Generic Neural Network to Realize Universal Circuit Motif Modeling. Yansen Wang, Xinyang Jiang, Kan Ren, Caihua Shan, Xufang Luo, Dongqi Han, Kaitao Song, Yifei Shen, Dongsheng Li ICML 2023.*
>
> *[2] CircuitNet 2.0: An Advanced Dataset for Promoting Machine Learning Innovations in Realistic Chip Design Environment. Xun Jiang, zhuomin chai, Yuxiang Zhao, Yibo Lin, Runsheng Wang, Ru Huang. ICLR 2024.*
>
> **3. Task DIfference with circuitNet:**
>
> We have discussed the use of CNNs and GNNs with the authors of CircuitNet, but due to anonymity policies, we cannot disclose further details. According to experimental results, CNNs outperform GNNs in all the tasks except for TIME within CircuitNet. Regarding the timing task, we agree that GNNs perform better in this context. In fact, the paper[10] cited by reviewer kct8 is precisely the source of the dataset we used for the TIME task in our benchmark.
>
>
> **4. Experiments on TIME dataset:**
>
> We thank the reviewer for raising this issue, this also caught our attention. Upon further inspection of the pipeline for positional encoding and the calculation on these super-large graphs, we report that for the 'aes' series, numerical errors occurred during the Laplacian decomposition. Many values were computed as zero, resulting in inaccurate PE calculations. As a result, we have temporarily withdrawn the results for the 'aes' series graphs. Apologies for not conducting a thorough check earlier. We are actively exploring alternative packages to perform the computations. Once resolved, we will update the results promptly.
>
> For OOD definition: Each design in the TIME dataset has a distinct graph structure. We trained on the graph structures of some designs and tested on others, treating the latter as OOD. We followed the original paper's train-test split to define the ID and OOD partitions.
>
> As to data split: For in-distribution graphs, we randomly divided all endpoints into train, validation, and test sets for in-distribution testing. For OOD graphs, all endpoints were used exclusively for testing. Additionally, as noted in the paper, we provided both MSE and R² metrics for evaluation. Please refer to Appendices 17, 19, and 21 for details.

---

> ### Comment · Reviewer_kctB · 2024-11-25
>
> - Thank you for your detailed response. I sincerely appreciate the authors' efforts in addressing the concerns and questions.
> - I would like to insist my opinion that, collecting **all** datasets from previous works is not enough to support the work. This weakness is also recognized by other reviewers. I also want to emphasize that "*the intention of constructing a comprehensive a benchmark*" is a strength, but the current benchmark is not complex enough. "*Some of the datasets might not be representative or complex enough*" is also a weakness recognized by other reviewers. I would like to edit my original review to emphasize this point.
> - Furthermore, I believe the primary contribution of CircuitNet is in constructing new datasets, rather than simply evaluating current methods on existing datasets. For specific PPA prediction tasks, it is common for researchers to propose sophisticated or well-designed model architectures to improve performance or inject domain-specific expert knowledge into the model—particularly in the hardware domain, where different tasks exhibit distinct characteristics.
> - The contribution of "introducing positional encoding into the hardware domain" appears relatively trivial. It seems more like a trick than a robust method or approach to solve a genuine problem. Moreover, the stability of the proposed method has not been well demonstrated. The authors name it "**stable** positional encoding", but the performance is not good as expected on larger graphs with node number ranging from 100,000 to 200,000. The authors failed to provide experiments on larger graphs during the paper submission stage, and during the rebuttal phase, they claimed that "*Many values were computed as zero, resulting in inaccurate PE calculations. As a result, we have temporarily withdrawn the results for the 'aes' series graphs.*" This issue suggests that the stability of the positional encoding needs further enhancement.
> - Taking BI-GIN for task setup slack prediction (it is the rank 1st model shown in Table 4) in table 19, 20, 21 for instance. In the ID testing, the MSE 8.203 and R2 0.524 is not good enough, even on these smaller graphs. And the in the OOD testing, the MSE = 498.357±239.343 and R2 = -962.146±462.565. The inaccuracy is significant, suggesting that the stability of the model needs further improvement. Whether the general GNN backbone + PE can be applied to the larger graphs is still questionable.
> - In conclusion, I think the current benchmark lacks the necessary complexity and diversity. It only includes a few datasets from previous work, which do not fully capture the diversity of hardware design. Future efforts should focus on building a more comprehensive benchmark by running an EDA toolchain on newly collected circuits. Additionally, the stability of the proposed positional encoding needs to be further improved to better handle larger graphs. I would like to maintain my original rating score and again thank you for your thorough response.

---

> ### Author Response · Authors · 2024-11-30
>
> Thank you very much for your critical comments. We acknowledge that there are aspects of this work that are not yet perfect, and we are committed to improving them in the next version.
>
> We would like to clarify one point. This paper is positioned as a benchmark effort, and in our view, the key elements of a benchmark paper are the diversity of approaches/datasets considered and the insights derived from the benchmarking results. CircuitNet, as you mentioned, primarily focuses on constructing new datasets, with the models evaluated being standard machine learning techniques. In contrast, our work takes a different angle by leveraging existing datasets to evaluate a wide range of advanced graph and geometric machine learning techniques, rather than introducing new datasets.
>
> Regarding the evaluated models and techniques, other reviewers have agreed that our benchmark is sufficiently diversified. Additionally, the insights derived from the benchmarking results are valuable to the application domain. While we agree that the dataset aspect of this benchmark could be improved, we believe that our work already provides a comprehensive evaluation of diverse, advanced techniques for DGRL in many hardware tasks and datasets. These benchmarking results can serve as valuable evidence for researchers in the hardware domain to inform their future studies.
>
> Once again, we sincerely appreciate the reviewer’s efforts in identifying the weaknesses of this work. We are committed to addressing these issues and further improving the manuscript!

---

### Official Review · Reviewer_F5eJ · 2024-11-01

**Soundness:** 3
**Presentation:** 3
**Contribution:** 2
**Rating:** 6
**Confidence:** 4

**Summary:**

The paper presents a benchmark for directed graph representation learning with a focus on hardware design tasks. It includes several hardware design datasets and prediction tasks, on which the paper evaluates 21 DGRL models that vary in their backbones, message passing directions, etc. The paper also provides insights regarding the set of most helpful network architectures for enhancing model performance.

**Strengths:**

1. This paper proposes a well-rounded benchmark that spans five datasets with various prediction tasks at different abstraction levels, thereby being able to reflect a wide range of practical hardware design scenarios.
2. The paper also carries out an extensive model evaluation that leads to some unique insights, e.g., the importance of handling OOD cases for a more generalizable DGRL.

**Weaknesses:**

- It is not clear how novel the provided insights are as the rationale behind such insights is not clearly summarized in the text of sections 4 & 6.
- The paper claims to discover some of the novel model designs that are helpful to DGRL, yet the novelty is not highlighted in the paper. The best-performing model also happens to be the most complex model design, which is a combination of different techniques mentioned in section 4.
- The conclusion from the experiments is rather simple and direct, missing task-specific analysis and insights.

**Questions:**

1. The conclusion of the paper points out that some particular model design works the best for DGRL, so what are the reasons behind such results? Why does ‘Bidirected’ (BI) message passing show better accuracy?
2. For different tasks, is there more detailed guidance on which model should be used?
3. What is the main contribution of the paper, in terms of the guidance it provides for designing DGRL? It should be noted that EPE is expected to outperform NPE even before the experiment, the graph transformer-based method is well-known to have scalability issues, and traditional GNN is naturally more efficient when the graph is large. Maybe the paper can provide more discussion on how to cope with the scalability issue of graph transformer in large-scale DGRL, or how to involve more domain knowledge in the model design.

---

> ### Author Response · Authors · 2024-11-23
> **Response to Reviewer F5eJ (1/3)**
>
> We thank reviewer F5eJ for taking the time to review our paper and for providing valuable questions and suggestions that help us better position our work. Below, we try our best to address the reviewer’s concerns.
>
> ----------------------
> **[Weakness 1. and Question 1.]** W1: It is not clear how novel the provided insights are as the rationale behind such insights is not clearly summarized in the text of sections 4 & 6.
>
> Q1: The conclusion of the paper points out that some particular model design works the best for DGRL, so what are the reasons behind such results? Why does ‘Bidirected’ (BI) message passing show better accuracy?
>
> **Response:**
>
> For DGRL, bidirected message passing and stable PE helps the performance, the rationale is as follows:
>
> * *[bidirected message passing]* Mathematically, using different weights for message passing along different edge directions could add to the neural network’s distinguishing power when using such directional information. Hardware data exhibits complex logical and computational flows, often requiring such directional information exchange. Specifically, child nodes may need to comprehend information from parents, while parents simultaneously interpret information from their children.
>
> * *[positional encodings (PE)]*  Theoretically PE can increase the expressive power of graph neural networks. In hardware data, some key logic decisions may require expressive power beyond the 1-dimensional Weisfeiler-Lehman (WL) test. The WL test is an important measure of graph algorithms' expressive power. Notably, traditional GNNs have an expressive power limited to 1-WL[1]. With the inclusion of PE, GNNs can surpass the expressive power of 1-WL[2], and thus achieves better performance on the hardware data.
>
> * *[Stable PE]* Furthermore, Stable PE outperforms unstable PE because it has a guarantee to handle PE in a more stable and basis-invariant manner, which is explained in detail in Section 4.2 and Table 3. in our revised version. With complex structures, hardware data requires higher level of generalization for structural information, and thus we observe a significant performance improvement with stable PE on hardware data..
>
>
> *[1] How Powerful are Graph Neural Networks? Keyulu Xu, Weihua Hu, Jure Leskovec, Stefanie Jegelka. ICLR 2019.*
>
> *[2]Distance Encoding: Design Provably More Powerful Neural Networks for Graph Representation Learning. Pan Li, Yanbang Wang, Hongwei Wang, Jure Leskovec. Neurips 2020.*
>
> ---------------------
> **[Weakness 2.]** The paper claims to discover some of the novel model designs that are helpful to DGRL, yet the novelty is not highlighted in the paper. The best-performing model also happens to be the most complex model design, which is a combination of different techniques mentioned in section 4.
>
> **Response:** We would like to argue that `the best-performing model also happens to be the most complex model design’ is inaccurate. Our point is that EPE is essentially less complex than NPE:
>
> Although EPE seems to use more design strategies and technical ideas, To stably handle PE, the permutation equivariant functions in EPE require mapping the eigenvectors into a lower dimension, while NPE directly concatenates eigenvectors with the node feature and increases the feature dimension. NPE introduces higher-dimensional features, thus more complexity for the model.
>
> Therefore, a more complex model is not always better. Instead, to improve performance, it is crucial to design with techniques that are more strategically crafted for the problem.
>
> ----------------
> **[Weakness. 3]**  The conclusion from the experiments is rather simple and direct, missing task-specific analysis and insights.
>
> **Response:** Based on the previously listed insights to respond to weakness 1 and weakness 2, the conclusions from our benchmark are not direct. And we want to argue `simple’ conclusions are actually good but not bad, as it would be easier for hardware experts to apply them for their future tasks.
>
> We also fully agree that task-specific insights are very important. Actually, the models identified in our paper are ready to be further injected with domain-specific knowledge from hardware experts, though investigating task-specific insights is beyond the scope of this work.

---

> ### Author Response · Authors · 2024-11-23
> **Response to Reviewer F5eJ (2/3)**
>
> **[Question 2.]** For different tasks, is there more detailed guidance on which model should be used?
>
> **Response:** We fully acknowledge that different tasks may expect different models to help with performance. However, surprisingly, our effort in this benchmark just shows that there are also some  principles in designing these models (e.g. bidirected message passing and stable PE) that may generally help in hardware problems, which is an important contribution made by this work. Of course, more domain specifics may be further added to these models when hardware experts leverage the observed principle.

---

> ### Author Response · Authors · 2024-11-23
> **Response to Reviewer F5eJ (3/3)**
>
> -----------------------
>
> **[Question 3.]** What is the main contribution of the paper, in terms of the guidance it provides for designing DGRL? It should be noted that EPE is expected to outperform NPE even before the experiment, the graph transformer-based method is well-known to have scalability issues, and traditional GNN is naturally more efficient when the graph is large. Maybe the paper can provide more discussion on how to cope with the scalability issue of graph transformers in large-scale DGRL, or how to involve more domain knowledge in the model design.
>
> **Response:** We appreciate this highly insightful question.
>
> Q3.1) First, we discuss the two concepts:
>
> 1. How to cope with the scalability issue of graph transformers in large-scale DGRL.
>
> We have considered four potential approaches, such as 1) efficient transformer[1]; 2)subgraph sampling techniques[2]; 3) Local attention, which, while still using attention, restricts it to a certain number of hops; 4) flash attention techniques[3]. These approaches could serve as promising directions for future research.
>
> 2. How to involve more domain knowledge in the model design.
>
> One could further incorporate domain expertise, techniques such as knowledge-guided feature engineering (e.g. designing feature input better aligning with the input)[4][5][7] and subgraph feature summarization (e.g.Leverage domain knowledge to extract deeper features within the subgraph).[6][8][9].
>
> Q3.2) Second, we summarize our contributions as follows:
>
> * For the ML domain, we are the first to introduce hardware data benchmarks to evaluate modern DGRL approaches. With their uniqueness of logic/computational flow and long-distance dependence, hardware data serves as a unique and important testbed for DGRL research.
>
> * For the hardware domain, although it’s known that EPE outperforms NPE recently in ML community[10], to the best of our knowledge, no prior work has applied PE on hardware data before, let alone exploring methods for processing PE in the hardware context. We are the first to investigate the effectiveness of PE on hardware data and explore more effective methods for handling PE in this context.
>
>
> * For the hardware domain, we also share insights with hardware experts that bi-directed message passing could achieve the best performance among different message passing methods. Note that this is non-trivial, as highlighted in reviewer 8GB5's Question 1, bidirected message passing on directed graphs has not been widely recognized or explored within the hardware domain.
>
> *  We highlight the current challenge and urgent need for improving the OOD generalization capabilities of the DGRL methods when lacking abundant training graph structures.
>
>
> *[1]Rethinking Attention with Performers. Krzysztof Choromanski, Valerii Likhosherstov, David Dohan, Xingyou Song, Andreea Gane, Tamas Sarlos, Peter Hawkins, Jared Davis, Afroz Mohiuddin, Lukasz Kaiser, David Belanger, Lucy Colwell, Adrian Weller. ICLR 2021.*
>
> *[2]From Stars to Subgraphs: Uplifting Any GNN with Local Structure Awareness Lingxiao Zhao, Wei Jin, Leman Akoglu, Neil Shah. ICLR 2022.*
>
> *[3]FlashAttention: Fast and Memory-Efficient Exact Attention with IO-Awareness. Tri Dao, Daniel Y. Fu, Stefano Ermon, Atri Rudra, Christopher Ré. Neurips 2022.*
>
> *[4] Learning Semantic Representations to Verify Hardware Designs. Shobha Vasudevan, Wenjie (Joe) Jiang, David Bieber, Rishabh Singh, hamid shojaei, C. Richard Ho, Charles Sutton. Neurips 2021.*
>
> *[5]High Performance Graph Convolutional Networks with Applications in Testability Analysis. Yuzhe Ma, Haoxing Ren, Brucek Khailany, Harbinder Sikka, Lijuan Luo, Karthikeyan Natarajan, Bei Yu. DAC 19.*
>
> *[6] GNN-RE: Graph Neural Networks for Reverse Engineering of Gate-Level Netlists. Lilas Alrahis, Abhrajit Sengupta, Johann Knechtel, Satwik Patnaik, Hani Saleh, Baker Mohammad, Mahmoud Al-Qutayri, Ozgur Sinanoglu. IEEE Transactions on Computer-Aided Design of Integrated Circuits and Systems 2022.*
>
> *[7] GRANNITE: Graph Neural Network Inference for Transferable Power Estimation. Yanqing Zhang; Haoxing Ren; Brucek Khailany. DAC 20.*
>
> *[8]AppGNN: Approximation-Aware Functional Reverse Engineering using Graph Neural Networks. Tim Bucher, Lilas Alrahis, Guilherme Paim, Sergio Bampi, Ozgur Sinanoglu, Hussam Amrouch. ICCAD 22.*
>
> *[9]Deep H-GCN: Fast Analog IC Aging-Induced Degradation Estimation. Tinghuan Chen; Qi Sun; Canhui Zhan; Changze Liu; Huatao Yu; Bei Yu. IEEE Transactions on Computer-Aided Design of Integrated Circuits and Systems 2021.*
>
> *[10] On the Stability of Expressive Positional Encodings for Graphs. Yinan Huang, William Lu, Joshua Robinson, Yu Yang, Muhan Zhang, Stefanie Jegelka, Pan Li. ICLR 2024.*

---

> > ### Comment · Reviewer_F5eJ · 2024-11-25
> >
> > Thanks for the clarification and thoughtful answers to all my questions. I acknowledge the merit of the paper as it provides a benchmark of learning representations on various hardware design tasks. As pointed out in the review of kctB and other reviewers, with which I partially agree, the contribution of this paper is slightly limited due to the extensive use of existing algorithms and datasets without bringing insights that are task-specific (or specific to hardware design). Nevertheless, I will maintain a relatively positive score because of the thorough experiment and potential to serve as a solid baseline for future algorithm works.

---

> ### Author Response · Authors · 2024-11-30
>
> Thank you very much for your constructive feedback and objective evaluation. We agree with your observation that incorporating more task-specific insights would further enhance the quality of this work. We are also greatly encouraged by your recognition of the positive aspects of this study. Indeed, one research paper often can excel in every aspect, but we are committed to further improving the manuscript in its next version. Many thanks!

---

### Official Review · Reviewer_8G85 · 2024-11-03

**Soundness:** 3
**Presentation:** 3
**Contribution:** 3
**Rating:** 6
**Confidence:** 5

**Summary:**

Directed graph representation learning (DGRL) has become essential for handling the growing complexity of modern computing systems, especially in areas like circuit netlists and computational graphs. The authors state that the DGRL in the hardware domain remains underdeveloped due to the lack of robust, user-friendly benchmarks. This paper introduces a benchmark with five hardware design datasets and 13 prediction tasks across different circuit abstraction levels. Evaluating 21 DGRL models, including graph neural networks and graph transformers enhanced with positional encodings, the study finds that models using bidirected message passing networks (BI-MPNNs) and robust PEs perform best. Additionally, it highlights a critical need to improve out-of-distribution (OOD) generalization in DGRL, with a modular codebase provided for ease of evaluation by hardware and ML researchers.

**Strengths:**

+ Benchmark that integrates control conditions with data dependencies extracted from IR graphs

**Weaknesses:**

- Benchmark structure is not fully explained and especially the differences across the domains.
- Analysis of graph representation learning seems incomplete because only the size of the graphs has been considered.

**Questions:**

A) The paper finds that bidirected (BI) message passing neural networks can substantially improve the performance. Does it imply that the direction of the edges is not meaningful in the DAGs for HW design?
B) The plus sign in Figure 2 could be explained in the caption. It will be really helpful for understanding.
C) The paper refers to “control and data flow graphs” obtained from LLVM IR of C++ codes but fails to discuss actually the very first papers that provided this compiler and graph construction such as "A load balancing inspired optimization framework for exascale multicore systems: A complex networks approach." In 2017 IEEE/ACM International Conference on Computer-Aided Design (ICCAD), pp. 217-224. IEEE, 2017,  that enable further graph optimizations.
D) Section 4.3 mentioned “We auto-tune the hyper-parameters with seed 123 with 100 trial budgets and select the configuration with the best validation performance.” Can author specify the validation performance?
E) The discussion of related work on directed graph representation learning needs to consider both recent works on graph based presentations of compiler based hardware-software codesign and compiler based high level synthesis like "GAHLS: an optimized graph analytics based high level synthesis framework." Scientific Reports 13, no. 1 (2023); "High-level synthesis using the Julia language." arXiv preprint arXiv:2201.11522 (2022); "End-to-end programmable computing systems." Communications Engineering 2, no. 1 (2023); "CEDR: A compiler-integrated, extensible DSSoC runtime." ACM Transactions on Embedded Computing Systems 22, no. 2 (2023). Graph representation learning has also been used in device placement, etc.
F) For CG, why ‘Proxylessass’, ‘ResNets’, and ‘SqueezeNets are selected as OOD? Is there any reasoning behind?
G) Usually recent graph representation learning techniques are exploiting a number of graph properties like graph geometry. Can the authors comment also on the role of graph geometry?
H) A small minor issue is that IR does not seem to be defined but I assume the authors refer to IR as intermediate representation.

---

> ### Author Response · Authors · 2024-11-23
> **Response to Reviewer 8G85 (1/2)**
>
> We sincerely thank the reviewer for taking the time to review our paper and providing actionable suggestions that help us better position our work. Below, we try our best to address the issues raised by the reviewer.
>
> ------------------------
> **[weakness 1.]** Benchmark structure is not fully explained and especially the differences across the domains.
>
> **Response:** Thank you for highlighting this issue. Previously we provided an overview of benchmark structure in Figure 1. (data) and Figure 2. (method). To address reviewer’s concern, we add a more detailed explanation of benchmark selection in Section 3 (in blue): the tasks were chosen to cover different abstraction levels, ranging from the lowest level (operation amplifier), to the highest levels (hardware platforms).
>
> As to methods, we include the most advanced directed graph representation learning(DGRL) approaches, such as message-passing methods, graph transformers and positional encoding, as demonstrated in Figure 2 and Table 3.
>
> ---------------------------
> **[weakness 2.]** Analysis of graph representation learning seems incomplete because only the size of the graphs has been considered.
>
> **Response:** We would like to emphasize that this benchmark not only highlights the need for different models to varying graph sizes but also further provides insights as discussed in Section 6. in the paper:
>
> 1. For DGRL tasks with hardware data, we find bidirected message passing achieves the best performance. Mathematically, using different weights for message passing along different edge directions could add to the neural network’s distinguishing power when using the directional information. Hardware data exhibits complex logical and computational flows, often requiring directional information exchange. Specifically, child nodes may need to comprehend information from parent nodes, while parent nodes simultaneously interpret information from their children.
>
> 2. Positional encodings (PE) enhances the models’ performance.  Theoretically PE can increase the expressive power of graph neural networks. In hardware data, some key logic decisions may require expressive power beyond the 1-dimensional Weisfeiler-Lehman (WL) test. The WL test is an important measure of graph algorithms' expressive power. Notably, traditional GNNs have an expressive power limited to 1-WL[1]. With the inclusion of PE, GNNs can surpass the expressive power of 1-WL[2], resulting in better performance in hardware data.
>
> 3. Stable PE (EPE) achieves better performance than unstable PE (NPE). Furthermore, Stable PE outperforms unstable PE because they have a guarantee to handle PE in a more stable and basis-invariant manner, which is explained in detail in Section 4.2 and Table 3. in our revised version. Hardware data requires more generalizable structural and relative positional information. With complex structures, hardware data requires higher level of generalization for structural information, and thus we observe a significant performance improvement with stable PE on hardware data.
>
> *[1] How Powerful are Graph Neural Networks? Keyulu Xu, Weihua Hu, Jure Leskovec, Stefanie Jegelka. ICLR 2019.*
>
> *[2]Distance Encoding: Design Provably More Powerful Neural Networks for Graph Representation Learning. Pan Li, Yanbang Wang, Hongwei Wang, Jure Leskovec. Neurips 2020.*
>
> ------------------
> **[Question A]** The paper finds that bidirected (BI) message passing neural networks can substantially improve the performance. Does it imply that the direction of the edges is not meaningful in the DAGs for HW design?
>
> **Response:** Actually, the success of (BI) message passing  just indicates the opposite conclusion that reviewer 8G85 pointed out. We are sorry for the caused confusion. It seems that the confusion comes from the unclear definition of three different types of message passing ‘-’, ‘BI’ and ‘DI’:
>
> * `-’ denotes undirected message passing, which use the same set of parameters along both directions;
>
> * `DI’ refers to directed message passing, which refers to only conducting message passing along edge directions;
>
> * `BI’ stands for bidirected message passing, which applies different NN parameters along the forward and backward directions of the edges.
>
> BI consistently shows superior performance, because it has better distinguishing power of the child and parent nodes.

---

> ### Author Response · Authors · 2024-11-23
> **Response to Reviewer 8G85 (2/2)**
>
> **[Question B]** The plus sign in Figure 2 could be explained in the caption. It will be really helpful for understanding.
>
> **Response:** We are sorry for the caused confusion. The 'plus’ in Figure 2 means combination among 1)message passing direction, 2)GNN backbone selection/ transformer selection 3) positional encoding and. Elements within each rectangle could be combined with others in different rectangles, resulting in different DGRL methods. We have added an explanation in the caption below Figure 2.
>
> **[Question C and E]**  QC: The paper refers to “control and data flow graphs” obtained from LLVM IR of C++ codes but fails to discuss actually the very first papers that provided this compiler and graph construction such as  "A load balancing inspired optimization framework for exascale multicore systems: A complex networks approach." In 2017 IEEE/ACM International Conference on Computer-Aided Design (ICCAD), pp. 217-224. IEEE, 2017, that enable further graph optimizations.
>
> QE: The discussion of related work on directed graph representation learning needs to consider both recent works on graph based presentations of compiler based hardware-software codesign and compiler based high level synthesis like
> "GAHLS: an optimized graph analytics based high level synthesis framework." Scientific Reports 13, no. 1 (2023);
> "High-level synthesis using the Julia language." arXiv preprint arXiv:2201.11522 (2022); "End-to-end programmable computing systems." Communications Engineering 2, no. 1 (2023);
>  "CEDR: A compiler-integrated, extensible DSSoC runtime." ACM Transactions on Embedded Computing Systems 22, no. 2 (2023).
> Graph representation learning has also been used in device placement, etc.
>
> **Response:** We thank reviewer 8G85 for pointing out several relevant works that were not fully discussed in the paper. These references will better position our work. We have added them to the dataset introduction (Section 3)  and related works (Section 2), highlighted in blue for clarity.
>
> ------------------------
> **[Question D]** Section 4.3 mentioned “We auto-tune the hyper-parameters with seed 123 with 100 trial budgets and select the configuration with the best validation performance.” Can author specify the validation performance?
>
> **Response:** For each task, the dataset is divided into three parts: train, validation, and test. To perform hyperparameter tuning, we set the random seed to 123 and use auto-tuning algorithms to train the models on the training set. During this process, we identify the hyperparameters that yield the lowest validation loss on the validation set. Using these hyperparameters, the model is trained ten times with new random seeds ranging from 0 to 9 and tested on the test set, which remains unseen during hyperparameter tuning. The results on the test set are averaged across these ten runs, and the averaged results are reported in the paper.
>
> -----------------------
> **[Question F]** For CG, why ‘Proxylessass’, ‘ResNets’, and ‘SqueezeNets are selected as OOD? Is there any reasoning behind?
>
> **Response:** In the CG dataset, each neural network has a different graph structure and results in a distinct graph. Therefore, selecting any of the networks for training and the remaining graphs could be viewed as out-of-distribution. Then we just select `Proxyless’, `Resnets’ and `SqueezeNets’ as the OOD graphs.
>
>
> -------------------
> **[Question G]** Usually recent graph representation learning techniques are exploiting a number of graph properties like graph geometry. Can the authors comment also on the role of graph geometry?
>
> **Response:** We would like to seek clarification from the reviewer regarding the meaning of "geometry" in this context.
>
> Based on our understanding, "geometry" here may refer to the relationships between nodes as expressed by the graph structure. In our benchmark, we use PE to incorporate such graph geometry information: Essentially PE involves the decomposition of the graph Laplacian, which represents the distances between nodes within the graph. Adding PE could theoretically add to the expressiveness of graph neural networks, which is also shown to improve the empirical performance through our experiments on hardware graphs.
>
> --------------------
> **[Question H]** A small minor issue is that IR does not seem to be defined but I assume the authors refer to IR as intermediate representation.
>
> **Response:** Thanks for pointing this out. We have added this description to the manuscript.

---

> > ### Author Response · Authors · 2024-11-30
> >
> > Dear Reviewer, we were wondering if the response above has addressed your concerns. As it has been some time since we left these comments and the discussion period is approaching its close, we would greatly appreciate your feedback.
> >
> > Thank you in advance!

---

> ### Author Response · Authors · 2024-12-01
>
> Thank you for taking the time to review our response. We sincerely appreciate your support for this work! Your suggestion to use graph distance as a means to measure the out-of-distribution nature of graph datasets is extremely insightful. Exploring graph motif kernels, as you suggested, could indeed be a promising approach.
>
> Additionally, we can consider the adoption of the Fréchet Inception Distance (FID) to evaluate the distance between the latent node representations of these graphs. FID has been widely used to compare the distributions of image datasets [1] and might offer valuable insights in this context. Moreover, prior work [2] has analyzed the computational graphs of neural networks through the lens of network science/graph theory, which seems relevant to this discussion.
>
> [1] The Role of ImageNet Classes in Fréchet Inception Distance, Kynkäänniemi et al., ICLR 2023
>
> [2] Graph structure of neural networks, You et al., ICML 2020

---

### Official Review · Reviewer_deWG · 2024-11-04

**Soundness:** 3
**Presentation:** 3
**Contribution:** 3
**Rating:** 8
**Confidence:** 3

**Summary:**

This paper proposes a novel benchmark to evaluate GNN models on hardware designs. Hardware designs can generally be represented as directed graphs, such as the control data flow graph, circuit netlists, etc. Previous benchmark papers usually focus on one specific hardware area. To fill in the gap of a comprehensive benchmark, this benchmark contains five previous hardware design datasets in different areas, and there are 13 prediction tasks in total. The five areas include high-level synthesis, symbolic reasoning, pre-routing timing prediction, computational graphs, and operational amplifiers. To make it more user-friendly, this paper constructs a toolbox using PyTroch Geometric. Besides, it proposes edge positional embedding for the graph Transformer model, which is more stable than the regular node positional embedding. It conducts experiments on 21 GNN models. Bidirected GNN with edge positional embedding performs the best.

**Strengths:**

(1)   The datasets cover many types of hardware tasks, spanning different levels of hardware design. It could be beneficial to different levels of hardware design.

(2)   The provided toolbox is convenient to use.

(3)   The proposed stable positional embedding is reasonable.

(4)   This paper conducts extensive experiments to compare 21 GNN models and 13 tasks. It shows lots of work.

**Weaknesses:**

(1)   The datasets are all originally created by previous works.

(2)   Some of the datasets might not be representative or complex enough. For example, the HLS dataset does not contain any HLS pragmas. However, we usually need to add HLS pragmas when we use HLS. Also, according to Table 1, the HLS dataset only has 95 nodes on average, which are very small CDFGs. Real-world HLS applications have longer codes. There might be a gap between some of the datasets and real applications. Besides, it will be better if the paper contains a dataset selection criteria or motivation.

(3)   The formula of the edge positional embedding in Table 3 is not very clear. The authors could improve the writing of this subsection.

**Questions:**

(1)   In Table 5, sometimes NPE hurts the performance. Why does it happen, and why is EPE more stable?

(2)   You selected one dataset for each task. For some of the tasks, there might be more existing datasets. Why did you select these specific datasets?

---

> ### Author Response · Authors · 2024-11-23
> **Response to Reviewer deWG (1/3)**
>
> We appreciate reviewer deWG’s valuable time in evaluating our paper. We are grateful for the recognition of our benchmark and the insightful suggestions provided. Below, we have made every effort to address the reviewer’s questions.
>
> -----------------------
> **[Weakness 1.]**  The datasets are all originally created by previous works.
>
> **Response:** We acknowledge this limitation in our work. We would like to first point out that benchmarking does not necessarily involve new datasets. Instead, our primary contribution lies in benchmarking existing advanced DGRL techniques on well-established datasets to a fair comparison of their  effectiveness and to provide insights and support for future DGRL design in practical applications.
>
> Meanwhile, we open-source a modular toolbox, which facilitates customization of new hardware data, making it easier for future work to collect high-quality datasets and incorporate a broader range of tasks.
>
> ---------------------
> **[Weakness 2.]** Some of the datasets might not be representative or complex enough. For example, the HLS dataset does not contain any HLS pragmas. However, we usually need to add HLS pragmas when we use HLS. Also, according to Table 1, the HLS dataset only has 95 nodes on average, which are very small CDFGs. Real-world HLS applications have longer codes. There might be a gap between some of the datasets and real applications. Besides, it will be better if the paper contains a dataset selection criteria or motivation.
>
> **Response:** Thanks for reviewer deWG’s insightful suggestion. We totally agree that in practice HLS CDFG graphs can be much larger. While the average number of nodes in our HLS dataset is 95, the variance is considerable, ranging from 4 nodes to as many as 474 nodes, which could reflect some practical cases. To support this, we include the data statistics of a widely used real-world HLS benchmark PolyBench as follows, which on average has 146.5 nodes. (See the attached Table in Response 2/3)
>
> However, we know that the current CDFG dataset size might be still smaller than many practical HLS data. When we did this project, we did not have sufficient computational resources to generate those many large HLS graphs, so we just adopted the datasets in the published benchmark work[1]. But we would like to add larger and more practical cases and work with domain experts for this when there is an opportunity.
>
> *[1]High-Level Synthesis Performance Prediction using GNNs: Benchmarking, Modeling, and Advancing. Wu, Nan ; Yang, Hang ; Xie, Yuan ; Li, Pan ; Hao, Cong. DAC 22.*

---

> ### Author Response · Authors · 2024-11-23
> **Response to Reviewer deWG (2/3)**
>
> Attach: The statistics of the PolyBench dataset
>
> | **PolyBench**             | **# nodes** | **# edges** | **DSP** | **LUT**  | **FF**   | **SLICE** | **CP (ns)** |
> |---------------------------|-------------|-------------|---------|----------|----------|-----------|-------------|
> | kernel_gramschmidt        | 151         | 221         | 14      | 6550     | 11343    | 2559      | 8.254       |
> | kernel_jacobi_1d_imper    | 70          | 95          | 14      | 1056     | 1372     | 440       | 8.471       |
> | kernel_doitgen            | 95          | 137         | 21      | 1282     | 1392     | 574       | 8.649       |
> | kernel_correlation        | 237         | 361         | 21      | 7522     | 12618    | 2770      | 8.476       |
> | kernel_floyd_warshall     | 98          | 168         | 15      | 1523     | 2080     | 636       | 9.177       |
> | kernel_trisolv            | 75          | 119         | 14      | 4393     | 7665     | 1690      | 8.194       |
> | kernel_jacobi_2d_imper    | 117         | 164         | 21      | 1860     | 2460     | 801       | 8.452       |
> | kernel_cholesky           | 108         | 168         | 14      | 6468     | 11092    | 2491      | 8.697       |
> | kernel_atax               | 93          | 131         | 14      | 1078     | 1464     | 531       | 7.692       |
> | kernel_2mm                | 149         | 214         | 22      | 1698     | 2219     | 753       | 8.246       |
> | kernel_symm               | 102         | 147         | 29      | 1625     | 2489     | 727       | 8.413       |
> | kernel_bicg               | 80          | 110         | 14      | 1092     | 1612     | 512       | 7.825       |
> | kernel_dynprog            | 157         | 239         | 7       | 785      | 1135     | 368       | 6.527       |
> | kernel_covariance         | 155         | 239         | 20      | 4965     | 8380     | 1902      | 8.292       |
> | kernel_ludcmp             | 269         | 403         | 14      | 5763     | 8936     | 2056      | 9.039       |
> | kernel_reg_detect         | 243         | 363         | 0       | 795      | 1057     | 364       | 7.693       |
> | kernel_seidel_2d          | 117         | 168         | 6       | 4688     | 5180     | 1539      | 8.781       |
> | kernel_fdtd_apml          | 377         | 551         | 77      | 18806    | 28764    | 6136      | 9.466       |
> | kernel_durbin             | 153         | 217         | 17      | 1918     | 2183     | 737       | 8.150       |
> | kernel_gemm               | 80          | 110         | 18      | 1240     | 1707     | 556       | 7.870       |
> | kernel_trmm               | 76          | 122         | 18      | 1358     | 1810     | 555       | 8.528       |
> | kernel_syrk               | 106         | 156         | 18      | 1269     | 1871     | 639       | 7.897       |
> | kernel_gemver             | 175         | 253         | 33      | 2613     | 3495     | 1121      | 8.416       |
> | kernel_fdtd_2d            | 189         | 278         | 40      | 3897     | 4701     | 1618      | 8.681       |
> | kernel_3mm                | 196         | 287         | 26      | 1959     | 2656     | 988       | 7.704       |
> | kernel_gesummv            | 66          | 91          | 37      | 2731     | 2507     | 1134      | 8.625       |
> | kernel_mvt                | 97          | 140         | 14      | 1236     | 1565     | 537       | 7.943       |
> | kernel_lu                 | 108         | 180         | 22      | 4867     | 8407     | 1855      | 8.288       |
> | kernel_syr2k              | 116         | 171         | 18      | 1440     | 2119     | 705       | 8.189       |
> | kernel_adi                | 339         | 550         | 22      | 7624     | 10810    | 2769      | 9.236       |
> | **Mean**                  | **146.5**   | **218.4**   | **20.7**| **3470.0**| **5169.6**| **1335.4**| **8.3290**  |

---

> ### Author Response · Authors · 2024-11-23
> **Response to Reviewer deWG (3/3)**
>
> **[Weakness 3.]** The formula of the edge positional embedding in Table 3 is not very clear. The authors could improve the writing of this subsection.
>
> We appreciate the actionable suggestion. We have revised Section 4.2 and Table 3 (with highlight color indicating the revised parts) to include a more detailed description of edge PE.
>
> ------------------------
> **[Question 1.]** In Table 5, sometimes NPE hurts the performance. Why does it happen, and why is EPE more stable?
>
> **Response:** We apologize for not being sufficiently clear. The main discussion and detailed analysis of graph PE stability can be seen in [1]. Here, we give a brief overview:
>
> Directly concatenating eigenvectors as PE introduces two significant issues:
>
> 1. Lack of Basis Invariance: Eigenvalues are not unique for the same Laplacian, and Node PE does not enforce constraints ensuring that neural networks produce consistent outputs for different inputs corresponding to the same Laplacian.
>
> 2. Stability Concerns: Node PE lacks guarantees for stability. Stability, at a high level, implies that small perturbations to the input Laplacian should induce only limited changes in the resulting positional encodings.
>
> In contrast, stable EPE is basis invariant and stable. The key is the permutation equivariance as well as smoothness of $\kappa$, which ensures smooth weights $\kappa(\lambda)$ of eigenvectors' inner product across different eigenvalues.
>
> *[1]On the Stability of Expressive Positional Encodings for Graphs. Yinan Huang, William Lu, Joshua Robinson, Yu Yang, Muhan Zhang, Stefanie Jegelka, Pan Li.  ICLR 2024.*
>
> -------------------------
> **[Question 2.]** You selected one dataset for each task. For some of the tasks, there might be more existing datasets. Why did you select these specific datasets?
>
> **Response:** Thanks to reviewer deWG for their insightful question. We selected the datasets to represent hardware design tasks across various levels of abstraction from a bottom-up perspective (from the lowest device level, physical synthesis level to higher levels like mapping onto general-purpose hardware platforms) (See Section.3 for detail). The selected ones are representative of this content.

---

> ### Comment · Reviewer_deWG · 2024-11-26
>
> Thank you for your detailed reply! Your reply has addressed the third weakness (about the clarity of the edge positional encoding) and two questions. The updated version is clearer about it. Regarding the first weakness, I still think that using existing datasets might reduce the novel contribution. Regarding the second weakness, after reading your rebuttal, I agree that most of the existing HLS benchmark datasets are still much smaller than real-world examples, so I agree with you that the graph size is not a problem. However, I think not using HLS pragma is a limitation, because in the real-world application, we need to insert HLS pragmas in order to achieve a good HLS performance. If it does not have any pragma design, the dataset cannot be used to train a machine learning model to predict the HLS performance of a pragma design. The authors could consider including this limitation and the limitations mentioned by other reviewers in the paper.
>
> Overall, I think this benchmark has covered various levels of hardware datasets, provided a convenient toolbox, and conducted solid experiments, so I would like to maintain my score.

---

> ### Author Response · Authors · 2024-11-30
>
> Thank you very much for your constructive feedback and objective evaluation. We agree with your observation that, although new datasets are not strictly required given the "Call for Papers" policy, including them would certainly enhance the quality of this work. We are greatly encouraged by your recognition of the many positive aspects of this study. Indeed, one research paper can hardly excel in every aspect, but we are committed to further improving the manuscript in its next version.

---

### Comment · Area_Chair_PGy8 · 2024-11-25

Dear Reviewers,

This is a kind reminder that the dicussion phase will be ending soon on November 26th. Please read the author responses and engage in a constructive discussion with the authors.

Thank you for your time and cooperation.

Best,

Area Chair

---

### Meta-Review · Area_Chair_PGy8 · 2024-12-20

**Metareview:**

This paper proposes a benchmark for evaluating GNN models on hardware design and offers insights that GNN models utilizing bidirectional message-passing networks and robust positional encodings yield the best performance.

The most positive aspect of the paper is the comprehensive benchmark it presents, covering a wide range of practical hardware design tasks. However, the reviewers had some unresolved concerns during the discussion.

First, the insights provided in the paper are relatively limited. The use of bidirectional message passing and node/edge encodings is not a novel approach within the context of Deep Graph Representation Learning (DGRL). Second, the datasets used in this benchmark are provided by previous works, which diminishes the contribution of the paper to some extent.

Therefore, I recommend rejecting the paper and strongly encourage the authors' to revise the paper based on the review comments for resubmission.

**Additional Comments On Reviewer Discussion:**

Reviewers kctB, deWG, 8G85, F5ej rated this paper as 3: reject (keep the score), 8: accept (keep the score), 5: borderline reject (raised to 6), and 6: borderline accept (keep the score), respectively.

The reviewers raised the following concerns.

- Limited insights (raised by Reviewers kctB and F5eJ).

- previously used benchmarks (rasied by Reviewers kctB and deWG).

- scalability (rasied by Reviewer kctB).

- Insufficient experiments (rasied by Reviewer kctB and deWG).

- Unclear experiment details (rasied by Reviewer 8G85).

Through additional experiments and more detailed explanations in the rebuttal, the authors address some concerns regarding scalability, unclear experimental details, and insufficient experiments. However, certain fundamental weaknesses remain unaddressed in the rebuttal.

First, the insights provided in the paper are relatively limited. The use of bidirectional message passing and node/edge encodings is not a novel approach within the context of Deep Graph Representation Learning (DGRL). Second, the datasets used in this benchmark are provided by previous works, which diminishes the contribution of the paper to some extent.

Therefore, I will not recommend accepting this paper in its current state.

---

### Decision · Program_Chairs · 2025-01-22

Reject